# Circadian clock dysfunction in human omental fat links obesity to metabolic inflammation

Eleonore Maury [ORCID] [1✉], Benoit Navez[2] & Sonia M. Brichard[1]

To unravel the pathogenesis of obesity and its complications, we investigate the interplay between circadian clocks and NF-κB pathway in human adipose tissue. The circadian clock function is impaired in omental fat from obese patients. ChIP-seq analyses reveal that the core clock activator, BMAL1 binds to several thousand target genes. NF-κB competes with BMAL1 for transcriptional control of some targets and overall, BMAL1 chromatin binding occurs in close proximity to NF-κB consensus motifs. Obesity relocalizes BMAL1 occupancy genome-wide in human omental fat, thereby altering the transcription of numerous target genes involved in metabolic inflammation and adipose tissue remodeling. Eventually, clock dysfunction appears at early stages of obesity in mice and is corrected, together with impaired metabolism, by NF-κB inhibition. Collectively, our results reveal a relationship between NF-κB and the molecular clock in adipose tissue, which may contribute to obesity-related complications.

---

[1] Endocrinology, Diabetes and Nutrition Unit, Institute of Experimental and Clinical Research, UCLouvain, Brussels, Belgium. [2] Digestive Surgery Unit, Saint-Luc University Hospital, UCLouvain, Brussels, Belgium. ✉email: eleonore.maury@uclouvain.be

I n all multicellular organisms, transcriptional feedback loops are at the core of circadian (molecular) clocks[1]. The mammalian circadian clock network is programmed by transcription–translation feedback loops comprised of basic helix–loop–helix activators (CLOCK/BMAL1 heterodimers) that induce the transcription of their own repressors [PERIODs (PER), CRYPTOCHROMEs (CRY), REV-ERBs] through binding to E-box elements within the promoters, thereby synchronizing physiological rhythms[1,2]. PER/CRY protein complexes, in turn, repress CLOCK/BMAL1 activity thus creating a negative feedback loop[1]. In parallel, the coordinated action of REV-ERBα and the activator ROR modulates the rhythmic transcriptional regulation of Bmal1, while additional regulations involve the PAR-bZIP transcription factors [including D-box binding protein (DBP)][1]. CLOCK/BMAL1 heterodimers also drive the expression of thousands of clock-regulated genes that control a vast array of biological processes such as metabolism[3]. Moreover, CRYs and PERs can exert supplementary regulations independent of other core clock factors, adding a layer of complexity to this simplified model[4,5]. Consequently, environmental or genetic perturbations of the molecular clocks may lead to metabolic disorders[3]. For example, adipocyte or myeloid cell-specific deletion of Bmal1 results in an obese phenotype[6] or in inflammation and insulin-resistance phenotype[7], respectively. This may be explained by the fact that BMAL1 binds to target genes in a tissue-specific manner due to patterns of chromatin accessibility and interaction with transcription factors[8]. Moreover, nutritional challenges dysregulate the molecular clocks with a differential impact on each tissue. Thus, clock gene expression is severely altered in the peripheral tissues of animal models of obesity (challenged with High-Fat Diet, HFD), with the most significant changes occurring in white adipose tissue[2,9,10]. Such changes can be prevented or reversed when HFD-feeding is restricted to the active phase[11,12]. Importantly, time-restricted feeding may also lead to metabolic improvements in obese patients[13].

Human obesity is characterized by low-grade inflammation and accumulation of visceral rather than subcutaneous adipose tissue plays a crucial role in the development of co-morbidities[14,15]. Dysfunctional preadipocytes and adipocytes can trigger the accumulation of macrophages, neutrophils, lymphocytes, and mast cells, driving adipose tissue inflammation[16,17] and fibrosis[18–20]. Overall, those modifications are more pronounced in omental than in other fat depots[20–22].

Mounting evidence from studies in mice has demonstrated that obesity and associated complications (including cardiovascular disease, dysregulation of appetite, prothrombotic state, and insulin resistance) are mediated through metabolic inflammation induced by activation of the transcription factor NF-κB[23–25]. Moreover, a recent work has interconnected inflammation and the circadian clockwork in mice. NF-κB signaling has been found to modify chromatin states and modulate the expression of genes of the core clock network in mouse liver[2]. However, whether improper inflammation may contribute to metabolic disease through dysregulation of circadian systems remains, as yet, unsettled in humans. Given the key role of both BMAL1[6] and NF-κB[26] in adipocyte function, we therefore investigated the interplay between these respective pathways in omental adipocytes and precursors from obese subjects.

## Results

**Obesity is associated with an inflammatory state and altered clock gene expression in human omental adipose tissue.** The clinical and laboratory characteristics of non-obese and obese patients are depicted in Table 1. The average body mass index (BMI) of obese patients was almost twice that of non-obese

**Table 1 Clinical and laboratory characteristics of non-obese and obese patients.**

|  | Non-Obese | Obese |
|---|---|---|
| Age (years) | 51.8 ± 3.9 | 50.1 ± 3.4 |
| Number (sex ratio men/women) | 8 (3/5) | 8 (4/4) |
| BMI (kg/m²) | 24.9 ± 0.6 | 47.9 ± 2.9*** |
| Systolic blood pressure (cmHg) | 13.2 ± 0.6 | 14.1 ± 0.5 |
| Diastolic blood pressure (cmHg) | 8.0 ± 0.4 | 8.5 ± 0.2 |
| Fasting glucose (mmol/l) | 5.1 ± 0.1 | 6.9 ± 0.5** |
| HDL-cholesterol (mmol/l) | 1.3 ± 0.1 | 1.2 ± 0.1 |
| Total cholesterol/HDL | 2.9 ± 0.4 | 3.5 ± 0.3 |
| Triglycerides (mmol/l) | 1.1 ± 0.1 | 1.3 ± 0.1 |

Clinical and laboratory parameters were measured after an overnight fast before surgery. Data are presented as mean ± SEM with $n = 8$ patients per group (**$p < 0.01$, ***$p < 0.001$, unpaired two-tailed $t$-test).

subjects, and all obese patients exhibited several features of metabolic syndrome[27]. All non-obese subjects were euglycemic (fasting glucose <5.6 mmol/L), while obese patients were either recently diagnosed (untreated) with diabetes or had fasting glucose just below the diabetic threshold (≤7 mmol/L). All the obese patients also presented an elevated blood pressure (systolic ≥ 13 cm Hg and/or diastolic ≥8.5 cm Hg; Table 1). The lipid profile was not significantly different between the two groups.

We first showed that adipocytes and precursors participate to adipose tissue inflammation in obesity. Mature omental adipocytes (OAdipocytes) from obese subjects exhibited higher expression of pro-inflammatory cytokines (TNFα and IL6) and chemokines (CCL2, CCL4, and CCL5; Fig. 1A). Because clock gene expression is altered in inflamed adipose tissue of obese mice[2,9], we hypothesized that a relationship may exist between the inflammatory and molecular clock pathways in human obesity. We thus measured clock gene expression in OAdipocytes and found decreased levels of the clock repressors PER2 and REV-ERBα in obese subjects (Fig. 1B).

A similar pattern of inflammatory and clock gene expression occurred in omental adipocyte precursors (OAPs; PDGFRα+ CD34+D31−CD45− cells obtained from omental stromal-vascular cells, see Methods section). OAPs from obese patients exhibited higher expression of CD9, a marker associated with adipose tissue inflammation, fibrosis, and metabolic disorders[18,20] (Fig. 1C, D). Obese OAPs also exhibited higher expression of standard cytokines and chemokines, as well as higher NF-κB activity (Fig. 1E, F). Conversely, these cells displayed decreased expression of the repressors PER2 and REV-ERBα, while the expression of other core-clock genes was unchanged (except for RORα, Fig. 1G). These data mirror those found in adipocytes.

These findings were recapitulated in omental adipose tissue (i.e., undigested) where inflammation and downregulated expression of PER2 and REV-ERBα were consistently observed in obesity (Supplementary Fig. 1A–C). We next validated the fact that inflammation was more pronounced in omental than in subcutaneous fat (Supplementary Fig. 1D), and that clock gene expression was more severely altered in this depot (Supplementary Fig. 1E), thereby justifying the rationale for focusing on the omental fat depot throughout the study.

To further characterize the perturbation of the molecular clock in obesity, we used synchronized OAPs that stably expressed the circadian reporter Per2-dLuc, which allows real-time reporting of circadian dynamics[28]. Continuous monitoring of luciferase activity revealed significant period lengthening of Per2-dLuc oscillations in OAPs from obese individuals compared to non-obese subjects (Fig. 1H, I). Collectively, our results demonstrate clock dysfunction in omental adipose tissue of obese patients,

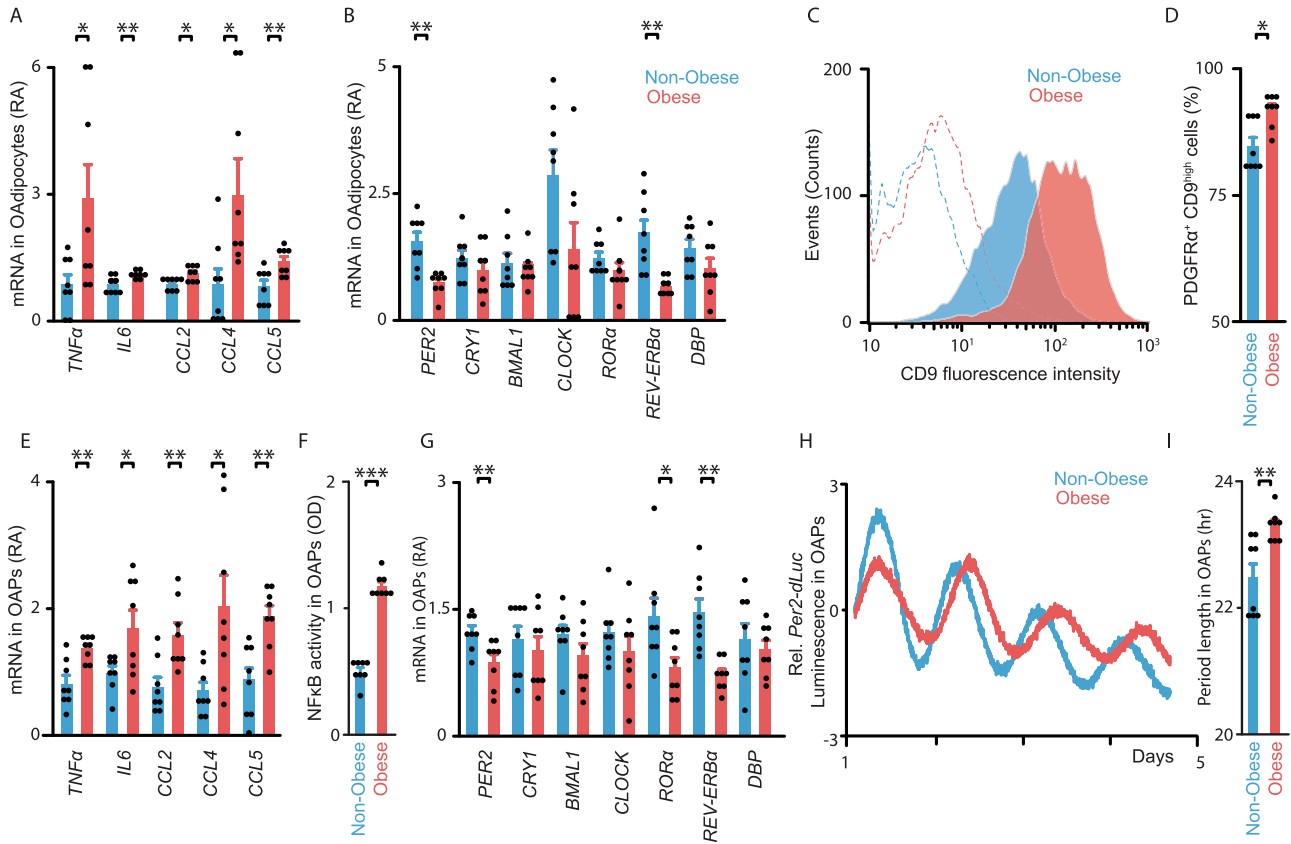

**Fig. 1 Expression of pro-inflammatory factors and circadian clock function are altered in human omental adipocytes and precursor cells. A**, **B** Expression of pro-inflammatory (cytokines and chemokines) (**A**) and core clock (**B**) genes in non-synchronized Omental Adipocytes (OAdipocytes) from non-obese (blue) or obese (red) patients. Values are displayed as mRNA Relative Abundance (RA). **C**, **D** Detection of the cell-surface glycoprotein CD9 in PDGFRα⁺ CD34⁺ CD31⁻ CD45⁻ adipocyte precursors (OAPs) from non-obese (blue) and obese (red) patients by flow cytometry. Phycoerythrin fluorescence (CD9) and Alexa Fluor red (CD140a) have been detected simultaneously, to determine the frequency of PDGFRα⁺ CD9^high cells. **C** One representative trace (i.e., OAPs from one patient) per condition is shown. Dashed lines are the negative controls. **D** Percentage of CD9^high OAPs in non-obese vs. obese patients. **E** Expression of pro-inflammatory genes in non-obese or obese OAPs. **F** NF-κB activity in non-obese or obese OAPs (phosphorylated p65, in Optical density, OD). **G** Expression of core-clock genes in non-obese or obese OAPs. **H** Normalized bioluminescence of *Per2-dLuc* reporter oscillations in synchronized OAPs obtained from non-obese (blue line) or obese (red line) patients. One representative trace per group is shown. **I** Period length of *Per2-dLuc* bioluminescence in hours (hr). All data are represented as mean ± SEM with n = 8 patients/ group. *p < 0.05, **p < 0.01, ***p < 0.01, unpaired two-tailed *t* test. See also Supplementary Fig. 1.

which might be due, at least in part, to alterations occurring in adipocytes and precursors.

**Obesity disrupts circadian clock function through NF-κB activation in OAPs**. As *PER2* expression was decreased, along with a period lengthening of its oscillations in obese OAPs, we hypothesized that *PER2* might be key event in clock dysfunction resulting from NF-κB activation. Sequence analysis identified putative NF-κB binding sites in the promoter regions of human core-clock genes (Supplementary Table 1). We used chromatin immunoprecipitation (ChIP) with antibodies directed against the p65 subunit of NF-κB to interrogate p65 binding to the promoter of *PER2* (Fig. 2A). We demonstrated the physical occupancy of p65 on a κB site within the *PER2* promoter and found that p65 binding to *PER2* was enhanced by almost 10 times in obese vs. non-obese OAPs (Fig. 2A). Thus, we hypothesized a direct role for NF-κB in the regulation of *PER2* transcription and investigated whether NF-κB could regulate *PER2* expression by modulating the transcriptional activity of BMAL1. This hypothesis was further supported by the comparative analysis of Gene Expression Omnibus (GEO) repository of p65 and BMAL1 ChIP-sequencing (ChIP-seq) data in human cell lines[29,30], revealing that p65 and BMAL1 peaks localize close to the *PER2*

Transcription Start Site (TSS) (Supplementary Fig. 2). Interestingly, by performing ChIP assays with antibodies directed against BMAL1, we found that BMAL1 binding to the E-box of *PER2* was significantly decreased in obesity (Fig. 2B) while NF-κB binding was increased (Fig. 2A). These results suggest that NF-κB impedes binding of the transcriptional activator BMAL1 to the *PER2* promoter, thereby decreasing *PER2* expression and impairing the circadian clock function.

We explored the hypothesis that decreased BMAL1 occupancy at the *PER2* promoter in obesity might also be associated with changes of RNA polymerase II (RNA POLII) recruitment. When RNA POLII is recruited for the initiation of mRNA synthesis, the carboxy-terminal domain (CTD) is phosphorylated on serine 5 (Ser-5P), a mark of the guanylyltransferase catalytic activity[31]. We thus performed ChIP with antibodies directed against Ser-5P of the RNA POLII CTD (to measure its occupancy and estimate RNA POLII initiation). We observed a decreased occupancy of RNA POLII at the *PER2* promoter in obesity (Fig. 2C), consistent with the decreased *PER2* expression in this condition (Fig. 1G). Such a difference was lost in gene coding regions (e.g., *Exon 19*, *E19*; but also in *3′ Untranslated Region UTR*) (Fig. 2C), which might be explained by Ser-5 dephosphorylation occurring during elongation[32].

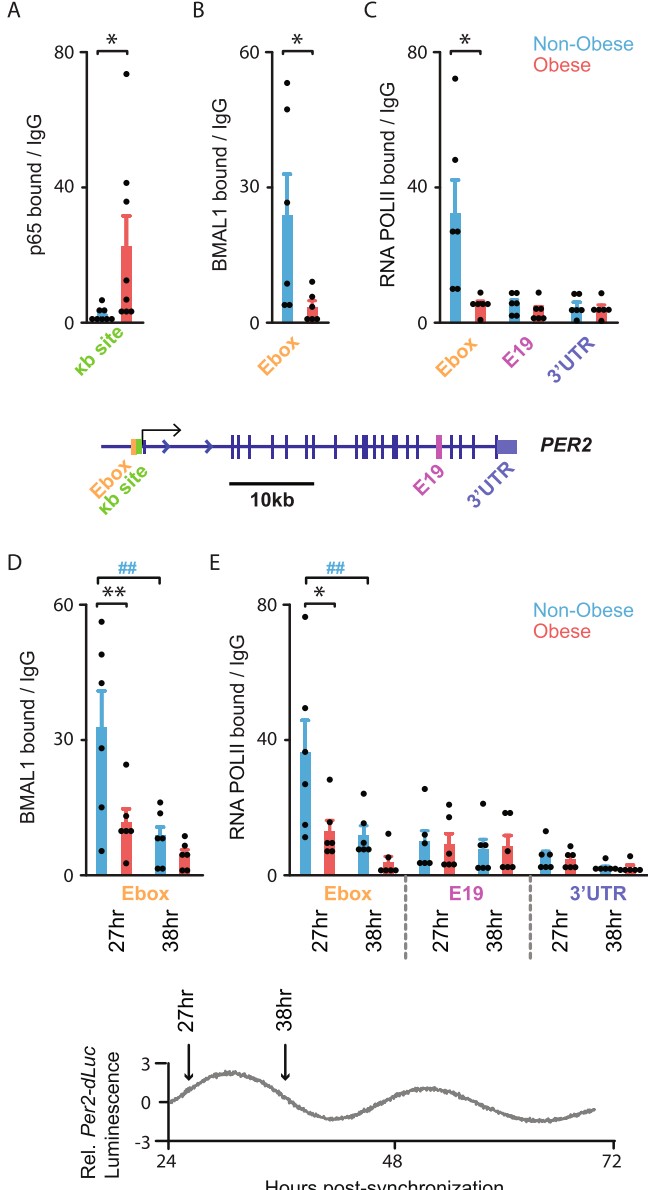

**Fig. 2 NF-κB p65 activation in human obesity results in decreased BMAL1-mediated transcription of _PER2_.** ChIP analyses of NF-κB p65, BMAL1, and RNA POLII on _PER2_ sites in OAPs, as indicated. Cells were either non-synchronized (**A–C**) or harvested 27 h and 38 h post-synchronization (**D**, **E**). **A** p65 bound to _PER2_ κB site in the promoter (~200 bp upstream to TSS, represented by a single arrow on _PER2_ gene), **B**, **D** BMAL1 to _PER2_ E-box (~400 bp upstream to TSS), **C**, **E** RNA POLII Ser-5P CTD repeat (RNA POLII) to _PER2_ E-box vs. _exon 19_ (_E19_) and _untranslated region_ (_UTR_) in OAPs from non-obese (blue) or obese patients (red). Collection times are depicted by 2 arrows on the inserted _Per2-dLuc_ reporter trace (bottom panel). Results are expressed in fold enrichment over IgG. Data are represented as mean ± SEM; $n = 8$ (**A**) or $n = 6$ (**B–E**) independent cultures (i.e., run at different times and for each time, from a new vial of cryopreserved OAPs) from 4 (**A**) or 3 (**B–E**) subjects in each non-obese and obese group. **A–C** *$p < 0.05$, **$p < 0.01$, unpaired two-tailed $t$-test (non-obese vs. obese comparisons); **D**, **E** the effect of BMI (non-obese/obese) and time was tested by two-way ANOVA followed by post hoc Sidak's test, *$p < 0.05$, **$p < 0.01$, unpaired measures (non-obese vs. obese), ##$p < 0.01$, paired measures (27 h vs. 38 h). See also Supplementary Fig. 2.

To further decipher BMAL1 dynamics at the chromatin level in inflamed adipose tissue, we examined its binding to the _PER2_ promoter in synchronized OAPs from obese and non-obese patients. To do so, we performed ChIPs for BMAL1 when BMAL1 binding enrichment was expected to be high (27 h post-synchronization) or low (38 h post-synchronization) based on the OAP _Per2-dLuc_ bioluminescence records (Fig. 1H) and previously described ChIP-seq data from different tissues[2,30,33]. We confirmed the time-dependent (rhythmic) binding of BMAL1 to _PER2_ E-box in non-obese OAPs (Fig. 2D). Meanwhile, we found dampened BMAL1 binding to _PER2_ E-box and failed to observe a significant rhythm of BMAL1 binding in obese OAPs (Fig. 2D). In parallel, we found a time-dependent recruitment of RNA POLII specifically to _PER2_ E-box in non-obese OAPs (Fig. 2E). However, RNA POLII recruitment to _PER2_ E-box was reduced and also failed to display a significant rhythmic pattern in obese OAPs (Fig. 2E), which might explain the downregulation of _PER2_ expression and its altered rhythm in obesity (Fig. 1G–I). The significantly different RNA POLII initiation in obese vs. non-obese OAPs, together with a different rhythm, suggests that the period of _PER2_ transcription, its amplitude or both might be different between the two conditions. In agreement, this period was significantly lengthened in obesity (Fig. 1H, I). Collectively, these data suggest that altered _PER2_ expression results from NF-κB antagonization of _PER2_ transcription through decreased chromatin binding of BMAL1 and RNA POLII.

**NF-κB inhibition improves circadian clock function in omental adipose tissue from obese patients.** To challenge the impact of NF-κB signaling on the adipose clock in humans, we took pharmacological or gene silencing approaches to inhibit NF-κB activity.

We first used two well-accepted NF-κB inhibitors, Bay11-7085[34] and JSH-23[35]. In non-synchronized OAPs from obese patients, both treatments decreased NF-κB activity (by 29% with Bay11-7085 and by 79% with JSH-23, Fig. 3A), while inducing a 6-fold increase in _PER2_ expression (Fig. 3B). This suggests that a small decrease in NF-κB binding is sufficient to reach a plateau of _PER2_ transcription. However, _BMAL1_ expression was unchanged (Supplementary Fig. 3A). In synchronized obese OAPs expressing the _Per2-dLuc_ reporter, both treatments resulted in a significant period shortening of _Per2-dLuc_ oscillations (~20 min with Bay11-7085 and >1 h with JSH-23, Fig. 3C–E). We next examined whether NF-κB inhibition could restore BMAL1 binding to the _PER2_ promoter in obese OAPs. Using the most potent NF-κB inhibitor (JSH-23), we observed a ~4-fold decrease in p65 binding to _PER2_ (Fig. 3F), associated with a ~4-fold increase in BMAL1 binding in OAPs (Fig. 3G). Taken together, these results demonstrate that pharmacological inhibition of NF-κB signaling restores BMAL1 binding to the _PER2_ promoter and reinstates the circadian clock period to non-obese levels (Fig. 1I for comparison). We found that JSH-23 treatment also increased _PER2_ expression ~3-fold in OAdipocytes (Fig. 3H), suggesting that the same mechanism occurs in mature fat cells.

To confirm the role of NF-κB on the molecular clock function in omental fat, we also used small interference RNA-mediated knockdown (siRNA) of p65 subunit (_RELA_) in obese OAPs. p65 knockdown was efficient as shown by decreased _p65_ mRNAs and NF-κB activity (−85 to 90% vs. control, Fig. 3I, J). Concomitantly, _PER2_ mRNA levels increased in p65 knockdown OAPs (by 81%, Fig. 3K), but this increase appeared to be relatively small compared to the one induced by NF-κB inhibitors (Fig. 3B). This apparent discrepancy was explained by compensatory

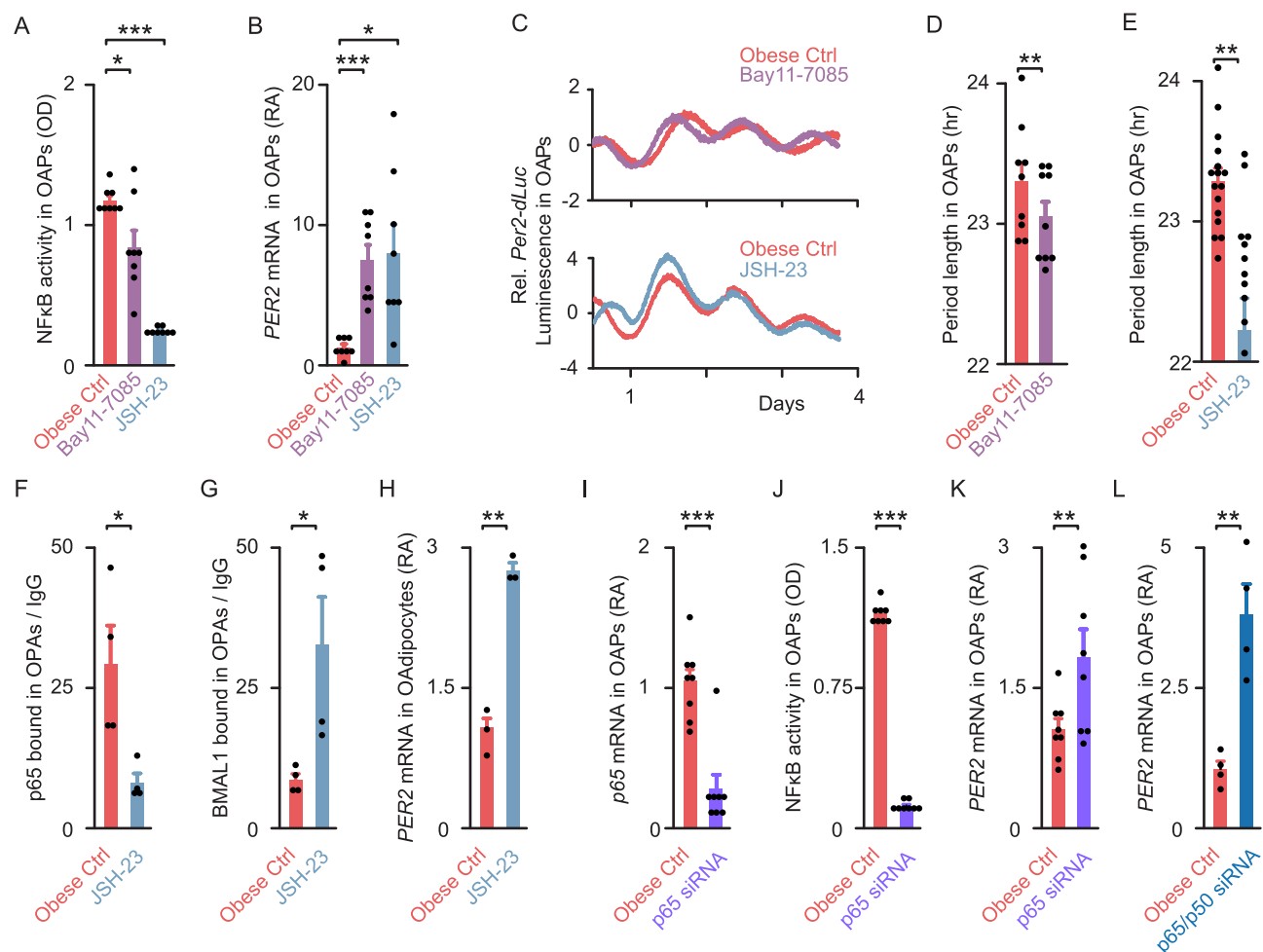

**Fig. 3 NF-κB inhibition restores circadian period and BMAL1 binding in adipocyte precursors from obese patients. A** NF-κB activity in non-synchronized obese OAPs treated with either DMSO (vehicle, Ctrl, red), 10 μM NF-κB inhibitor Bay11-7085 (purple) or JSH-23 (gray); n = 8 patients. **B** PER2 gene expression in non-synchronized obese OAPs treated with either DMSO (Ctrl), 10 μM NF-κB inhibitor Bay11-7085 or JSH-23. Values are displayed as mRNA RA compared to values of untreated obese Ctrl OAPs; n = 8 patients. **C–E** Normalized bioluminescence of Per2-dLuc reporter oscillations in synchronized obese OAPs following in vitro treatment with 10 μM NF-κB inhibitors Bay11-7085 or JSH-23, vs. DMSO (Ctrl). **C** One representative trace per condition is shown. **D**, **E** Period length of Per2-dLuc bioluminescence in hour; n = 9 (**D**) or n = 16 (**E**) independent cultures per treatment from 5 (**D**) or 8 (**E**) patients. **F**, **G** ChIP analysis of NF-κB p65 and BMAL1 on PER2 sites. **F** p65 bound to PER2 κB site and **G** BMAL1 bound to PER2 E-box in non-synchronized obese OAPs, either treated with 10 μM JSH-23 or untreated (DMSO, Ctrl). Results are expressed in fold enrichment over IgG; n = 4 independent cultures from two obese patients. **H** PER2 gene expression in non-synchronized obese OAdipocytes treated with DMSO (Ctrl) or JSH-23; n = 3 patients. **I** Expression of p65 in non-synchronized obese OAPs, transfected with siRNA against human p65 (RELA, violet) or non-targeting siRNA (Ctrl, red); n = 8 independent cultures from four patients. **J** NF-κB activity in obese OAPs transfected with either siRNA against human p65 or non-targeting siRNA; n = 8 patients. **K–L** PER2 gene expression in non-synchronized obese OAPs transfected with either siRNA against human p65 (RELA, violet), siRNA against p65/p50 (RELA/NFKB1, dark blue) or non-targeting siRNA (Ctrl, red); n = 4 independent cultures from two patients (**L**), or n = 8 independent cultures from four patients (**K**). Data are represented as mean ± SEM. *p < 0.05, **p < 0.01, ***p < 0.001, by one-way repeated ANOVA followed by post hoc Dunnett's test (**A**, **B**) or paired two-tailed t-test (**D–L**). See also Supplementary Fig. 3.

mechanisms due to other NF-κB subunits. When we performed a double knockdown of p65 and p50 subunits, PER2 mRNAs increased ~4-fold (Fig. 3L, with an additional 85% decrease in p50 mRNA levels, Supplementary Fig. 3B). We concluded that targeted inhibition of NF-κB activity, either through pharmacological or gene silencing approaches, was able to restore clock gene expression and function in obese OAPs, a mechanism that also occurred in mature adipocytes. Yet, the same pharmacologic or genetic tools failed to shorten the period of Per2-dLuc oscillations in non-obese OAPs (Supplementary Fig. 3C, D), which might be explained by the already low basal NF-κB activity.

Eventually, we used a reverse approach and overexpressed p65 in OAPs. p65 overexpression (OX) significantly increased NF-κB activity in obese OAPs (by 54%), while decreasing PER2 mRNA

(by ~40–70% at the two highest concentrations, Supplementary Fig. 3E, F). These changes were accompanied with a strong period lengthening in obese OAPs (~2 h at the highest concentration, Supplementary Fig. 3G, H). A trend towards a period lengthening was also induced by p65 OX in non-obese OAPs (Supplementary Fig. 3I). We concluded that unbalanced NF-κB activity is able to further alter inflammation and molecular clock function in omental adipose tissue in obesity.

**NF-κB-induced circadian dysfunction enhances CCL2 production in human omental adipose tissue.** The impact of circadian misalignment in human adipose tissue physiology and obesity is still largely unknown. We examined adipose tissue

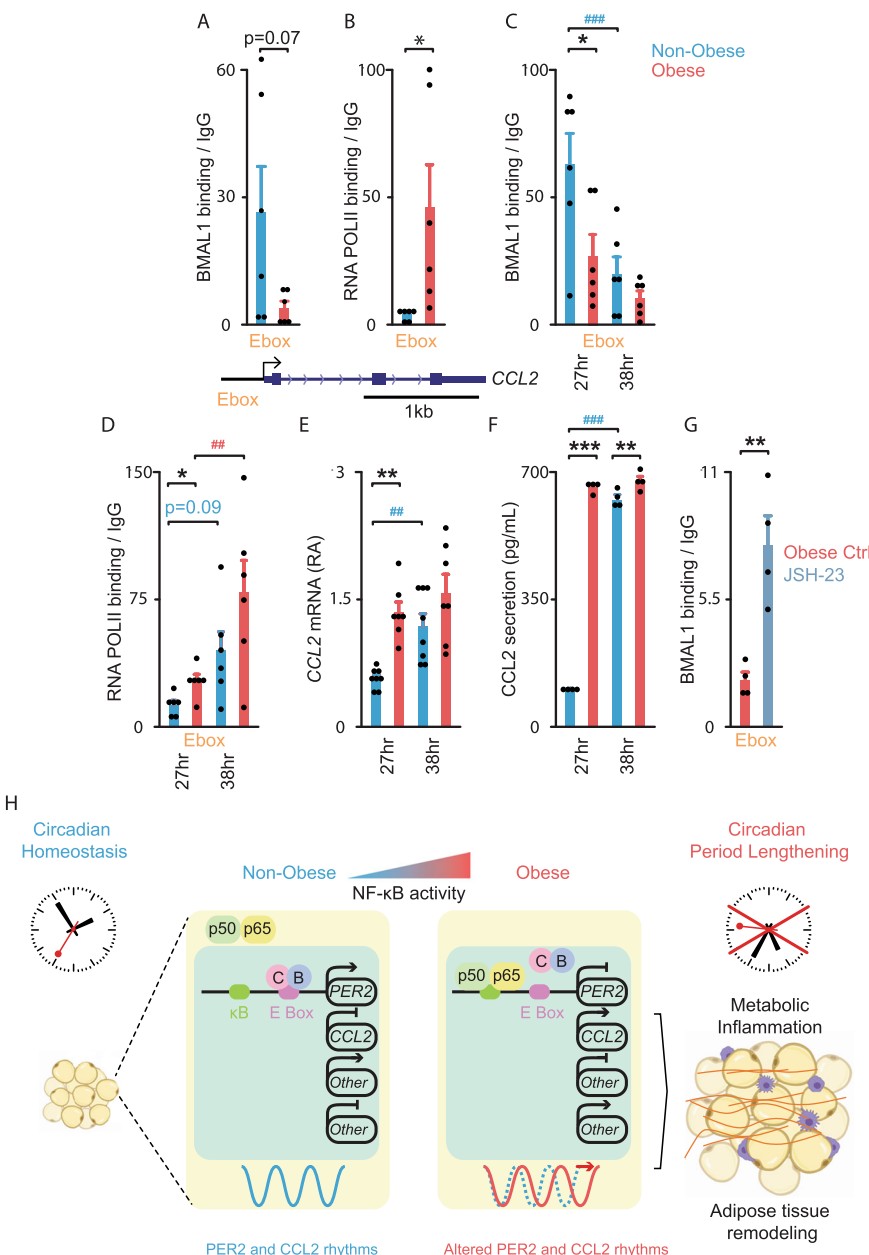

**Fig. 4 BMAL1-mediated transcription of *CCL2* is altered in adipocyte precursors from obese patients. A–D** ChIP analyses of BMAL1 and RNA POLII on *CCL2* promoter in non-obese (blue) or obese (red) OAPs, either non-synchronized or harvested 27 h and 38 h post-synchronization. Mean ± SEM in fold enrichment over IgG; $n = 6$ independent cultures/group from three patients in each non-obese and obese group. **E** *CCL2* gene expression in synchronized non-obese vs. obese OAPs. Mean ± SEM in mRNA RA, $n = 8$ (non-obese) and $n = 7$ (obese) independent cultures from four patients in each group. **F** Measurement of CCL2 secretion in synchronized non-obese vs. obese OAPs, between 27–33 h and 38–44 h post-synchronization. Mean ± SEM in pg/ml; $n = 4$ patients/group. **G** ChIP analyses of BMAL1 in non-synchronized obese OAPs exposed to 10 µM JSH-23 (gray) vs. untreated (DMSO, Ctrl, red). Mean ± SEM in fold enrichment over IgG; $n = 4$ independent cultures from two patients. *$p < 0.05$, unpaired two-tailed *t*-test (**A**, **B**); **$p < 0.01$, paired two-tailed *t*-test (**G**); The effect of condition and time was tested by two-way ANOVA followed by post hoc Sidak's test (**C–F**), *$p < 0.05$, **$p < 0.01$, ***$p < 0.001$ for unpaired measures (non-obese/obese), #$p < 0.05$, ##$p < 0.01$, ###$p < 0.001$ for paired measures (27 h vs. 38 h or 27–33 h and 38–44 h).

**H** Proposed model of obesity linking circadian clock dysfunction in human omental fat and metabolic inflammation. The molecular clock is encoded by transcription–translation feedback loops composed of activators BMAL1 (B) and CLOCK (C) that induce the transcription of repressors, the most important being *PER2*. PER2 feeds back to inhibit the forward limb. This process generates a rhythm of ~24 h. In obesity, NF-κB activity is increased: increased NF-κB p65 binding prevents BMAL1 binding to *PER2* E-box, thereby reducing BMAL1-mediated transcription of *PER2* and inducing lengthening of the circadian period. The concomitant decreased BMAL1 binding to *CCL2* leads to actively transcribed *CCL2* and subsequent upregulated CCL2 secretion. In obesity, increased or decreased BMAL1 binding at other loci was associated with an altered production of the targets ("Other"; BMAL1 exerting a transactivator/repressor role, see infra), contributing to metabolic inflammation and adipose remodeling (macrophages, purple; fibrosis, red). See also Supplementary Fig. 3.

inflammation and focused on CCL2 production because of its critical role in the accumulation of adipose tissue macrophages in obesity and its potential impact on insulin resistance[16,36,37]. Using ChIP assays, we observed a trend towards a decrease in BMAL1 binding to the *CCL2* promoter (Fig. 4A) and a 10-fold increase in RNA POLII recruitment in obese vs. non-obese OAPs (Fig. 4B). This was consistent with the increased *CCL2* expression in obesity (Fig. 1E) and associated with an increase in p65 occupancy at the κB sites located near the TSS of *CCL2* in obese OAPs (Supplementary Fig. 3J). Both BMAL1 and RNA POLII occupancy at the *CCL2* promoter were significantly different between non-obese/obese in synchronized OAPs (27 hr post-synchronization, Fig. 4C, D), coinciding with an increased *CCL2* expression and protein secretion (Fig. 4E, F). Taken together, these data suggest a repressive role of BMAL1 on *CCL2* transcription in OAPs, which is altered in obesity. Moreover, the significant time-dependent binding of BMAL1 and RNA POLII to the *CCL2* promoter (Fig. 4C, D) was consistent with the rhythmic expression of *CCL2* in OAPs (Fig. 4E). However, BMAL1 binding rhythms (Fig. 4C), as well as *CCL2* expression rhythms (Fig. 4E) were dampened in obese OAPs. As a result, the rise in CCL2 secreted between the 2 collection times was small in obese OAPs (~8%, Fig. 4F), but was marked in non-obese OAPs (more than 6-fold, Fig. 4F).

Finally, using JSH-23, we demonstrated that NF-κB inhibition could restore the binding of BMAL1 to the *CCL2* promoter in obese OAPs. We observed a ~4-fold increase in BMAL1 binding in OAPs (Fig. 4G). Accordingly, the inhibition of NF-κB decreased *CCL2* expression up to 50% with the pharmacological treatments (Supplementary Fig. 3K) and by ~33% with *p65* siRNA (Supplementary Fig. 3L), while *p65* overexpression in obese OAPs significantly increased *CCL2* mRNA (more than 4-fold with the highest concentration, Supplementary Fig. 3M). These results confirmed that NF-κB-induced circadian disturbances lead to excessive *CCL2* expression in OAPs (proposed model in Fig. 4H) as well as in mature OAdipocytes (Figs. 1A, B and 3H).

**Obesity repositions BMAL1 binding genome-wide in human omental adipocyte precursors**. To decipher the regulatory roles of BMAL1 in human omental adipose tissue, we profiled its chromatin binding genome-wide in OAPs (Fig. 5). To do so, we performed ChIP-seq of BMAL1 in OAPs from 3 non-obese and 3 obese (Class III) patients who were representative of each group from Table 1. We identified peaks that were enriched with BMAL1 antibody over input chromatin and found 3927, 2188, and 3284 peaks in non-obese OAPs compared with 1778, 2236, and 2941 peaks in obese OAPs. Tag distributions across TSS target regions have been presented as average plots, showing the strong enrichment in BMAL1 occupancy among non-obese and obese OAPs at TSS regions (Fig. 5A). On average, BMAL1 binding to promoters clusters at the annotated TSSs of target genes (Fig. 5A). In all samples obtained from non-obese and obese patients, we found an extremely significant over-representation of the canonical E-box motif (CACGTG), a DNA response element bound by CLOCK/BMAL1[38,39], in 76–81% of the targets (Fig. 5B). BMAL1 peaks located on clock repressors were found in non-obese and obese OAPs (Fig. 5C–F and Supplementary Fig. 4A), although tag density was - or tended to be - reduced in obese subjects (e.g., *PER2* by ~−12%, $P = 0.07$, Fig. 5D; *REV-ERBα* by ~−22%, $P < 0.05$, Fig. 5F). This is consistent with conventional ChIP data (at least for *PER2*, Fig. 2B) and further supports the concept that reduced expression of *PER2* and *REV-ERBα* (Fig. 1G) could result from the decreased binding of the transcriptional activator BMAL1 in obese OAPs. BMAL1

peaks were also identified on *CCL2* (Supplementary Fig. 4B), but not on other pro-inflammatory factors, like *TNFα, IL6, CCL4*, or *CCL5* (not shown). We next examined whether obesity was associated with changes in BMAL1 binding genome-wide. Although a large proportion of BMAL1 peaks were overlapping, a number of peaks were specific to non-obese OAPs (1124), while others were unique to obese ones (513), indicating that obesity caused a genome-wide relocalization of BMAL1 binding (Fig. 5G, details in Supplementary Fig. 4C). A recent work performed in mice has reported the colocalization of NF-κB and BMAL1 chromatin binding in liver[2], therefore we sought to examine whether this observation could be extended to human adipose tissue. We found that BMAL1 also binds in close proximity to NF-κB consensus motifs in human OAPs (Fig. 5H), suggesting that this pathway might regulate many BMAL1 targets in these cells.

Among the regions with differential BMAL1 binding between non-obese and obese OAPs, we found a stronger binding enrichment of BMAL1 at genes coding for proteins involved in vesicular trafficking and/or cilium function (e.g., *Flotillin1, FLOT1; Tubby bipartite, TUB*) and for collagen chains or collagenases (e.g., *COL13A1; Matrix Metalloproteinase 2, MMP2*) in non-obese OAPs (Fig. 6A). Instead, genes like those encoding metabolic enzymes (e.g., glycolysis enzyme *Pyruvate dehydrogenase kinase isoform 2, PDK2*; proteases *MINDY Lysine 48 Deubiquitinase 2, MINDY2*) or factors regulating transcription (e.g., *Nuclear Repressor Corepressor 2, NCOR2*) exhibited higher BMAL1 occupancy in obese OAPs (Fig. 6B). Yet, this list was not exhaustive (see Fig. 5G). Among the differential regions identified by DESeq2 with false discovery rate- adjusted *p*-value (p.adj) <0.1, we found genes coding for proteins involved in extracellular matrix remodeling [*Lysyl Oxidase Homolog 2, LOXL2*, and *Plasminogen Activator Inhibitor-1, PAI1 (SERPINE1)*] (Fig. 6C, D). Obesity also altered BMAL1 binding enrichment at additional targets, further reinforcing the concept of BMAL1 repositioning (Fig. 6D). More specifically, the most robust BMAL1 peak (see the maximum track height on the *Y*-axis) that was found in both non-obese and obese OAPs corresponded to a gene coding for the activating chromatin modifier *Histone-lysine N-methyltransferase 2A, MLL1 (KMT2A)*; yet, BMAL1 peak value at this locus was lower in obesity (Fig. 6D, by ~−14%, $P < 0.05$). A similar trend was observed for the gene encoding glutathione peroxidase 1 (*GPX1*), one of the most important antioxidant enzymes (Fig. 6D, $P = 0.07$).

We next validated these ChIP-seq data. To this end, we selected some targets that may be involved in the pathogenesis of obesity (from Fig. 6) and we confirmed the obesity-induced changes in BMAL1 binding by using conventional ChIP ($n = 4$ patients/group; Fig. 7). Next, we measured target gene expression and protein levels in our whole cohort of patients ($n = 8$/group; Table 1 and Fig. 7). Strikingly, changes in BMAL1 occupancy were associated with qualitatively similar changes in target mRNA and protein levels (Fig. 7) except for *PAI1*. Thus, BMAL1 may act as a transcriptional activator for most of these target genes. Yet, for *PAI1*, BMAL1 may repress transcription as decreased BMAL1 binding was associated with higher *PAI1* transcript and protein levels in obese OAPs.

We further explored ChIP-seq data by using the KEGG pathway database: we performed functional gene pathway analysis of the sites bound by BMAL1. As expected, the analysis established a significant enrichment of factors within the circadian rhythms in both conditions (Fig. 8A). Besides circadian systems, the analysis revealed a significant enrichment of factors involved in many other functional pathways, and especially in endocytosis in non-obese patients (Fig. 8A). Proteoglycans and FOXO signaling pathways, both known to regulate adipose tissue

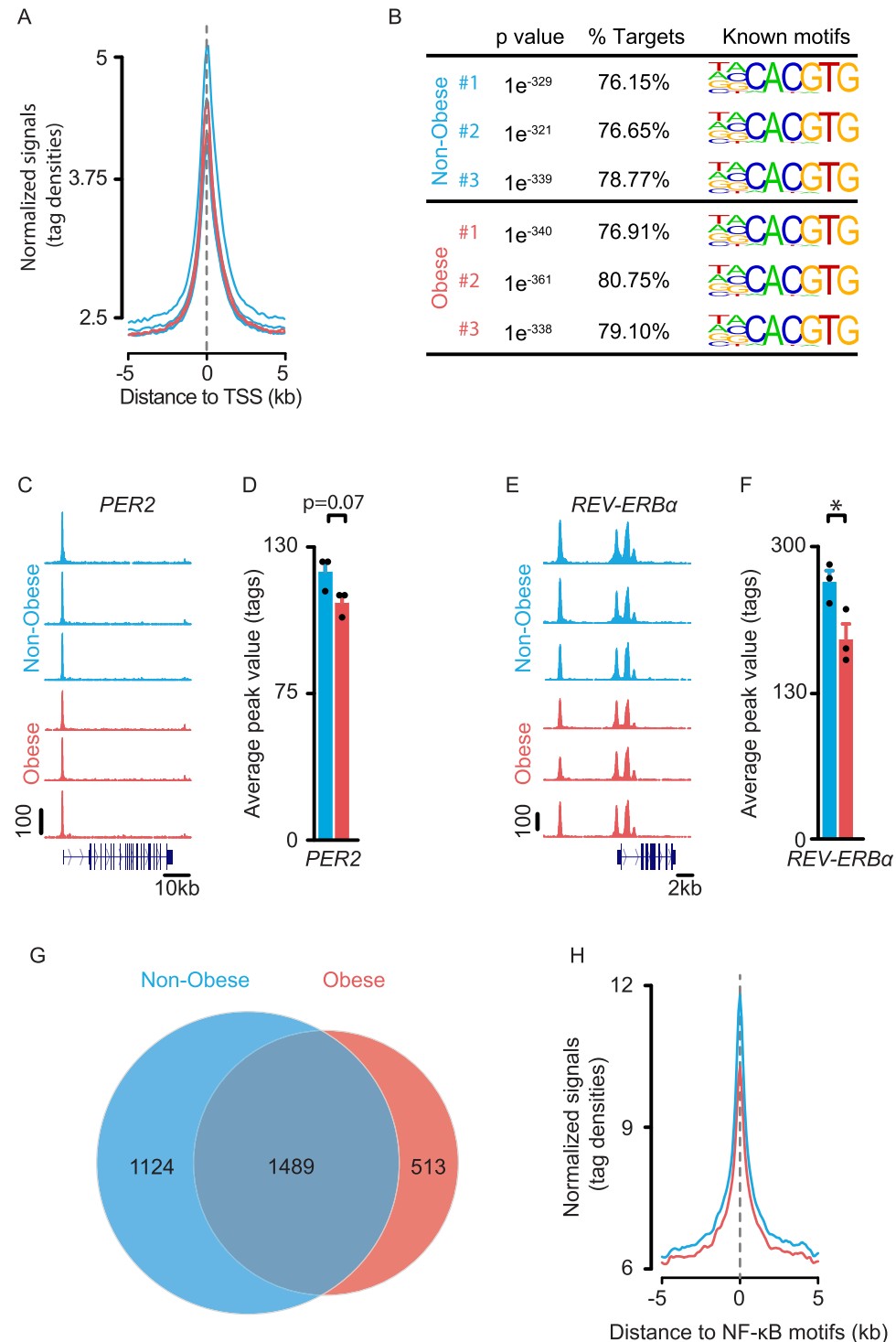

**Fig. 5 Obesity repositions BMAL1 binding genome-wide in human adipocyte precursors. A** Average plot of BMAL1 ChIP-seq reads on the target gene promoters near TSS in OAPs from non-obese ($n = 3$; blue) and obese (Class III) patients ($n = 3$; red). **B** Top known HOMER motifs enriched at BMAL1-binding sites from ChIP-seq analysis in OAPs from non-obese and obese patients. The highest ranking motif for each patient is shown and corresponds to the canonical E-box motifs. Significant motif enrichment was determined using a cumulative binomial statistical test. **C–F** University of California at Santa Cruz (UCSC) genome browser images of BMAL1 ChIP-seq tracks at clock repressors and related histograms. UCSC genome browser images of BMAL1 ChIP-seq tracks at *PER2* (**C**) and *REV-ERBα* (**E**). Normalized tag counts are indicated on the *Y*-axis and maximum track height is the same for all samples ($n = 3$ subjects/group). The orientation for each gene is indicated below each browser track. **D, F** Respective histograms depicting the average peak value per condition are also represented. Data are means ± SEM, *$p < 0.05$, by unpaired two-tailed *t*-test. **G** VENN diagram depicting the number of BMAL1 peaks and overlap in non-obese and obese OAPs (Raw data in OAPs from 3 patients/ group are provided in Source Data). **H** Average plot of BMAL1 ChIP-seq reads depicting their distance to NF-κB consensus motifs on target genes. NF-κB sites were obtained from UCSC Genome Browser's Factorbook data, which were lifted over from hg38. The normalized signal files for each group (average $n = 3$ subjects/group) were merged to show that BMAL1 binds in close proximity to NF-κB consensus motifs. See also Supplementary Fig. 4.

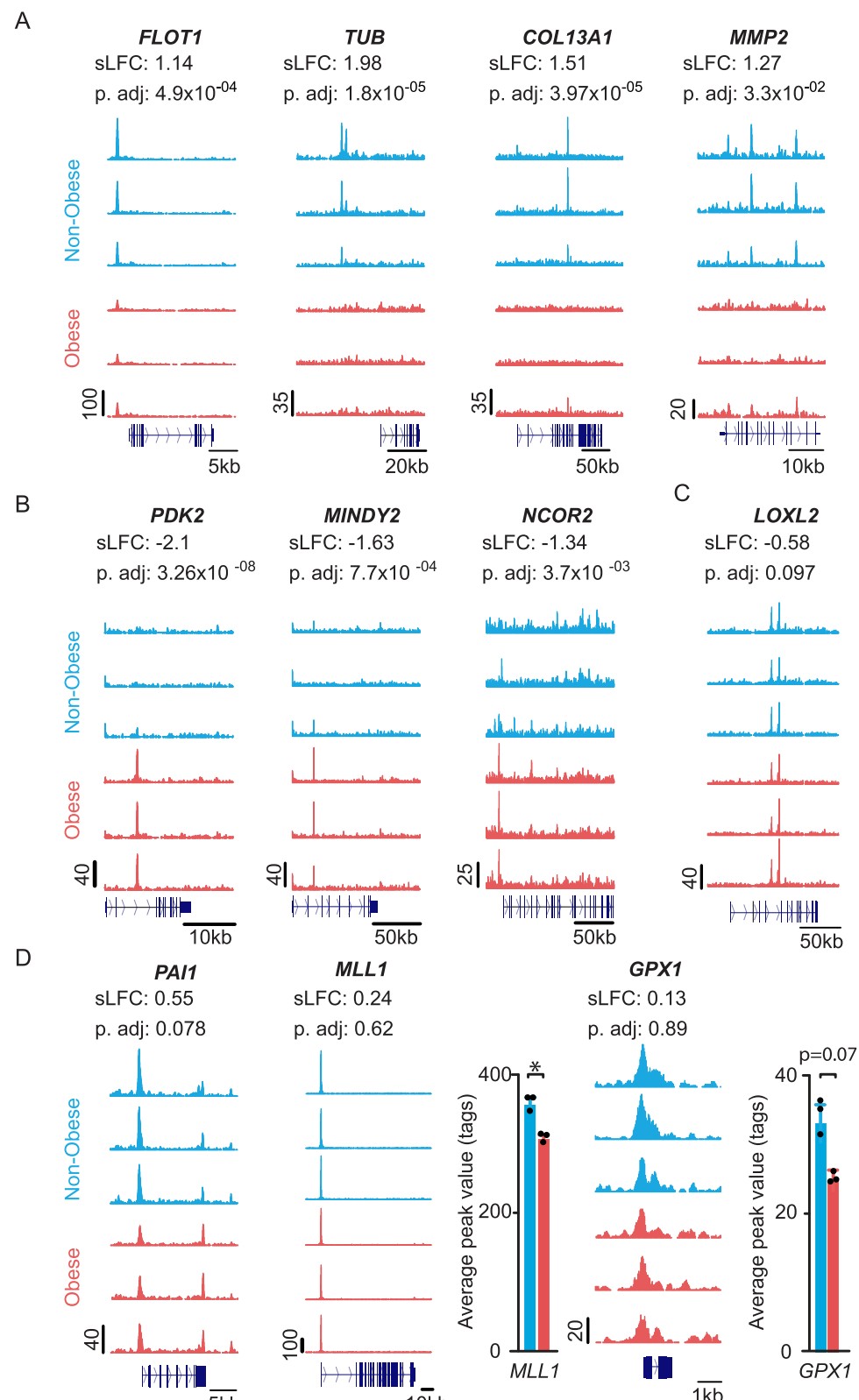

function[18,40] were also overrepresented in both obese and non-obese subjects (Fig. 8A). Similarly, gene ontology analysis revealed that bound targets of BMAL1 in either non-obese or obese OAPs are enriched in genes important for canonical circadian functions ("rhythmic process" in non-obese OAPs, or the subcategory "circadian regulation of gene expression" in obese

OAPs, Fig. 8B). However, the most significant biological processes were related to cell growth and response to oxygen levels in non-obese OAPs and to regulation of cellular amide metabolic process and RNA metabolic process in obese OAPs (Fig. 8B). Collectively, these data support an important regulatory role for BMAL1 in OAPs.

**Fig. 6 Examples of targets differentially bound by endogenous BMAL1 in OAPs from non-obese and obese patients. A–D** UCSC genome browser images of BMAL1 ChIP-seq tracks at clock repressors in OAPs ($n = 3$ subjects/group). Normalized tag counts are indicated on the Y-axis. The differential regions were identified using DESeq2, with shrunken Log$_2$ Fold Change (sLFC) non-obese/obese and false discovery rate- adjusted p-value (p.adj), indicated above each browser track. The orientation for each gene is indicated below each browser track. **A** Targets with specifically high BMAL1 occupancy in non-obese OAPs. **B** Targets with specifically high enrichment in obese OAPs. **C, D** Additional differential BMAL1 targets with p.adj < 0.1 or average peak value <0.1. Respective histograms depicting the average peak value per condition are represented for *MLL1* and *GPX1*, with data expressed as means ± SEM, *$p < 0.05$, by unpaired two-tailed *t*-test. *FLOT1* Flotillin1, *TUB* Tubby bipartite, *COL13A1* Collagen type XIII Alpha chain 1, *MMP2* Matrix Metalloproteinase 2, *PDK2* Pyruvate dehydrogenase kinase isoform 2, *MINDY2* MINDY Lysine 48 Deubiquitinase 2, *NCOR2* Nuclear Repressor Corepressor 2, *LOXL2* Lysyl Oxidase Homolog 2, *PAI1* Plasminogen Activator Inhibitor-1, Histone-lysine N-methyltransferase 2A, *MLL1* glutathione peroxidase 1, *GPX1*. See also Supplementary Fig. 4.

**High-fat diet-induced adipose clock dysfunction in mice is prevented or rescued by in vivo NF-κB inhibition.** To elucidate the relevance of adipose clock dysfunction in vivo, we used a genetic model (*mPer2$^{Luc}$* mice[28]) that allowed us to directly monitor the molecular clocks in both epididymal adipocyte precursors (APs) and mature adipocytes. Both cell types display a self-sustained circadian oscillator (Supplementary Fig. 5). We sought to determine whether high-fat feeding-induced obesity impacts the circadian clock system in these mouse cells as observed in human biopsies. We therefore fed *mPer2$^{Luc}$* mice either a HFD or Low-Fat Diet (LFD) for 3 months. As expected, HFD-fed mice gained more weight than LFD-fed mice (Supplementary Fig. 6A) and the epididymal adipose tissue (AT) depot was increased (Supplementary Fig. 6B). This condition was associated with elevated circulating glucose and insulin levels (Supplementary Fig. 6C, D), suggesting the presence of insulin resistance. The expression of pro-inflammatory cytokines and chemokines was significantly increased in AT from HFD-fed mice (Fig. 9A) at Zeitgeber Time 8 (ZT 8, 8 h after lights on), which corresponds to both elevated NF-κB response and BMAL1 binding[2]. We further demonstrated that NF-κB activity was significantly increased in AT in response to HFD (by ~40%; Fig. 9B), while the expression of the circadian repressors *Per2, Rev-erbα* and *Dbp* was downregulated (Fig. 9C). The low *Per2* expression in AT of obese mice could be explained, at least in part, by the marked decrease of *Per2* expression in both APs and adipocytes (Fig. 9D, E, respectively). Continuous monitoring of PER2::luciferase fusion protein revealed significant period lengthening of oscillations in both AT and APs from HFD- compared to LFD-fed *mPer2$^{Luc}$* mice (almost 4 h in AT and 2 h in AP, Fig. 9F, G). Moreover, a significant period lengthening was already detectable in adipose explants after 1 month of HFD challenge (Fig. 9H), when *Ccl2* expression was still unchanged (Fig. 9I). Similarly, other genes identified as BMAL1 targets were altered after 3 months of HFD, but were unchanged after 1 month (Fig. 9I). Specifically, mRNA levels of *Mmp2* were significantly downregulated while those of *Loxl2* and *Pai1* were significantly upregulated after 3 months of HFD. Yet, these mRNA levels were still unchanged after 1 month (Fig. 9I). Together, these results indicate that clock dysfunction precedes the regulation of genes encoding proteins involved in obesity pathogenesis, suggesting that it might initiate these alterations.

Consistently, treatment of explants or APs from HFD-fed mice with NF-κB inhibitors ex vivo improved clock function (Supplementary Fig. 7). Collectively, these data indicate that the mechanisms for clock dysfunction occurring in epididymal or omental adipose tissue might be common in obese mice and humans.

Next, we investigated whether clock dysfunction induced by HFD could be prevented by NF-κB inhibition in vivo. For this purpose, we used two approaches targeting the IKKβ/NF-κB signaling pathway in AT of HFD-fed mice. First, we used a pharmacological approach. To this end, *mPer2$^{Luc}$* mice that were

on HFD (vs. LFD) for 1 month were simultaneously treated with the anti-inflammatory drug sodium salicylate, which is known to inhibit IKKβ and NF-κB[41–43]. Salicylate treatment did not affect weight gain in HFD-fed mice (Supplementary Fig. 6E), but decreased circulating glucose and insulin levels (Supplementary Fig. 6F, G). Upregulation of AT NF-κB activity was also prevented by salicylate (Fig. 9J), while the downregulation of *Per2* expression was attenuated (Fig. 9K), suggesting that this drug prevents clock dysfunction. Yet, salicylate was not tested in fat from control mice. Second, we used a genetic approach. To establish that IKKβ and NF-κB directly target clock function specifically in mature adipocytes, we generated animals with conditional inactivation of the NF-κB regulatory kinase (IKKβ) restricted to adipocytes. These mice received HFD for 3 months, then inactivation was induced by tamoxifen (oil being used as control). Next, we examined gene expression in AT of oil- (control) and tamoxifen-treated adipocyte *Ikkβ*-KO mice[23,44]. Although both groups of mice had similar body weight, AT mass, and circulating glucose levels (Supplementary Fig. 6H–J), tamoxifen-treated mice displayed a decrease in *Ikkβ* mRNA levels (Supplementary Fig. 6K) and an increase in *Per2* expression (Fig. 9L), suggesting that the IKKβ/NF-κB pathway plays a key role in the regulation of *Per2* in adipocytes. Expression of *Ccl2* and the macrophage marker *F4/80* was significantly decreased in AT from these treated mice (Fig. 9L), in line with studies reporting the contribution of CCL2 to adipose tissue macrophage recruitment[37] and proliferation[36]. Expression of genes encoding extracellular matrix remodeling components was also significantly improved in AT from treated mice (Supplementary Fig. 6L–N). Collectively, our data indicate that adipose clock dysfunction occurs shortly upon high-fat feeding and that this process might be mediated through the IKKβ/NF-κB pathway in mouse AT, especially in APs and in mature adipocytes.

## Discussion

Circadian rhythms in metabolic functions are essential to metabolic homeostasis. Mice with whole-body or tissue-specific genetic disruption of the circadian clock show increased predisposition to obesity, chronic inflammation, and metabolic diseases[3,6,7].

To date, the few studies which have investigated the core-clock machinery in human adipose tissue have mostly explored the subcutaneous depot[45,46]. Yet, omental rather than subcutaneous fat plays a crucial role in the pathogenesis of the metabolic syndrome[14,15]. Herein, we focused on the omental depot, which when compared to the subcutaneous one, exhibits higher inflammatory state, along with more pronounced inhibition of clock repressors. We found an interplay between the circadian clock and NF-κB activation in human omental fat, thereby extending mouse liver data[2]. This interplay is enhanced in obese subjects: overactivation of NF-κB prevents BMAL1 binding to the promoter region of *PER2* and its subsequent transcription. Yet, *PER2* plays a determinant role in setting the period of the

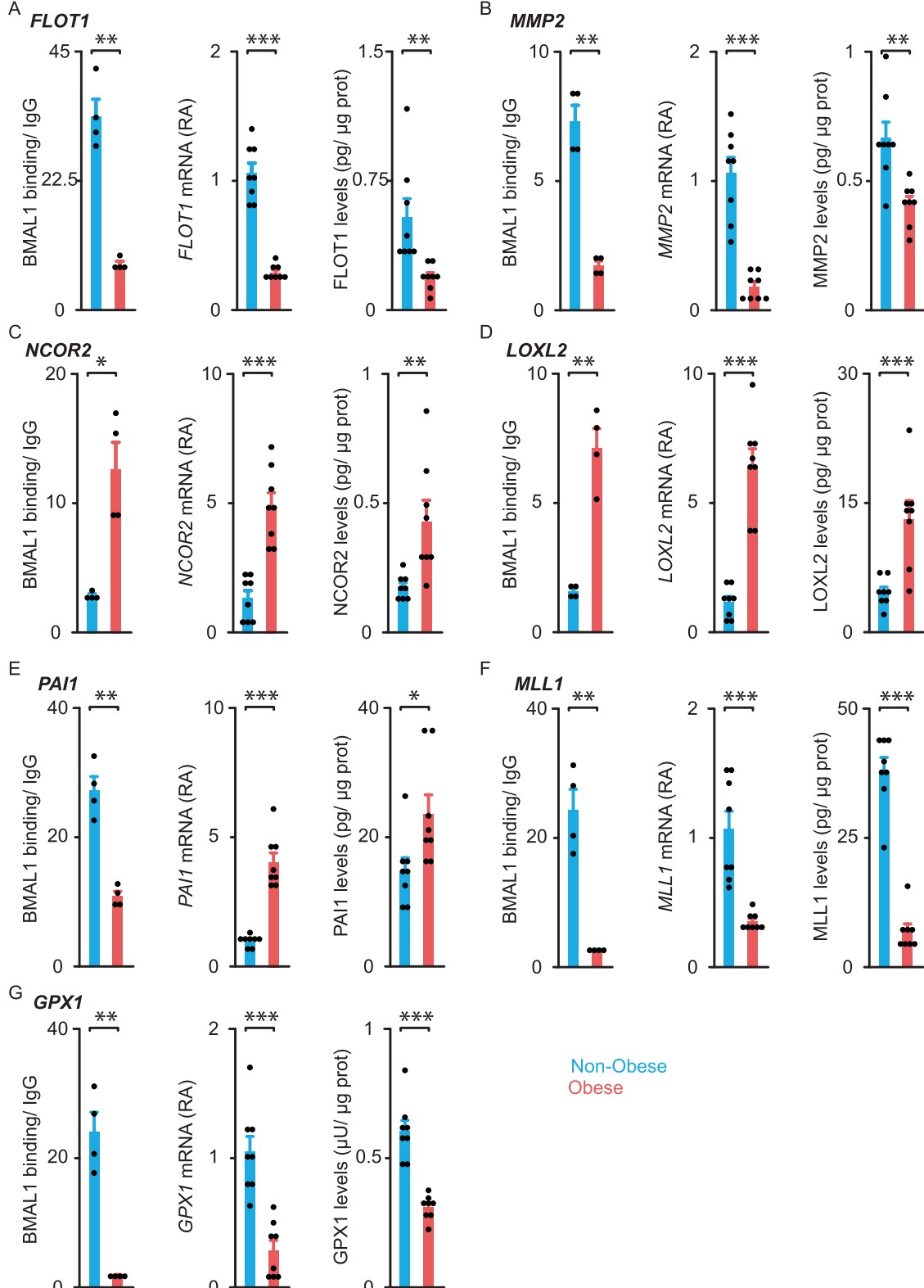

**Fig. 7 BMAL1 chromatin binding to selected targets, their transcript, and protein levels in human omental adipocyte precursors from non-obese and obese subjects.** We selected from Fig. 6 some BMAL1 targets for their relevance in the pathogenesis of obesity and performed, for each of them, conventional ChIP analyses, and measurements of mRNA and protein levels in OAPs. **A** FLOT1 Flotillin1, **B** MMP2 Matrix Metalloproteinase 2, **C** NCOR2 Nuclear Repressor Corepressor 2, **D** LOXL2 Lysyl Oxidase Homolog 2, **E** PAI1 Plasminogen Activator Inhibitor-1, **F** MLL1 Histone-lysine N-methyltransferase 2A, **G** GPX1 glutathione peroxidase 1. ChIP data are expressed as fold enrichment over IgG, mRNAs as relative abundance (RA), and protein levels in pg or μU/μg total proteins. Histograms represent the mean ± SEM for 4 (ChIP) or 8 (mRNA and protein) patients per group. p < 0.05, **p < 0.01, ***p < 0.001 by unpaired two-tailed t-test. See also Supplementary Fig. 4.

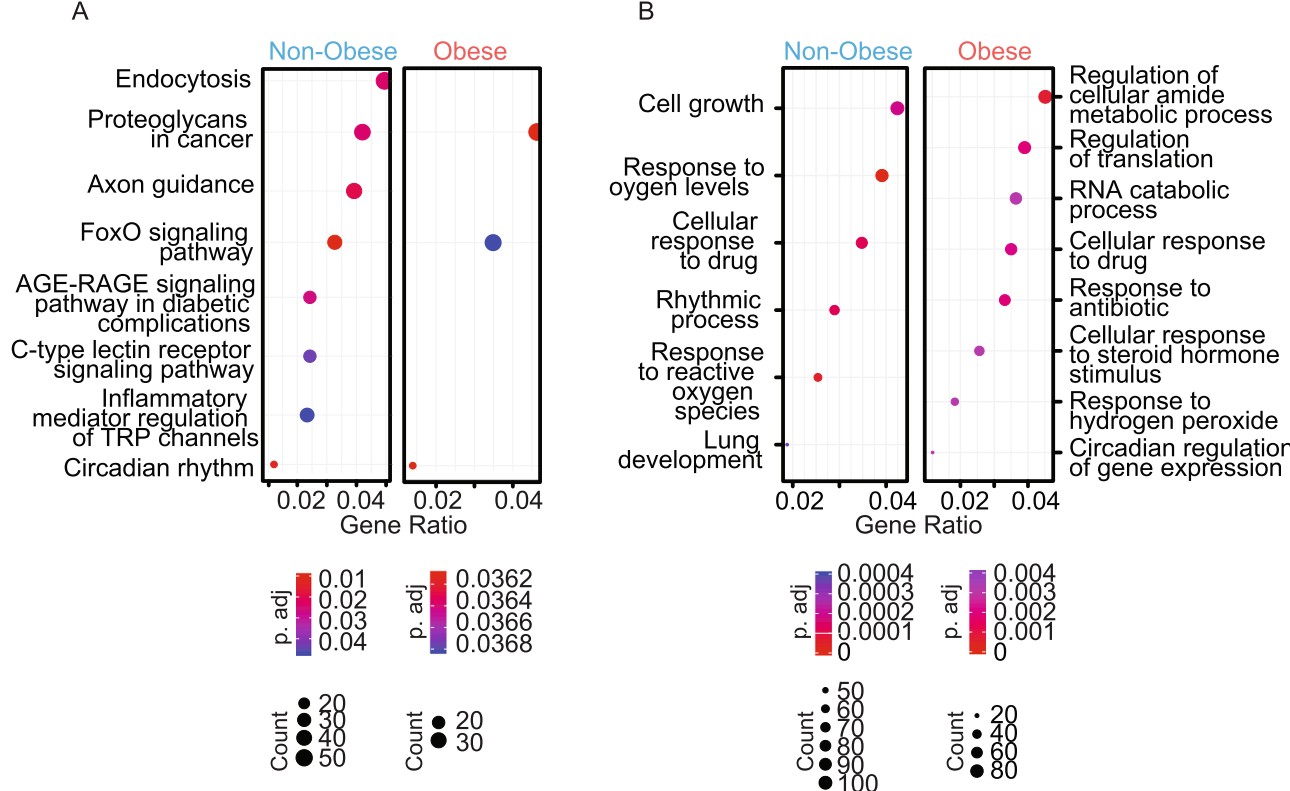

**Fig. 8 Targets bound by endogenous BMAL1 in human omental adipocyte precursors: functional pathways and biological processes revealed by ChIP-seq analysis. A** Functional pathway analysis of the sites bound by BMAL1 in OAPs from obese and non-obese patients. Genes that were within 10,000 bases of the center of each peak were selected for downstream analysis. Functional gene pathway analysis was performed using the KEGG pathway database. The dot plot shows significantly enriched pathways in each group and statistics for each pathway. "Count" indicates the number of genes enriched in a specific pathway and "Gene ratio" the percentage of genes differentially bound in the given pathway. **B** Gene ontology (GO) analysis ("biological process" sub-ontology) revealed that bound targets of BMAL1 in either non-obese or obese OAPs are enriched for genes involved in canonical circadian functions, but are also enriched for many other genes involved in different functions. **A**, **B** p-value was determined using a hypergeometric statistical test and adjusted with the Benjamini–Hochberg procedure. See also Supplementary Fig. 4.

circadian clock[47]. We showed that human OAPs display self-sustained circadian oscillations of *PER2* with a lengthened period in obesity. Another clock gene repressor, *REV-ERBα* was also decreased in both OAPs and adipocytes of our obese cohort, a finding which may also be explained by the reduced BMAL1 binding. In line with our data, *PER2* and *REV-ERBα* expression were increased in subcutaneous fat after weight loss imposed to overweight subjects[48]. REV-ERBα, together with BMAL1 and the factors encoded by some of its targets, MLL1 or NCOR2, may also shape the chromatin landscape and contribute to regulate circadian rhythms of metabolism[49–51].

Besides epigenome modifying factors, BMAL1 chromatin binding may also alter the transcription of other targets like *CCL2*. By using ChIP-Seq, we actually identified several thousand additional BMAL1 binding sites in human OAPs. Thus, functional gene pathway analysis revealed a significant enrichment for factors involved in endocytosis, proteoglycans, or FOXO signaling, which are known to regulate adipose tissue function[18,40]. FOXO pathways can also modulate a broad range of cellular processes including inflammation and metabolism[52,53]. On the other hand, BMAL1-bound genes code for proteins whose biological function ranges from cell growth and response to hypoxia, to overall regulation of RNA degradation and translation, suggesting a major role of BMAL1 in omental adipose tissue physiology. Some BMAL1 targets in our study were similar to those exhibiting a circadian expression in human subcutaneous fat, as reported in a recent transcriptomic

analysis (e.g., cellular nitrogen compound metabolic process[46]). Other BMAL1 targets identified herein also exhibited rhythmic patterns in other tissues in mice (e.g., involved in RNA biogenesis and translation[54], in collagen homeostasis[55]). Eventually, we found regions with differential BMAL1 binding between non-obese and obese OAPs, where the targets code for a variety of proteins involved in functions such as metabolic processes, collagen homeostasis and extracellular matrix remodeling, chromatin organization, vesicular trafficking and cilium function as well as oxidative stress. Measurement of mRNA and protein levels of these targets suggest that BMAL1 acts mainly like a transcriptional activator in omental fat. However, BMAL1 may act as a repressor when bound to specific loci including *CCL2* or *PAI1*, as described in other tissues[30,56–58]. It should be noted that PAI1 is a prothrombotic factor, which may be released in circulation and may therefore link central obesity to cardiovascular disease[57]. Importantly, a number of BMAL1 peaks were specific to non-obese OAPs, while others were unique to obese ones, further underlining that obesity causes a genome-wide relocalization of BMAL1 binding. The observation that BMAL1 binds in close proximity to NF-κB consensus motifs in human OAPs also suggests that IKKβ/NF-κB pathway plays a role in this repositioning. Taken together, these findings indicate that clock dysfunction might be directly involved in several pathological processes of obesity such as adipose tissue remodeling/fibrosis, altered metabolism and enhanced inflammation through direct transcriptional reprogramming.

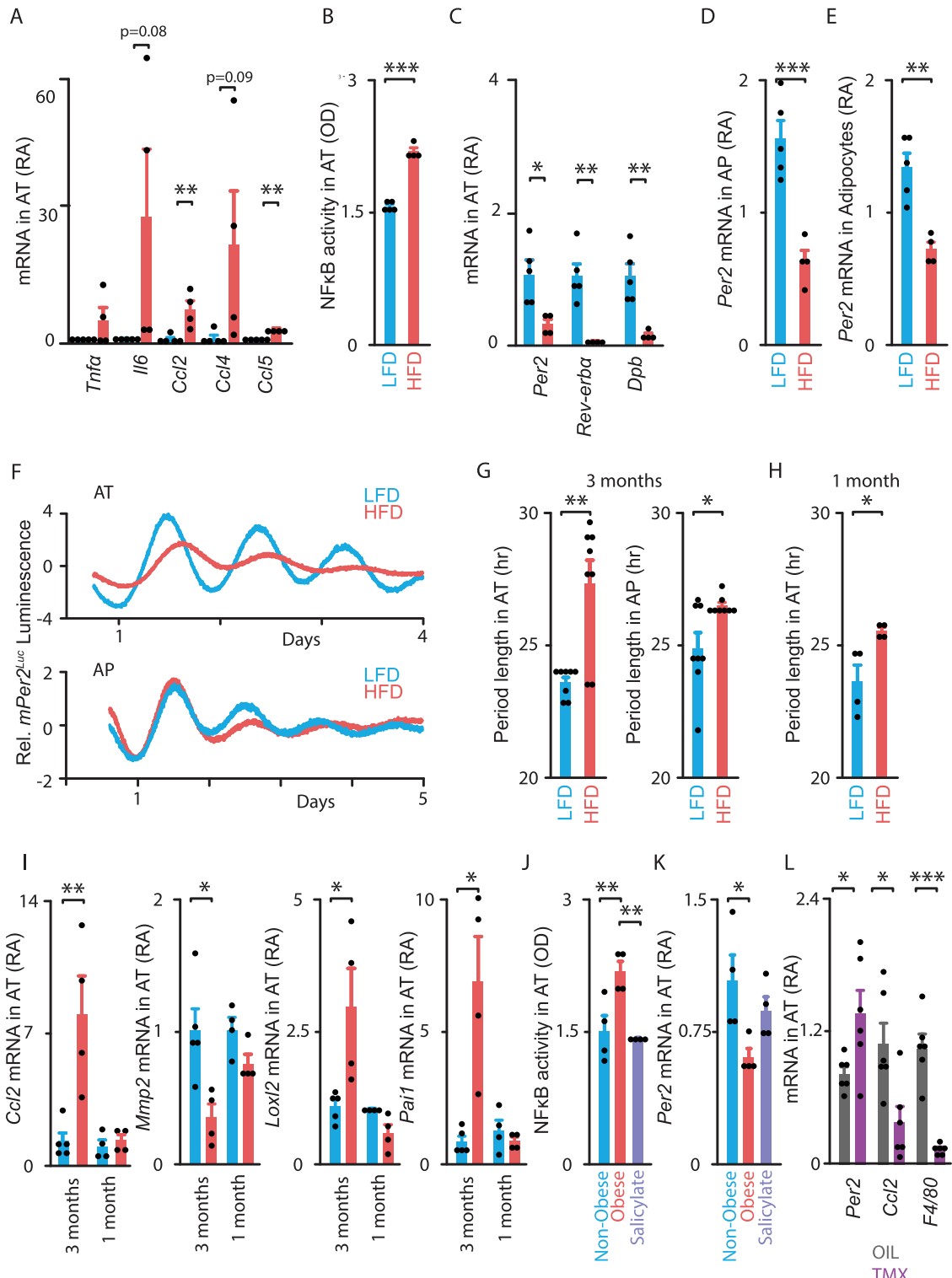

This concept was extended in mice. First, these observations were confirmed not only in adipocyte precursors, but also in mature adipocytes. Second, our results in HFD-fed mice provide evidence that circadian dysfunction actually precedes the development of metabolic inflammation (upregulation of *Ccl2, Loxl2, Pai1* and downregulation of *Mmp2*), thereby suggesting that changes in BMAL1 binding as well as other changes in chromatin state might occur very early during the development of obesity. Third, an *a contrario* approach demonstrated the importance of IKKβ/NF-κB pathway. Thus, obese mice with adipocyte-specific

deletion of *Ikkβ* showed an improvement of adipose clock properties together with an improvement of metabolic inflammation.

In conclusion, overactivation of NF-κB induces clock desynchrony in obese adipose tissue. Obesity relocalizes BMAL1 occupancy genome-wide in omental adipocyte precursors, thereby altering the transcription of numerous target genes and contributing to metabolic inflammation. Of note, our patients were severely obese and exhibited features of the Metabolic Syndrome. We cannot therefore infer that the observed changes

**Fig. 9 In vivo NF-κB inhibition prevents or rescues high-fat diet-induced adipose clock dysfunction in mice. A–K** Mice expressing the *Per2::Luciferase* (*mPer2^Luc^*) transcriptional reporter were fed a low-fat diet (LFD, blue) or a high-fat diet (HFD, red) for either 3 months (**A–G, I**) or 1 month (**H–K**) while receiving salicylate (light purple) (**J–K**). Tissues were collected at ZT 8. **A** Expression of pro-inflammatory genes, **B** NF-κB activity and **C** core-clock genes in whole (undigested) epididymal adipose explants (Adipose tissue, AT). **D, E** *Per2* gene expression in epididymal adipocyte precursors (APs) and mature adipocytes. **F–H** AT or APs were cultured to measure ex vivo PER2::LUCIFERASE fusion reporter protein from the endogenous *Per2* locus for 4-5 days. **F** Normalized bioluminescence of reporter oscillations in AT (*top*) or APs (*bottom*) ex vivo. One representative trace per condition is shown (from $n = 8$ traces/group). **G** Period length of bioluminescence in AT (left) or APs (right), in hour. (**H**) Period length of bioluminescence in AT (in hour). **I** *Ccl2, Mmp2, Loxl2, Pai1* gene expression as well as **J** NF-κB activity and **K** *Per2* gene expression in AT. All histograms above are the mean ± SEM with $n = 5$ LDF and $n = 4$ HFD-fed mice for **A–E**, $n = 8$ independent cultures/ group from 4 mice in each group for **G**, $n = 4$ mice/ group for **H, J, K**, and $n = 5/4/4/4$ mice for **I**. **L** Adipocyte *Ikkβ*-KO mice were fed a HFD for 3 months before tamoxifen treatment (dark purple) vs. oil (Ctrl, gray). Tissues were collected at ZT 8. Expression of *Per2, Ccl2* and *F4/80* genes in AT. Data are represented as mean ± SEM, with $n = 6$ mice/ group. Gene expression is displayed as mRNA Relative Abundance (RA) and NF-κB activity (phosphorylated p65) as Optical density (OD). $*p < 0.05$, $**p < 0.01$, $***p < 0.001$, by unpaired two-tailed *t*-test (**A–I, L**) or by one-way ANOVA followed by post hoc Tukey' s test (**J, K**). See also Supplementary Figs. 5–7.

in clock network were exclusively due to obesity per se rather than to insulin resistance or mild hyperglycemia for instance. Yet, it will be interesting to test whether the positive effects of the NF-κB inhibitor, salsalate, on metabolic fitness[41,42] are partially mediated by changes in adipose clock properties in obese patients in order to further confirm that preventing clock dysfunction may improve adipose tissue function. Another pharmacological strategy would be to enhance circadian rhythms[59] to reduce the development of obesity and associated complications, as shown in mice. Additional studies are therefore required to determine whether modulating adipose clock could also be a powerful strategy to reduce inflammation and improve metabolic fitness in obese patients.

## Methods

**Subjects.** The human study protocol was approved by the local Ethics Committee of Saint-Luc University Hospital (reference CEHF2017-240, with Belgian registration number B403), and patients provided informed consent. The study was performed in accordance with the regulatory guidelines. Omental adipose tissue was obtained from eight obese (class II and III; body mass index BMI ≥35 kg/m²) and eight age-matched non-obese patients (BMI <30 kg/m², with either normal weight or slightly overweight). Clinical and laboratory characteristics of these patients are summarized in Table 1. Patients underwent abdominal surgery after an overnight fast. Biopsies were obtained between 10 a.m. and 12, except for the biopsy from one lean patient obtained at 1 p.m. For non-obese patients, the surgeries included cholecystectomy and treatment of colonic diverticulosis, inguinal hernia or eventration. Obese patients underwent treatments for colonic diverticulosis, eventration or bariatric surgery. Four patients per group were treated for hypertension (two per group with amlodipine, a calcium channel blocker; one per group with an angiotensin-converting-enzyme inhibitor; and one per group with bisoprolol, a selective $\beta_1$ blocker) and one per group for hypercholesterolemia (statins). Patients with malignancies, receiving hormones (e.g., insulin) or treatments targeting adipose tissue metabolism (e.g., thiazolidinediones) were excluded.

Two additional cohorts used for comparison of omental and subcutaneous physiology (Supplementary Fig. 1D, E) and for in vitro cultures of omental adipocytes (Fig. 3H) are described in Supplementary Table 2. Biopsies were obtained between 10 a.m. and 4 p.m. In Cohort 2, paired biopsies of omental and subcutaneous adipose tissue were obtained from 12 obese patients (class III; BMI ≥40 kg/m²) undergoing bariatric surgery after an overnight fast. Two patients were treated for hypertension (amlodipine or angiotensin-converting-enzyme inhibitor) and two for type 2 diabetes (metformin or repaglinide). One patient was treated with a lipase inhibitor. In Cohort 3, omental adipose tissue was obtained from three obese patients (BMI ≥35 kg/m²) undergoing bariatric surgery after an overnight fast. All three patients had high blood pressure but only two were treated (angiotensin-converting-enzyme inhibitor or angiotensin II receptor antagonist). One patient was treated for type 2 diabetes (metformin) and one for hypothyroidism (L-thyroxine).

For each patient, plasma was obtained by refrigerated centrifugation (4 °C) of blood before surgery and stored at −20 °C. Measurements of glucose and lipids were performed by routine procedures as described[60]. Importantly, adipose tissue was quickly obtained during the surgery and either directly snap frozen and stored at −80 °C or processed for experimental purpose to avoid hypoxic conditions that could dramatically change clock gene expression[30].

### Culture and characterization of the omental adipocyte precursors. OAPs were obtained and cultured as previously described[60]. Briefly, adipose tissue was finely minced in Krebs buffer, pH 7.4, containing 2%BSA [Krebs-albumin buffer

(KRAB)]. The blood vessels were carefully dissected under binocular microscope and discarded. Minced tissue was incubated for ~30 min in a shaking water bath at 37 °C in KRAB with collagenase (type 1, ThermoFisher Scientific, # 17100-017) and filtered through a nylon mesh (200μm). Digestion was stopped by adding DMEM medium (ThermoFisher Scientific, #31885023) supplemented with 10% FBS (ThermoFisher Scientific, #10500064). The pelleted cells were separated from the adipocyte layer by centrifugation, washed in DMEM with 10%FBS, and this step was repeated three times. After treatment with an erythrocyte lysis buffer, the cells were further processed through a 70-μm cell strainer, washed once in DMEM, seeded at $3 \times 10^4$ cells/cm² and cultured as described[60]. At ~90% confluency, the cells were trypsinized, aliquoted, and the seed working stocks were cryopreserved for subsequent cultures required for the experimental procedures described below. In some experiments, we performed independent cultures (i.e., run at different times and for each time, from a new vial of cryopreserved OAPs) from patients chosen at random. We generated at most two independent cultures from a given biological donor. The donors were always chosen at random to avoid any bias of selection. The final number of independent cultures is always indicated in the legends. For the experiments with non-obese/ obese comparisons, the same number of cultures from non-obese and obese OAPs was always processed simultaneously.

We defined the OAPs as cells positive for Platelet-Derived Growth Factor Receptor α (PDGFRα⁺ or Cluster of Differentiation 140a⁺, CD140a⁺) and CD34⁺ but negative for markers of leukocytes or endothelial cells (CD45⁻ or CD31⁻) using FACSCantoII flow cytometer, with FACS Diva v8 and FlowJo v10 Software (BD Biosciences). Briefly, flow cytometry analysis using Alexa Fluor® 647 mouse anti-human CD140a (BD Biosciences, #562798) allowed to determine that 97.9 ± 0.4% cells were PDGFRα⁺ without significant difference between non-obese and obese cells ($n = 8$ patients/ group). Intensity of CD9 expression in PDGFRα⁺ cells was detected by simultaneous staining with Alexa Fluor® 647 mouse anti-human CD140a and CD9-PE antibodies (BD Biosciences, #555372). Simultaneous detection of CD34 using APC/Cyanine7 mouse anti-human CD34 (BioLegend, #343613) showed that 90 ± 1% cells were CD34⁺ without significant difference between non-obese and obese cells ($n = 8$ patients/ group). The absence of CD31⁺ and CD45⁺ cells was attested using mouse anti-human CD45-FITC (Immunotools, #21810453 × 2) and mouse anti-human CD31-FITC antibodies (Miltenyi Biotec, #130-117-539) separately in PDGFRα⁺cells ($n = 8$ patients/group). The staining was performed by incubating 10⁶ cells in a 100-μl volume, including 5 μl anti-PDGFRα, 20 μl anti-CD9, 5 μl anti-CD34, 5 μl anti-C45 or 2 μl anti-CD31 antibodies (gating strategy in Supplementary Fig. 8)[61].

Extracellular Medium Acidification Rate (ECAR), determined using a Seahorse XF96Analyzer and Glycolysis Stress Test following the manufacturer's instructions (Agilent, #102745-100, #103020-100, with $5 \times 10^4$ OAPs/ assay well using #101085-004 microplate), was similar between obese and non-obese OAPs in the basal state (Supplementary Fig. 9A).

Both non-obese and obese OAPs, when seeded at $1 \times 10^5$ cells/cm², grown to confluency and switched for 12 days to an adipogenic medium, displayed similar capability to fully differentiate into adipocytes in vitro[60], as demonstrated by lipid accumulation and late markers of differentiation (Supplementary Fig. 9B–D). Lipid accumulation was measured using Oil Red O Staining: Oil Red O was extracted with isopropanol and its absorbance was measured at 492 nm using an automatic plate-reader (*Multiskan Ascent V1.24*, Thermo Electron Corporation).

To collect RNA or total proteins from untreated OAPs, OAPs were seeded at $1 \times 10^5$cells/cm², cultured as described[60] and harvested 48-53 hr afterwards. In some experiments, $8 \times 10^4$ OAPs/cm² were seeded and transiently transfected with 90 ng of expression vector or 50 nM siRNA the following day. Briefly, increasing doses of *NF-κB p65* (30–90 ng, Addgene, Plasmid #21966)[62] and/or appropriate quantity of the empty backbone vector were transfected using Lipofectamine 2000 (Life Technologies) (medium was replaced 4 h after transfection), or with either ON-TARGETplus siRNA SMARTpool against human *NF-κB p65* (RELA, ThermoFisher Scientific, Dharmacon, L-003533-00-0005), human *NF-κB p50* (NFKB1, ThermoFisher Scientific, Dharmacon, L-003520-00-0005), or

ON-TARGETplus Non-targeting siRNA (Ctrl siRNA; ThermoFisher Scientific, Dharmacon, D001810-01-05) using Lipofectamine RNAiMAX reagent (Life Technologies). Cells were either harvested for RNA isolation or used for bioluminescence measurement 48 h post-transfection. In other experiments, cells were treated either with 10 μM NF-κB inhibitors JSH-23 (Sigma-Aldrich, #J4455), Bay11-7085 (Calbiochem, #196872) or vehicle (DMSO, baseline condition; Sigma-Aldrich, #D8418) only, either for 2 h (to collect proteins or RNA) or for long-term culture (for bioluminescence measurement). When required, cell synchronization was performed as described below.

### Preparation of omental mature adipocytes.
Human mature adipocytes were obtained and incubated in KRAB buffer for 30 min ($2 \times 10^5$ cells/ml KRAB), and snap frozen (Fig. 1A, B), or cultured as previously described[60], using $4.5 \times 10^4$ cells/$cm^2$ (Fig. 3H).

### Animals.
$mPer2^{Luc}$ transgenic reporter line[28] was obtained from Dr. MH Hastings (Division of Neurobiology, Cambridge, UK). B6.129-Tg(Adipoq-cre/Esr1*)1Evdr/J mice[44] were purchased from the Jackson Laboratory (stock #024671). $Ikk\beta^{fx/fx}$ mice[23] were obtained from Dr. G. Solinas (Department of Molecular and Clinical Medicine, University of Gothenburg, Sweden). All mice were group-housed under a 12:12-h light/dark cycle from birth and fed regular chow (Carfil quality, #10783815) and water *ad libitum* unless otherwise noted. Mice were maintained at an average temperature of 21 °C and average humidity of 55%. Seven-month-old $mPer2^{Luc}$ males were fed a high-fat diet (HFD, 60% kcal from fat, Research Diet, Inc., #D16492i) or a low-fat diet (LFD, 9% kcal from fat, Research Diet, Inc., # D10012Mi (AIN93Mi)) for 3 months. The experiment was repeated in 10-month-old sex-matched $mPer2^{Luc}$ mice for 1 month and the animals received simultaneously ~16 mg sodium salicylate (Sigma-Aldrich, #S2679) per day in the drinking water[43]. Groups were matched for body weight at the beginning of the feeding experiment. The administration of sodium salicylate did not affect daily food intake. The last cohorts of mice were generated by crossing the *Adiponectin-CreER* and *Ikkβ^{fx/fx}* mice mice. Two-month-old mice were fed a HFD for 3 months before tamoxifen treatment. Tamoxifen (Sigma-Aldrich, #T5648; 75 mg/kg body weight, prepared in corn oil) or corn oil (Santa Cruz Biotechnology, #sc-214761) was injected intraperitoneally once every 24 h for a total of 5 consecutive days, and tissues were dissected 17 days post injection. In parallel, the experiment was performed in littermate controls (i.e., *Ikkβ^{fx/fx}* and *Adiponectin-CreER* mice treated with tamoxifen vs. oil), in which we did not observe any effects of tamoxifen on *Per2* expression in AT. Blood was collected from the tail vein at ZT 8, and plasma was stored at −20 °C. Mouse insulin was measured by ELISA (Mercodia, #10-1247-01), according to the manufacturer's instructions. Mice were killed at ZT 8 and fat pads were immediately removed, weighed, snap frozen, and stored at −80 °C. The protocol was approved by the Ethics Committee for Animal Experimentation from the Medical Sector of UCLouvain (agreement number LA1230396). All experimental procedures were performed in accordance with the guidelines of the local Ethics Committee for Animal Experimentation. The regulatory guidelines followed the FELASA guidelines concerning laboratory animal welfare.

### Bioluminescence measurement and data analysis
*Human adipocyte precursors.* OAPs harbored a rapidly degradable form of luciferase, dLuc, driven by the *Per2* gene promoter through lentiviral infection (*Per2-dLuc*)[63] purchased from DNA/RNA Delivery Core, Northwestern University, using polybrene Infection Reagent (Millipore, #TR1003G). OAPs were maintained in a culture medium previously described[60], supplemented with 2 μg/ml blasticidin (Sigma-Aldrich, #15205) to select for stable *Per2-dLuc* integration. Selected cells were then trypsinized and seeded in 3.5 cm dishes ($1 \times 10^5$ cells/$cm^2$ for immediate luciferase assay the following day or $8 \times 10^4$ OAPs/$cm^2$ for siRNA or plasmid transfections prior to luciferase assay). The OAPs have been transfected before synchronization or received a pharmacological treatment after synchronization (described above). Cells were synchronized with 1 μM dexamethasone (Sigma-Aldrich, #D-1756) for 1 h. Cells were washed twice with PBS and the medium was replaced with 1.2 ml phenol-red free medium supplemented with 352.5 μg/ml sodium bicarbonate, 10 mM HEPES, 2 mM L-Glutamine, 5% FBS, 25 units/ml penicillin, 25 μg/ml streptomycin (all products from Life Technologies, Thermo-Fisher Scientific), and 0.1 mM luciferin potassium salt (Biosynth AG, #L-8240). Sealed cultures (VWR, cover glass #631-0177) were placed in a luminometer (Atto, AB-2550 Kronos Dio) at 37 °C.

*Mouse tissues or cells.* For $mPer2^{Luc}$ reporter assays, epididymal adipose tissue was dissected, and an explant (organotypic slice, tissue section of ~1-mm thickness) was cultured on Millicell culture membranes (PICMORG50, Millipore) and placed in 3.5 cm tissue culture dishes containing 1.2 ml medium described above, at 37 °C after a 5-min heat pulse (39 °C, for synchronization). In some experiments, $mPer2^{Luc}$ epididymal adipocytes ($4.5 \times 10^4$ cells/$cm^2$) were obtained following an established protocol[60], then cultured in fibrin gels[64] and maintained at 37 °C after a 5-min heat pulse (39 °C). Finally, $mPer2^{Luc}$ epididymal AP were synchronized and monitored following the protocol described above for *Per2-dLuc* OAPs.

### Bioluminescence data analysis.
Bioluminescence was recorded continuously for 4–7 days; data were analyzed with the Kronos Dio software. Data were acquired and graphed using a custom made script in RStudio. To detect the *Per2-dLuc* luminescence period from the time-series datasets, the 'meta2d' function from R' "MetaCycle" package (https://CRAN.R-project.org/package=MetaCycle)[65] was used. The analyses were performed on 4 cycles or more, except for Figs. 3D, E and 9F top, where 3.6 cycles were used.

### NF-κB activity assays.
OAPs were homogenized in a lysis buffer (Cell Signaling Technology, # 9803) supplemented with protease inhibitor cocktails (Cell Signaling Technology, #5872; Sigma-Aldrich, 11836170001). NF-κB Activity was measured with PathScan®Phospho-NF-κBp65 (pS536) ELISA Kit (Cell Signaling Technology, #7173 C) according to the manufacturer's instructions using 20 μg proteins per well. The results are expressed in optical density (OD).

Total and Phospho p65 levels were also determined by western blot as an additional control (Supplementary Fig. 9E, F) with anti-NFκB p65 antibodies (C terminal, Active Motif, 39369, diluted at 1:5000; phosphoS536, Abcam, #ab86299, at 1:2000) and Anti-rabbit IgG, HRP-linked Antibody (Cell Signaling Technology, #7074, at 1:2000). Uncropped scans are provided in the Source Data file.

### Measurement of protein concentrations.
CCL2 ELISA Kit (RayBio® Human MCP1 ELISA kit, #ELH-MCP1) was performed according to the manufacturer's instructions. Cells were seeded at $8 \times 10^4$ OAPs/$cm^2$, cultured and synchronized. The usual culture medium was removed and, following two washes with PBS, replaced for 6 hr with serum-free medium used as conditioned medium. The concentration of other proteins encoded by BMAL1 target genes was measured in OAP homogenates using ELISA kits purchased from MyBioSource [FLOT1 (#MBS9715917), NCOR2 (#MBS9715002), MMP2 (#MBS2701186), MLL1 (#MBS754904), GPX1 (#MBS919262), PAI1 (#MBS267611), and LOXL2 (#2023260)] according to the instructions with 10 μg total proteins per well.

### RNA isolation and quantitative real-time PCR.
RNA from cells and tissues were extracted and reverse transcribed as previously described[60]. QRT-PCR was performed and analyzed with an iCycler iQ real-time PCR detection system (Bio-Rad Laboratories), using the primers described in Supplementary Table 3 (refs. [2,9,60]). The threshold cycles (Ct) were measured in separate tubes and in duplicate, and relative expression levels (normalized to *GAPDH* for human adipose tissue, adipocytes and OAPs, *18s* for mouse adipose tissue and adipocytes and *Gapdh* for mouse APs) were determined using the comparative Ct method as previously described[60]. Values were displayed as mRNA Relative Abundance (RA).

### Chromatin immunoprecipitation and data analysis.
ChIP methods were adapted from previously described experimental procedures[66]. Briefly, OAPs were dual-crosslinked with Disuccinimidyl glutarate (Life Technologies, ThermoFisher Scientific, #20593) followed by 1% formaldehyde (Polysciences, #18814-20). Nuclei were isolated via needle lysis in IP buffer[66], and chromatin was sheared to 200–1000 bp fragments by sonication using a Diagenode Bioruptor using TPX microtubes (Diagenode, #C30010010) in Shearing Buffer[66]. Chromatin was then incubated overnight with either antibody against p65 (Active Motif, #39369), BMAL1 (Abcam, #Ab3350), RNA POLII CTD repeat YSPTSPS Ser-5P (Abcam, #Ab5131), diluted at 1:250, or with rabbit immunoglobulin G (Preprotech, #500-P00) in Dilution Buffer followed by 2 h with protein A Agarose beads (Santa Cruz Biotechnology, #sc-2001). Beads were washed six times followed by de-crosslinking using Chelex beads (Sigma-Aldrich, #C7901) and proteinase K (Sigma-Aldrich, #3115828001) digestion. Eluted immunoprecipitated DNA and input DNA were purified using MinElute PCR purification columns (Qiagen, #28004) and subjected to QPCR analysis using primer sequences described in Supplementary Table 3. Potential binding sites for NF-κB were determined using Genomatix Matinspector (http://www.genomatix.de/matinspector/) and p65 ChIP-seq database[29], while primers for BMAL1 sites were either specific to well-known E-box target sites or recently identified in BMAL1 ChIP-seq database[30] in human cell lines. Specific BMAL1 and NF-κB p65 binding to the studied genes were validated with negative controls through the interrogation of DNA regions distant from suspected binding sites. QPCR signals from the input chromatin controls were never different between the different conditions (non-obese and obese patients, treatment conditions, or time after synchronization). Results are presented in fold enrichment over IgG, after validation of consistent results when expressed in percentage input chromatin.

### ChIP-sequencing and data analysis.
ChIP reactions, library preparation, and bioinformatics analyses were performed by Active Motif. Briefly, 30 μg of sheared chromatin (300–500 bp) was used per ChIP reaction. Protein-DNA complexes were incubated with 8 μl antibody against BMAL1 (Abcam, #Ab3350). Illumina sequencing libraries were prepared from ChIP and pulled input chromatin DNAs. Sequencing was generated using 75-bp single-end reads on an Illumina NextSeq 500 instrument to a depth of >30 million reads. Reads were aligned to the human genome (hg38) using Burrows-Wheeler alignment algorithm with default settings (http://bio-bwa.sourceforge.net/). Only reads that passed Illumina's purity filter, aligned with no more than two mismatches, and mapped uniquely to the genome were used in the subsequent analysis. In addition, duplicate reads were removed. For comparative analysis,

standard normalization was achieved by down-sampling the usable number of tags (5′-ends of the aligned reads) for each sample in a group (non-obese vs. obese OAPs) to the level of the sample in the group with the fewest usable number of tags. This resulted in an equal number of tags (20,191,577 tags) within each group. The tags were extended in silico using Active Motif software at their 3′- ends to a length of 200 bp (average fragment length in the size selected library bp). To identify the density of fragments (extended tags) along the genome, the genome was divided into 32-nt bins and the number of fragments in each bin was determined. This information was stored in bigWig files for display. Peak calling was performed using the MACS 2.1.0 algorithm[67] with a default cutoff of $p$-value $10^{-7}$ for narrow peaks and $10^{-1}$ for broad peaks. Peak filtering was performed by removing false ChIP-Seq peaks as defined within the ENCODE blacklist[68]. This process allowed to determine the significant enrichments in the ChIP/IP data file when compared to the Input data file (~background). Then, known motifs were identified with the findMotifsGenome program of the HOMER package (http://homer.ucsd.edu/homer/index.html)[69] using default parameters and input sequences comprising ±100 bp from the center of the top 1000 peaks. Significant motif enrichment was determined using a cumulative binomial statistical test within HOMER. Identification of peaks with statistical enrichment over input across conditions was performed using DESeq2 analysis[70]. The threshold for the number of tags that determines a valid peak was selected for a FDR-adjusted $p$-value of <0.1. This analysis enabled us to detect a differential BMAL1 binding genome-wide between the three non-obese and three obese OAPs (Supplementary Fig. 10; Raw data in three non-obese and three obese OAPs are provided in Source Data). VENN diagram, KEGG pathway, and GO enrichment analyses were performed on peaks that were present in 2–3 patients/group. The analysis of KEGG pathway database was performed using clusterProfiler https://www.genome.jp/kegg/(https://bioconductor.org/packages/release/bioc/html/clusterProfiler.html). The GO analysis was also performed with clusterProfiler, using the "Biological Process" sub-ontology, with Bioconductor's human annotation database (http://bioconductor.org/packages/release/data/annotation/html/org.Hs.eg.db.html). For both KEGG and GO analyses, statistical significance was determined using a hypergeometric statistical test and adjusted with the Benjamini–Hochberg procedure.

**Statistical analysis**. Results are represented as mean ± SEM for the indicated numbers of patients or independent cultures. Normality was calculated by Kolmogorov–Smirnov test. Comparisons between two different groups (non-obese/obese) were carried out using two-tailed unpaired Student's $t$ test (with Welch correction where appropriate), unless indicated otherwise. Comparisons between different conditions within a same group were made using two-tailed paired Student's $t$ test or repeated ANOVA followed by post hoc Dunnett's test for multiple comparisons. The effect of BMI and time was tested by two-way ANOVA followed by post hoc Sidak's correction for multiple comparisons. Comparisons between three groups of mice (LFD/HFD/HFD salicylate) were carried out by one-way ANOVA followed by post hoc Tukey's test for multiple comparisons (Prism 8; GraphPad Software).

**Reporting summary**. Further information on research design is available in the Nature Research Reporting Summary linked to this article.

## Data availability

ChIP-seq data generated and analyzed during this study are available in a GEO repository with the identifier GSE149064. (publicly available). We used the publicly available datasets GSE19486 and GSE85096: https://doi.org/10.1126/science.1183621. https://doi.org/10.1016/j.cmet.2016.09.009. The other datasets generated and analyzed during the study are provided in Source Data. Source data are provided with this paper.

## Code availability

The analysis of bioluminescence data was performed with a custom script deposited in Github: https://github.com/EmauryUCL/Extraction-Analysis-and-Graph-of-Lumicycle-data/releases/tag/V1.0. The analysis of KEGG pathway database was performed using clusterProfiler https://www.genome.jp/kegg/.(https://bioconductor.org/packages/release/bioc/html/clusterProfiler.html). The gene ontology analysis was also performed with clusterProfiler, using the "Biological Process" sub-ontology, using Bioconductor's human annotation database (http://bioconductor.org/packages/release/data/annotation/html/org.Hs.eg.db.html). The code for the Burrows-Wheeler alignment algorithm can be downloaded from: http://bio-bwa.sourceforge.net/

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

## Acknowledgements

E.M. received research support from INNOVIRIS ATTRACT *"Brains for Brussels"* (reference 2016-BB2B-4) and Société Francophone du Diabète/AstraZeneca to perform this work in S.M.B's laboratory. We thank Pr. J.S. Takahashi (Howard Hughes Medical Institute, Department of Neuroscience, University of Texas Southwestern Medical Center, Dallas, USA) and Pr. M.H. Hastings (Division of Neurobiology, Cambridge, UK) for *mPer2^Luc* transgenic reporter mice. We thank Pr. M. Karin (Department of Pharmacology, University of California, San Diego, USA) and Pr. G. Solinas (The Wallenberg Laboratory for Cardiovascular and Metabolic Research, Department of Molecular and Clinical Medicine, University of Gothenburg, Sweden) for IKKβ^fx/fx mice. We thank Elise Buchin (IREC, EDIN Department, UCLouvain) for the excellent technical support with colony maintenance, genotyping, cell culture, ChIP, and gene expression experiments. We also received the technical assistance from Laurence Noel (mouse husbandry, genotyping, ELISAs, and reading of a former version of the manuscript), Camille Selvais (participated in colony maintenance, gene expression, reading of a former version and helped for the cartoon in Fig. 4H) and Dr. M. Abou-Samra (western blot) from the IREC, EDIN Department, UCLouvain. We thank Pr. W-H. Lien (De Duve Institute, UCLouvain, Brussels, Belgium) for her technical assistance with the Diagenode Bioruptor device; Dr. D. Brusa (Institute of Experimental and Clinical Research IREC, Flow Cytometry Core, UCLouvain, Brussels, Belgium) for his technical assistance with the FACS device; Pr. C. Corbet (IREC, FATH Department, UCLouvain) for his technical assistance with the Seahorse Bioenergetic Analyzer. We thank Pr. B. Beck (IRIBHM Institute, ULB, Brussels, Belgium) for his comments on a former version of the manuscript. Finally, we thank the Digestive Surgery team and especially the nurses for their help in collecting adipose tissue biopsies.

## Author contributions

E.M. acquired funding and designed, performed and analyzed the experiments, and wrote the manuscript. N.B. and the staff from the Digestive Surgery Unit sampled the biopsies. S.M.B. gave suggestions and revised the manuscript.

## Competing interests

The authors declare no competing interests.
