## [Peer Review File · Nature Communications]

Reviewers' comments:

Reviewer #1 (Remarks to the Author):

In the present manuscript, the authors dissect the molecular link between NF- κ B and the expression of PER2 in cells from human omental adipose tissue. The study is interesting since it highlights inter-relation between PER2- and inflammatory-related pathways in human omental fat depots.

However, several conclusions are not supported by the data and will require additional experiments:

-“the study reveals that the inflammatory state in omental adipose tissue obesity is associated with marked inhibition of clock repressors induced by IKK β /NF- κ B signaling pathway”. The present data show impact on PER2 only. Studies on REV-ERBa and ROR α exhibiting differences between non-obese and obese cells will permit to extend the primary observation to other clock components and will strengthen the link between circadian and inflammatory-related pathways.

- “Our findings indicate that the dysregulation of circadian systems resulting from improper activation of NF- κ B in omental adipocytes and preadipocytes contributes, in turn, to chemokine overproduction”. The present data show impact on CCL2 only. To note, in OA adipocytes Figure 1A, the upregulation of CCL2 expression in obese is not as high as TNF α or CCL4. κ B binding sites have been reported in CCL2 regulatory elements and NF- κ B independent pathways such as FOXP1 have been also involved in the modulation of CCL2 expression. It will have been interesting to take into account such regulatory elements and in addition to study other relevant chemokines and cytokines up-regulated in obese conditions.

-“Consequently, these cells prompt the infiltration of macrophages and other immune cells exacerbating a pro-inflammatory environment” and “This observation is relevant as macrophages preferentially proliferate in visceral compared to subcutaneous adipose tissue in response to CCL2 secretion by adipocytes”. The present study do not provide data concerning the pathophysiological consequences of an increased CCL2 expression (secretion?) in omental fat depot and additional approaches are needed to determine the impact of CCL2 on macrophage infiltration and/or proliferation and consequences on fat depot function.

-“Alternatively, preventing clock dysfunction in obese adipose tissue might reduce inflammation and improve metabolic fitness”. Although particularly interesting, no data are provided to support such hypothesis.

The main concerns are the patient cohort and the cell model. The clinical parameters associated with the cohort of obese patients exhibit features of metabolic syndrome as mentioned by the authors. Therefore, the present study concerns the impact of obesity-associated metabolic syndrome and not of obesity only. Since clock gene expression is closely linked with metabolism and obese patients from the studied cohorts exhibit high fasting blood glucose, it is unclear whether the changes in gene expression depicted in figure 1 are linked to obesity per se or to hyperglycemia. Moreover, when considering alteration of circadian-related pathways in obesity, the relevance to compare circadian-related gene expression levels between non-obese and obese patients at undefined time is unclear.

The OPA cell model requires additional characterization especially the method to obtain OPA. It is mentioned that cells are all PDGFR α positive, CD45 negative and CD31 negative but without specific selection, it is unclear how the authors succeed to obtain a “pure” cell population without macrophages and endothelial cell contamination. Moreover cells are named “preadipocytes” but no further characterizations are provided. The adipogenic differentiation protocol does not appear to be performed at confluency from S2B pictures. A broader view of cell differentiation at day 12 must be shown (only three- four cells are shown) as well as pictures allowing direct comparison between non-obese and obese cells. mRNA expression must be shown at day 0 and day 12 for relevant adipogenic markers (and not only at day 12) to clearly show the extent of adipogenic induction in both obese and non-obese cells. The relevance of studying CD9 expression on cells is not well justified. It is unclear whether experiments were performed on primary or passaged cells. Normalized bioluminescence of Per2-dLuc reporter oscillations in synchronized OPA is performed in cells selected for stable Per2-dLuc integration. It is unclear how the differences in endogenous cell-autonomous oscillations between non-obese and obese OPA are maintained in passaged cells if originating from enhanced NF- κ B activation. Changes in chromatin states must play a major role. The animal model is studied at 3 months under high fat diet, time point where both inflammation

and circadian pathways are already altered. It will have been interesting to perform kinetics in order to determine whether inflammatory-related events may precede (or not) alteration of circadian systems but also to compare both subcutaneous and visceral fat depots in order to define the potential fat depot specificity.

Reviewer #2 (Remarks to the Author):

The manuscript by Maury et al investigated the interplay between NF-kB-mediated inflammation and clock function in human obesity. It demonstrates that several core clock repressors are down-regulated in omental adipose tissue from obese patients and in isolated adipocytes and preadipocytes. They also suggest that this down regulation results from NF-kB-mediated repression of CLOCK/BMAL1 –dependent transcriptional activation of Per2. Altogether this work suggests that NF-kB may alter circadian function in adipose tissue of obese patients. There are several lines of criticism that will not recommend this manuscript for publication at least in its present form.

First, it lacks originality as the major phenomenon (NF-kB-mediated effect on circadian function) has been described in detail in Hong et al (Genes and Dev., 2018) in various models of acute (TLR-induced) and chronic (high fat diet) NF-kB activation. Essentially, current manuscript provides similar data using the model of human obesity, which is fine if clearly states it.

Second, there are a lot of statements that are not supported by original experimental data. It refers to all ChIP data that show only the results of quantitation and all data on functional status of p65. These require original Western blots of both total and phosphor-p65 side by side as the readers as well as reviewers cannot rely just on bars. Each Figure legend should provide info for the number of samples used and statistical test used to assess significance (if it seems to be redundant).

And finally, the manuscript is not clearly written and is very hard to go through. This includes both ambiguous statements and mistakes. Below are several examples of this:

There are numerous examples when the order of Figure panels does not correspond to the order of the reference in the text making it very uneasy to follow

There is no reference in the text to Fig. 1A

Lane 104 –according to Fig. S1C Clock expression is decreased in contrast to the statement The phrase “interplay between clock disruption and NF-kB –mediated inflammation” (Abstract) is misleading as it gives the impression that clock disruption is the primary event in this cross-talk Moreover, there is no clock disruption observed here, the clock is still functional with reduced amplitude and period lengthening

Are the red bars on Fig.S1B and C the same as red bars in S1D and E? May these need to be combined (S1b and D and S1C and E) to avoid confusion?

According to established nomenclature names in all caps (PER, CLOCK, BMAL1) refer to proteins, not to genes (Per, Clock, Bmal1 italicized)

Based on this essential criticism it is suggested that the manuscript needs major revision in order to be considered for publication.

Reviewer #3 (Remarks to the Author):

This article by Maury and colleagues describes the regulation of the circadian clock by the inflammatory-related transcription factor NF-kB in adipose tissue. Using human omental adipose tissues, as well as omental adipocytes and preadipocytes from different cohorts, authors show that obesity alters the circadian clock function in this tissue, associated with tissue inflammation. In a second time authors show more specifically that NF-kB interfere with the regulation of the circadian clock PER2 gene by BMAL1. Through competition on binding sites at the TSS. Such competition occurs also at the promoter of the CCL2 chemokine, influencing its transcription during obesity. Finally, authors confirm this regulation in a mouse model of obesity.

While interesting, most of these results have been already published in mouse models, namely the regulation of the circadian clock in obesity (Kohsaka et al., 2007), the role of NF-kB in this regulation through competition with BMAL1 (Hong et al., 2018), and the regulation of the Ccl2

transcription by BMAL1 (Nguyen et al., 2013). The new findings consist only in the confirmation of these results in human adipose tissue. In addition, several points need to be addressed before a potential publication:

- Figure 1B, 1H, S1C and S1E show the expression of circadian clock genes in different tissues and cells from human donors. However, considering the small number of donors, to what extent this difference could be only due to different sampling times in the different groups. It could be interesting to correlate clock gene expression with the sampling time. Without this information, it is almost impossible to analyze this data.
- In Figure 2, authors focus on the regulation of PER2 by BMAL1 and NF- κ B. It will be interesting to see if their findings are also true genome wide, as described in (Hong et al., 2018).
- In Figure 3 and 4, authors show that inhibition or overexpression of the NF- κ B pathway in obese cells increases PER2 expression and shorten the circadian period. What is the effect of NF- κ B inhibition, p65/p50 siRNA or p65 overexpression on PER2 expression in non-obese cells? Is the effect of the inhibitors similar or blunted?
- In Figure 5, authors show the role of BMAL1 binding on the Ccl2 expression. To what extent the binding of NF- κ B on this promoter is affected in obese and non-obese samples?

Hong, H.-K., Maury, E., Ramsey, K.M., Perelis, M., Marcheva, B., Omura, C., Kobayashi, Y., Guttridge, D.C., Barish, G.D., and Bass, J. (2018). Requirement for NF- κ B in maintenance of molecular and behavioral circadian rhythms in mice. *Genes Dev* 32, 1367-1379.

Kohsaka, A., Laposky, A.D., Ramsey, K.M., Estrada, C., Joshu, C., Kobayashi, Y., Turek, F.W., and Bass, J. (2007). High-Fat Diet Disrupts Behavioral and Molecular Circadian Rhythms in Mice. *Cell Metab* 6, 414-421.

Nguyen, K.D., Fentress, S.J., Qiu, Y., Yun, K., Cox, J.S., and Chawla, A. (2013). Circadian Gene *Bmal1* Regulates Diurnal Oscillations of Ly6Chi Inflammatory Monocytes. *Science* 341, 1483-1488.

Answers to the reviewers

Changes in the main text indicated in red

Enclosed, we provide detailed point-by-point responses to the specific critiques.

Reviewer #1

We would like to thank the Reviewer for the positive remark “The study is interesting since it highlights inter-relation between PER2- and inflammatory-related pathways in human omental fat depots”.

1. *“The study reveals that the inflammatory state in omental adipose tissue obesity is associated with marked inhibition of clock repressors induced by IKKb/NK-kB signaling pathway”. The present data show impact on PER2 only. Studies on REV-ERBa and RORalpha exhibiting differences between non-obese and obese cells will permit to extend the primary observation to other clock components and will strengthen the link between circadian and inflammatory-related pathways.*

- The referee is right. *REV-ERB α* mRNA levels were decreased in both omental adipocytes and preadipocytes (OPAs) (Fig. 1B, G) as well as in whole omental adipose tissue (Fig. S1C) from obese patients. We have performed BMAL1 ChIP-sequencing in omental preadipocytes (OPAs). We show UCSC genome browser images of BMAL1 ChIP-sequencing tracks at *REV-ERB α* in OPAs from 3 non-obese and 3 obese patients. Interestingly, we found that BMAL1 peak density was reduced in the obese condition (by ~22% compared to BMAL1 binding to *REV-ERB α* in non-obese preadipocytes $P < 0.05$, Fig. 5F), suggesting that the lower expression of *REV-ERB α* in obese OPAs might result from the decreased binding of the transcriptional activator BMAL1. As suggested by the Reviewer, we further discussed the potential role of *REV-ERB α* in adipose tissue inflammation.

These data have been summarized in the new Fig.5E-F. These results have also been described in Results (p. 6-7, lines 239-244). The Discussion has been modified accordingly (p. 9, one new sentence, lines 359-361 and two sentences, lines 368-370 “Decreased REV-ERB α could also contribute (...) *Ccl2* in mice”), and the appropriate reference (52) has been inserted.

- Several studies establish a role for ROR α in the regulation of inflammatory processes. However, *ROR α* expression was not significantly different in omental adipose tissue from non-obese and obese patients (Fig. S1C). Moreover, we found no differences in *ROR α* mRNA levels in OPAs (Fig. 1G). Consistently, BMAL1 ChIP-sequencing in OPAs failed to identify BMAL1 peaks to the *ROR α* gene promoter in these cells (not shown). We only found a downregulation of *ROR α* expression in mature adipocytes from obese patients (Fig.1 B). Therefore, there was no convincing evidence to support an important impact of ROR α on omental fat inflammation, and we did not discuss this point.

2. a) *“Our findings indicate that the dysregulation of circadian systems resulting from improper activation of NF- κ B in omental adipocytes and preadipocytes contributes, in turn, to chemokine overproduction”. The present data show impact on CCL2 only. To note, in OA adipocytes Figure 1A, the upregulation of CCL2 expression in obese is not as high as TNF alpha or CCL4.*

We removed this sentence. Although we confirmed a significant BMAL1 occupancy on the CCL2 promoter in omental preadipocytes, as shown in the **new Fig.S4C**, BMAL1 ChIP-sequencing in OPAs failed to identify BMAL1 peaks to the promoters of TNF α or CCL4 (not shown), thereby justifying to focus on CCL2 only.

These results have also been described in the Results (p. 7, lines 255-257). A link to the UCSC browser has been provided in the Reporting Summary.

b) κ B binding sites have been reported in CCL2 regulatory elements and c) NF- κ B independent pathways such as FOXK1 have been also involved in the modulation of CCL2 expression. It will have been interesting to take into account such regulatory elements and in addition to study other relevant chemokines and cytokines up-regulated in obese conditions.

We have carried out p65 ChIP to measure p65 binding to the CCL2 promoter in omental preadipocytes from non-obese and obese patients. We found an increase in p65 occupancy at the κ B sites located near the TSS of CCL2 in obese OPAs (vs. non obese OPAs).

These data have been summarized in the new Fig.S3I. These results have also been described in the Results (p. 6, lines 206-207).

c) NF- κ B independent pathways such as FOXK1 have been also involved in the modulation of CCL2 expression.

BMAL1 ChIP-sequencing in omental preadipocytes now indicates an overrepresentation of FOXO signaling pathway within the BMAL1 bound targets. The potential involvement of FOXO signaling in the modulation of CCL2 expression has been further discussed.

These data can be found in the new Fig.7A. These results have also been described in Results (p. 7, lines 268-270). We have now added 2 sentences in the Discussion (p. 10, 380-384) and 1 new reference (54).

3. *“Consequently, these cells prompt the infiltration of macrophages and other immune cells exacerbating a pro-inflammatory environment” and “This observation is relevant as macrophages preferentially proliferate in visceral compared to subcutaneous adipose tissue in response to CCL2 secretion by adipocytes”. The present study does not provide data concerning the pathophysiological consequences of an increased CCL2 expression (secretion?) in omental fat depot and additional approaches are needed to determine the impact of CCL2 on macrophage infiltration and/or proliferation and consequences on fat depot function.*

We removed this part of the discussion.

Our new data obtained from mice with conditional inactivation of *Ikk β* restricted to adipocytes show a positive relationship between the expression of CCL2 and the most commonly used murine macrophage marker (F4/80) in adipose tissue. **These data have been summarized in the new Fig. 8L-M (and related Fig.S6H-K). These results have also been described (Results, p. 9, lines 329-339).** These new results in mice are in agreement with the positive relationship previously found between the expression of CCL2 and the human macrophage marker (CD68) in

omental adipose tissue from obese subjects (already shown in the first version of this manuscript, now in Fig. S1A-B). The increased expression of the macrophage marker in omental fat coincided with an elevated expression and secretion of *CCL2* by OPAs from obese patients (vs. non-obese ones) (already shown in the first version of this manuscript, now Fig. 1E and Fig. 4E-F), and with an elevated expression in mature adipocytes (already shown in the first version of this manuscript, now Fig. 1A). Collectively, our results show a consistent positive relationship between *Ccl2* expression (or secretion) levels and that of macrophage markers. In addition, there is already an extensive literature describing the impact of *CCL2* on macrophage infiltration and proliferation. The appropriate references were already cited, and are now p. 10 lines 377-378. Our study completes this literature by describing a new mechanism controlling *CCL2* expression (and secretion) in human omental adipose tissue.

We have rewritten this part of the Discussion (2 sentences, p. 10, lines 373-378).

4. *“Alternatively, preventing clock dysfunction in obese adipose tissue might reduce inflammation and improve metabolic fitness”. Although particularly interesting, no data are provided to support such hypothesis”.*

In the experiments detailed above in comment 3, there is always an inverse relationship between *Per2* expression (and/or period length) and *Ccl2* expression (and/or secretion) in both humans (already shown in the first version of this manuscript, now Fig.1I-J and Fig.4E-F) and mice (new data Fig.8K-M). Our new results in adipocyte *Ikkβ*-KO mice further support our initial observations, thereby suggesting a convergence between inflammation and clock dysfunction. Thus, in adipocyte *Ikkβ*-KO mice, the higher expression of *Per2* is associated with a decreased expression of *Ccl2* and a very significant decreased expression of a macrophage marker (**new data Fig.8K-M**, and related Fig.S6H-K).

In addition, our new results in mice provide evidence that adipose clock is dysfunctional at early stages of obesity. In particular, we observed that altered clock function precedes the upregulation of *Ccl2* mRNA levels. **These data have been summarized in the new Fig.8H. These results have also been described in Results (p. 8, lines 303-306). We have now added 1 sentence in the Discussion (p. 10, lines 370-373).**

5. *The clinical parameters associated with the cohort of obese patients exhibits features of metabolic syndrome as mentioned by the authors. Therefore, the present study concerns the impact of obesity-associated metabolic syndrome and not of obesity only. Since clock gene expression is closely linked with metabolism and obese patients from the studied cohorts exhibit high fasting blood glucose, it is unclear whether the changes in gene expression depicted in figure 1 are linked to obesity per se or to hyperglycemia.*

These limitations are now discussed (2 sentences in the Discussion, p.10, lines 405-408).

6. *Moreover, when considering alteration of circadian-related pathways in obesity, the relevance to compare circadian-related gene expression levels between non-obese and obese patients at undefined time is unclear.*

Biopsies from non-obese and obese patients have been obtained at the same time, within an interval of 2 hours (between 10am-12).

We now provide the times of biopsy collection in the Methods (one new sentence, p. 17, lines 614-615).

7. *The OPA cell model requires additional characterization especially the method to obtain OPA. It is mentioned that cells are all PDGFR α positive, CD45 negative and CD31 negative but without specific selection, it is unclear how the authors succeed to obtain a “pure” cell population without macrophages and endothelial cell contamination.*

The protocol is now carefully detailed (Methods, p.17-18, lines 640-672). We used this detailed protocol and discarded the cultures of preadipocytes contaminated by other cells rather than using Magnetic Activated Cell Separation to deplete the population for endothelial cells and leukocytes or sorting the cells by FACS as a purification step, as it provided a better yield.

8. *Moreover, cells are named “preadipocytes” but no further characterizations are provided.*

1) We measured the expression of commonly used preadipocyte markers. Our approach is based on recent reports from other teams (including the references 21 and 24 in our manuscript) and on our own previous studies (including reference 59 in our manuscript). **The protocol for characterizing these cells is now described in more detail.** 2) We also performed *in vitro* adipocyte differentiation, thereby providing an additional phenotypic characterization of the cells **(Methods, now p. 18 lines 677-682, see next comment).** Therefore, we feel that our preadipocytes are well characterized.

9. *The adipogenic differentiation protocol does not appear to be performed at confluency from S2B pictures. A broader view of cell differentiation at day 12 must be shown (only three- four cells are shown) as well as pictures allowing direct comparison between non-obese and obese cells. mRNA expression must be shown at day 0 and day 12 for relevant adipogenic markers (and not only at day 12) to clearly show the extent of adipogenic induction in both obese and non-obese cells.*

We now provide more accurate data (Methods, p. 18 lines 677-682, and edited Fig.S8B-D): a broader view of cell differentiation is now shown at day 12, mRNA expression is now shown at day 0 and day 12 for relevant adipogenic markers, in cells isolated from both obese and non-obese patients.

10. *The relevance of studying CD9 expression on cells is not well justified.*

Studying CD9 expression completes the phenotypic characterization of the isolated cells. A former elegant study reported that CD9 expression delineates two subsets of adipose PDGFR α ⁺ progenitors: CD9^{high} progenitors drove adipose tissue fibrosis (Marcelin et al, 2017, reference 21 in our manuscript). In obese subjects, increased CD9^{high} progenitors in visceral fat were associated with adipose tissue inflammation and diabetes.

We have rewritten the sentence justifying the relevance of studying CD9 expression in Results (p. 3, lines 91-93).

11. *It is unclear whether experiments were performed on primary or passaged cells. Normalized bioluminescence of Per2-dLuc reporter oscillations in synchronized OPA is performed in cells selected for stable Per2-dLuc integration.*

This part of Methods is now detailed (p. 19, lines 732-734): 2 sentences have been added to describe experiments performed on transduced cells (treated with blasticidin).

12. *It is unclear how the differences in endogenous cell-autonomous oscillations between non-obese and obese OPA are maintained in passaged cells if originating from enhanced NF- κ B activation. Changes in chromatin states must play a major role.*

We added one new sentence to mention that chromatin remodeling may occur in obesity (Discussion, p. 10, lines 370-373). The significant differences in BMAL1 occupancy genome-wide at the chromatin level in obese OPAs further supports this hypothesis.

13. *The animal model is studied at 3 months under high fat diet, time point where both inflammation and circadian pathways are already altered. It will have been interesting to perform kinetics in order to determine whether inflammatory-related events may precede (or not) alteration of circadian systems*

We thank the reviewer for this helpful comment (also related to comment 4). We now describe a significant period lengthening in adipose tissue from mice fed a HFD for a shorter time period (1 month instead of 3 months; new Fig. 8H to be compared with Fig 8G), but these changes precede the upregulation of *Ccl2* expression, which is observed after 3 months of HFD only (Fig. 8A). These new results are in agreement with the hypothesis that adipose clock dysfunction might precede other inflammatory alterations and determine some metabolic outcomes in obesity. This point has been discussed.

These data have been summarized in the new Fig.8H. These results have also been described in Results (p. 8, lines 303-306). We have now added one sentence in the Discussion (p. 10, lines 370-373).

14. *To compare both subcutaneous and visceral fat depots in order to define the potential fat depot specificity.*

We had already performed this experiment in human adipose tissue. It was relevant as visceral/central obesity is associated with metabolic complications. Thus, we had compared pro-inflammatory and clock gene expression in paired omental and subcutaneous fat depots. This result was already presented in the former version of the manuscript, and remains **described in Fig S1D-E, and in Results, p. 3, lines 100-103**. The differences between the 2 depots further justify the rationale for focusing on the omental fat depot throughout the study.

Reviewer #2

1. *First, it lacks originality as the major phenomenon (NF- κ B-mediated effect on circadian function) has been described in detail in Hong et al (Genes and Dev., 2018) in various models of acute (TLR-induced) and chronic (high fat diet) NF- κ B activation. Essentially, current manuscript provides similar data using the model of human obesity, which is fine if clearly states it.*

The main finding of our paper is to reveal an interplay between inflammation and circadian systems in human omental fat and to show its involvement in the pathogenesis of obesity and its complications. We feel that this message is original. Firstly, such interplay as well as BMAL1 cistrome have only been described in the liver of obese mice, not in adipose tissue (Hong et al., 2018; reference 2). As BMAL1 predominantly binds DNA in a tissue-specific manner (Beytebiere et al, 2019; reference 8), we cannot infer that similar data would have been obtained in mouse fat

tissue. Secondly, some data obtained in mouse may not be translated into humans. Thirdly, the few reports on clock machinery in human adipose tissue did not study the interplay with NF- κ B and mainly focused on the subcutaneous fat depot (e.g., references 47-48 (Stenvers et al, 2019, Christou et al, 2018)), while omental rather than subcutaneous fat plays a crucial role in the pathogenesis of the metabolic syndrome (reviewed in the references 14-15).

We added 4 sentences in the Introduction to further highlight the novelty (1 sentence p. 2, lines 44-46; 3 sentences p. 2, lines 61-65).

2. *Second, there are a lot of statements that are not supported by original experimental data. a) It refers to all ChIP data that show only the results of quantitation and b) all data on functional status of p65. These require original Western blots of both total and phosphor-p65 side by side as the readers as well as reviewers cannot rely just on bars.*

a) The quantification of ChIP data used in our work is very **well established (please see the reference 65** in the manuscript, and electrophoresis gels with the amplicons are not common rule). In addition, our BMAL1 ChIP-qPCR data have now been validated with ChIP-seq data.

b) The paragraph related to the “measurement of NF- κ B activity” **has been edited and is now detailed in Methods, p. 20, lines 755-764**. We chose a precise method to specifically measure p65 phosphorylation at serine 536. This ELISA method has been successfully used by ourselves and others **to quantify the active form of NF- κ B with a limited amount of human biopsies** (see Abou-Samra et al, 2020, *Journal of Cachexia, Sarcopenia and Muscle*) and is actually **more accurate** than stripping the blot for multiple targets with similar molecular weights (p65 and phosphorylated p65). Yet, we also provided the original Western blot of total p65 levels and showed that these levels did not change (Fig.S8E-F).

3. *Each Figure legend should provide info for the number of samples used and statistical test used to assess significance (if it seems to be redundant).*

We have now provided the details in all the legends (please note that the legends of Figs. 3-4 >350 words each). We also edited the Methods to provide a more **accurate definition of the term “independent cultures” (Methods, p.18, lines 654-656)**.

4. *The manuscript is not clearly written and is very hard to go through. This includes both ambiguous statements and mistakes. Below are several examples of this: There are numerous examples when the order of Figure panels does not correspond to the order of the reference in the text making it very uneasy to follow. There is no reference in the text to Fig. 1A*

In the former version, Fig. S1 was cited in the main text before Fig. 1, and **this has been edited**. We reorganized the figures according to their appearance in the main text. We also rewrote a large part of the manuscript. We feel that reading is now much easier.

5. *Lane 104 –according to Fig. S1C Clock expression in decreased in contrast to the statement*

We thank the Reviewer for this comment. This sentence has been deleted.

6. *The phrase “interplay between clock disruption and NF- κ B –mediated inflammation” (Abstract) is misleading as it gives the impression that clock disruption is the primary event in this cross-talk*

We replaced by “Here, we investigated the interplay between circadian clocks and NF- κ B-

mediated inflammation in human obesity”.

7. *Comment 7: Moreover, there is no clock disruption observed here, the clock is still functional with reduced amplitude and period lengthening.*

We initially used this terminology because “clock disruption” has been extensively used in a similar context (e.g., Kohsaka et al., 2007, reference 9; Hong et al., 2018, reference 2). In this revised version, **this term has been replaced** by others, which may be more appropriate like perturbations, dysfunction ...

8. *Are the red bars on Fig.S1B and C the same as red bars in S1D and E? May these need to be combined (S1b and D and S1C and E) to avoid confusion?*

In order to avoid confusion, **we added two subtitles within Fig. S1**, thereby making it clear to the reader that the bars (*in S1D and E*) are not the same.

9. *According to established nomenclature names in all caps (PER, CLOCK, BMAL1) refer to proteins, not to genes (Per, Clock, Bmal1 italicized)*

According to the guidelines provided by the HUGO gene nomenclature committee, human gene symbols are italicized, with all letters in uppercase. Protein designations are the same as the gene symbol, except that they are not italicized, and are in all caps.

For mouse, gene symbols are italicized, with only the first letter in uppercase and the remaining letters in lowercase. Protein designations are the same as the gene symbol, but are not italicized and all are upper case. We carefully went through the text and found one mistake for the mouse *Ikk* gene expression, which has been corrected.

Reviewer #3

1. *While interesting, most of these results have been already published in mouse models, namely the regulation of the circadian clock in obesity (Kohsaka et al., 2007), the role of NF-κB in this regulation through competition with BMAL1 (Hong et al., 2018), and the regulation of the Ccl2 transcription by BMAL1 (Nguyen et al., 2013). The new findings consist only in the confirmation of these results in human adipose tissue.*

The main finding of our paper is to reveal an interplay between inflammation and circadian systems in human omental fat and to show its involvement in the pathogenesis of obesity and its complications. We feel that this message is original. Firstly, such interplay as well as BMAL1 cistrome have only been described in the liver of obese mice, not in adipose tissue (Hong et al., 2018; reference 2). As BMAL1 predominantly binds DNA in a tissue-specific manner (Beytebiere et al, 2019; reference 8), we cannot infer that similar data would have been obtained in mouse fat tissue. The validity of this assertion may be extended to the regulation of Ccl2 transcription, which has been only showed in mouse monocytes (Nguyen et al., 2013, reference 7). Secondly, some data obtained in mouse may not be translated into humans. Thirdly, the few reports on clock machinery in human adipose tissue did not study the interplay with NF-κB and mainly focused on the subcutaneous fat depot (e.g., references 47-48 (Stenvers et al, 2019, Christou et al, 2018)), while omental rather than subcutaneous fat plays a crucial role in the pathogenesis of the

metabolic syndrome (reviewed in the references 14-15). Eventually, we feel that our main message may open new therapeutic perspectives.

We added 4 sentences in the Introduction to further highlight the novelty (1 sentence p. 2, lines 44-46; 3 sentences p. 2, lines 61-65).

2. *Figure 1B, 1H, S1C and S1E show the expression of circadian clock genes in different tissues and cells from human donors. However, considering the small number of donors, to what extent this difference could be only due to different sampling times in the different groups. It could be interesting to correlate clock gene expression with the sampling time. Without this information, it is almost impossible to analyze this data.*

Biopsies from non-obese and obese patients have been obtained at the same time, within an interval of 2 hours (between 10am-12). Collection time for paired biopsies of omental and subcutaneous adipose tissue has also been added.

We now provide the times of biopsy collection in the Methods (one new sentence, p. 17, lines 614-615; one new sentence, p. 17, lines 624-625).

3. *In Figure 2, authors focus on the regulation of PER2 by BMAL1 and NF-κB. It will be interesting to see if their findings are also true genome wide, as described in (Hong et al., 2018).*

Indeed, to date, the few studies which have investigated BMAL1 cistrome in human tissues or cells have only used human cell lines (references 33 and 51). We performed **BMAL1 ChIP-seq in omental preadipocytes from non-obese and obese patients**. We have identified several thousand BMAL1 binding sites in these cells. We now demonstrate that BMAL1 also binds in close proximity to NF-κB consensus motifs in human preadipocytes. Importantly, we found that **BMAL1 occupancy is repositioned genome-wide in inflamed omental adipose tissue in obesity** and we defined the pathways and biological processes involved.

These data have been summarized in the new Fig.5, Fig.6, Fig.7 and Fig.S4. These results have also been described in Results (p. 6-7, lines 227-278) and discussed (p. 10, lines 379-401). A link to the UCSC browser has been provided in the Reporting Summary.

4. *In Figure 3 and 4, authors show that inhibition or overexpression of the NF-κB pathway in obese cells increases PER2 expression and shorten the circadian period. What is the effect of NF-κB inhibition, p65/p50 siRNA or p65 overexpression on PER2 expression in non-obese cells? Is the effect of the inhibitors similar or blunted?*

We have now **presented the results of experiments in non-obese cells:**

- The effect of pharmacological NF-κB inhibition in non-obese cells is presented in a new panel of Fig. S3 (**Fig. S3B**)
- The effect of p65 siRNA in non-obese cells in a new panel of Fig. S3 (**Fig. S3C**)
As expected, NF-κB inhibition did not modify period length in non-obese subjects, likely because baseline NF-κB activity was already low in cells of these subjects.
These data have been described in the Results (p. 5, lines 186-188).
- The effect p65 overexpression in non-obese cells is now shown in a new panel of Fig. S3 (**Fig. S3H**) and related **data described in Results (p. 5, lines 193-194).**

5. *In Figure 5, authors show the role of BMAL1 binding on the Ccl2 expression. To what extent the binding of NF-κB on this promoter is affected in obese and non-obese samples?*

We have performed new experiments and measured p65 binding to *CCL2* promoter in omental preadipocytes from non-obese and obese patients. We found an increase in p65 occupancy at the κ B sites located near the TSS of *CCL2* in obese OPAs (vs. non obese OPAs).

These data have been summarized in the new Fig.S3I. These results have also been described in the Results (p. 6, lines 206-207).

REVIEWER COMMENTS

Reviewer #1 (Remarks to the Author):

The authors performed additional requested experiments and included new data. The revised manuscript is now focused on the inflammatory cytokine CCL2. This focus is justified (authors answer) since no BMAL1 peaks were identified on other pro-inflammatory factors including TNF α , IL6, CCL4 and CCL5 (authors answer and new results). The regions with differential BMAL1 binding between obese and non-obese cells are not enriched in inflammatory-related pathways but in metabolism (new figure 7C), strongly suggesting that inflammation is not the main event impacted by clock dysfunction. The link with NF- κ B (in the title of the article) is strengthened with p65 ChIP and new data obtained from mice with conditional inactivation of Ikk β restricted to adipocytes but the revised version stresses also potential involvement of FOXO signalling pathway (Figure 7). Altogether, the data of the revised version of the manuscript are confused and the inter-relation between PER2 and inflammatory-related pathways in human omental fat suggested in the first version of the manuscript is no more so obvious. In addition, concerning the BMAL1 ChIPseq, it is mentioned that peaks were selected based on presence in at least 2 patients/ group with 3 patients included per groups. This strategy is adequate for preliminary approaches but will require additional patients for data representativeness. Concerning the limitation of the cohort (potential impact of high fasting glucose and not of obesity per se), the authors added two sentences in the discussion part but inclusion of "metabolic healthy obese individuals" will have strengthened the data.

As requested, the authors modified the method section to describe more precisely the protocol to isolate OPA or "omental preadipocytes". It is now well accepted that stromal cells from fat depots are composed by vascular cells, immune cells, fibro-blast, preadipocytes and progenitors. In vitro expansion of cells obtained after colla-genase digestion of fat tissue as what used in the present study is described for adipose-derived stromal cells (ADSC) and not preadipocytes (will require additional selection steps). Single cell RNA seq approaches recently demonstrate the great heterogeneity of adipose tissue stromal cells, progenitors and preadipocytes. The sentence "Preadipocytes can account for up to 50 % of the cells in fat tissue" must be removed. "adipocytes, either mature or progenitor cells": Progenitor cells are neither adipocytes (per definition) nor only preadipocytes, so please correct the sentence. The sentence "PDGFR α + CD31- CD45- OPAs » isolated from obese patients" must be deleted since cells were neither immunoselected nor directly studied after isolation. Cell density (cell number/cm²) is never mentioned (neither for P0 nor Pas-saged cells from P0 cryopreserved cells) but to avoid changes due to distinct numbers of cell cycle between non obese and obese cells it must be controlled. Finally, in most all figures, significance of "n" are unclear. "All data are represented as mean SEM [n = 5 (A-D) or 8 (E-I) patients per group, each culture obtained from a single patient" Why data from Figure 1A to D are obtained from 5 patients per group and not from the whole cohort (8 patients)? Did you perform "culture" of mature adipocytes? Similarly, the term "independent culture" is unclear for OPA. Do you mean technical replicates (distinct P1 cells from P0 cryopreserved cells from one patients or distinct passages from cryo-preserved P0 cells from one patients)? What does mean "n = 8 cultures per condition, from 8 different patients"? Do the authors perform statistical analyses on technical replicates, biological replicates (cryopreserved P0 cells from distinct patients) or both? Same questions for the means \pm SEM (from technical replicates, biological replicates or both). « n = 4 IPs per treatment from 2 obese patients, with 2 independent cultures per patient" Does it mean that data originated from only two patients? Same question for figure 3L, 4G.

Reviewer #2 (Remarks to the Author):

The revised version of the manuscript was significantly improved in respect to its structure and clarity of presentation of the data. Impressive amount of information have been added (such as genome-wide identification of BMAL1 binding sites in adipose tissue); however it did not solve the main point of critique expressed by reviewers, it did not add exciting novelty of the findings except for confirmation of previously published results in mouse models of obesity using human adipose tissues. Moreover, adding results of genome-wide profiling dilutes the focus of the manuscript.

There are also some technical issues that require clarification or different way of presentation. Below are some specific points:

1. In contrast to the statement, data presented in Fig.2D and 2E suggest that BMAL1 and RNA POLII binding to the promoters in OPAs remain rhythmic but is reduced in the amplitude. This will be consistent with luciferase imaging (Fig.1H) demonstrating rhythmicity in cultured OPAs isolated from obese patient
2. Data presented in Fig.3C states that pharmacological inhibition of NF-kB shortens the period of oscillations in cells isolated from obese patients, which is particularly effective after JSH-23 treatment (Fig. 3E). Which interval was used to calculate period? On the recording both look the same except for initial several hours that may result from drug treatment
3. All assays for measuring NF-kB activity was done using ELISA assay. It would be important to show Western blot data using phosphor-specific antibodies as this is the most representative way to support the statement. There is one Western blot data in Suppl. Figures; however, it is for total protein only; no change in total protein and changes in phosphor-protein must be shown side by side
4. All major conclusions regarding clock dysfunction in obesity are based on transcriptional data; however, to claim physiological significance, they have to be supported by protein analysis, which is absent
5. Model presenter in Fig.4H is confusing as there is no direct evidence that repression of PER2 promoter directly affects CCL2 secretion rhythm. Moreover, the clock in obese tissue is not broken as cells remain rhythmic with a different period
6. Fig. 5 and 6 show very modest and not statistically significant effect of obesity on BMAL1 binding to PER2 (Fig. 5C) and dramatic effect on promoters of other genes (Fig.6). The natural question arises: why the entire work is focused on PER2 and not on other transcriptional targets?
7. Data demonstrating no change in Ccl2 expression at the time of HFD-induced period lengthening must be shown
8. Does sodium salicylate have an effect on cycling in tissue culture?
9. Fig. 8J – no statistically significant differences, therefore the authors cannot state that the downregulation of Per2 expression was corrected.
10. In real-time PCR data, how the expression was normalized? It says "GAPDH (???)", 18S or Gapdh. Were Gapdh levels affected by obesity? It is important to add this info to all mRNA quantitation data (i.e. Per2/Gapdh relative abundance).

Overall, the manuscript will benefit if after editing it will be submitted to a more specialized journal

Reviewer #3 (Remarks to the Author):

Authors performed a great job to adequately answer reviewers' queries. The new experiments, in particular the genome wide analysis of BMAL1 binding in human obese adipose tissue, strongly strengthen the conclusions of the article. There are however a few things to correct before publication:

- Fig 3L: authors did not provide any information about the efficiency of the p50 siRNA
- Fig 8J: authors claimed that "the downregulation of Per2 expression was corrected". However, according to the figure, there is no statistical difference in Per2 expression after treatment with salicylate, and just a tendency. The sentence has therefore to be tuned down.
- In the rebuttal letter, authors claimed that "A link to the UCSC browser has been provided in the Reporting Summary". The link is indeed provided but did not allow the visualisation of the BMAL1 ChIP-Seq data. In addition, a table reporting the analysis these ChIP-Seq data (peak quantification in the different samples) has to be provided as a supplemental table.

Reviewer #1 (Remarks to the Author):

The authors performed additional requested experiments and included new data.

The revised manuscript is now focused on the inflammatory cytokine CCL2. This focus is justified (authors answer) since no BMAL1 peaks were identified on other pro-inflammatory factors including TNF α , IL6, CCL4 and CCL5 (authors answer and new results). The regions with differential BMAL1 binding between obese and non-obese cells are not enriched in inflammatory-related pathways but in metabolism (new figure 7C), strongly suggesting that inflammation is not the main event impacted by clock dysfunction. The link with NF- κ B (in the title of the article) is strengthened with p65 ChIP and new data obtained from mice with conditional inactivation of I κ B β restricted to adipocytes but the revised version stress also potential involvement of FOXO signalling pathway (Figure 7). Altogether, the data of the revised version of the manuscript are confused and the inter-relation between PER2 and inflammatory-related pathways in human omental fat suggested in the first version of the manuscript is no more so obvious. In addition, concerning the BMAL1 ChIPseq, it is mentioned that peaks were selected based on presence in at least 2 patients/ group with 3 patients included per groups. This strategy is adequate for preliminary approaches but will require additional patients for data representativeness.

Concerning the limitation of the cohort (potential impact of high fasting glucose and not of obesity per se), the authors added two sentences in the discussion part but inclusion of "metabolic healthy obese individuals" will have strengthened the data.

As requested, the authors modified the method section to describe more precisely the protocol to isolate OPA or "omental preadipocytes". It is now well accepted that stromal cells from fat depots are composed by vascular cells, immune cells, fibro-blast, preadipocytes and progenitors. In vitro expansion of cells obtained after collagenase digestion of fat tissue as what used in the present study is described for adipose-derived stromal cells (ADSC) and not preadipocytes (will require additional selection steps). Single cell RNA seq approaches recently demonstrate the great heterogeneity of adipose tissue stromal cells, progenitors and preadipocytes. The sentence "Preadipocytes can account for up to 50 % of the cells in fat tissue" must be removed. "adipocytes, either mature or progenitor cells": Progenitor cells are neither adipocytes (per definition) nor only preadipocytes, so please correct the sentence. The sentence "PDGFR α + CD31-CD45- OPAs » isolated from obese patients" must be deleted since cells were neither immunoselected nor directly studied after isolation. Cell density (cell number/cm²) is never mentioned (neither for P0 nor Passaged cells from P0 cryopreserved cells) but to avoid changes due to distinct numbers of cell cycle between non obese and obese cells it must be controlled.

Finally, in most all figures, significance of "n" are unclear. "All data are represented as mean SEM [n = 5 (A-D) or 8 (E-I) patients per group, each culture obtained from a single patient" Why data from Figure 1A to D are obtained from 5 patients per group and not from the whole cohort (8 patients)? Did you perform "culture" of mature adipocytes? Similarly, the term "independent culture" is unclear for OPA. Do you mean technical replicates (distinct P1 cells from P0 cryopreserved cells from one patients or distinct passages from cryo-preserved P0 cells from one patients)? What does mean "n = 8 cultures per condition, from 8 different patients"? Do the authors perform statistical analyses on technical replicates, biological replicates (cryopreserved P0 cells from distinct patients) or both? Same questions for the means \pm SEM (from technical replicates, biological replicates or both). « n = 4 IPs per treatment from 2 obese patients, with 2 independent cultures per patient" Does it mean that data originated from only two patients? Same question for figure 3L, 4G.

Reviewer #2 (Remarks to the Author):

The revised version of the manuscript was significantly improved in respect to its structure and clarity of presentation of the data. Impressive amount of information have been added (such as genome-wide identification of BMAL1 binding sites in adipose tissue); however it did not solve the main point of critique expressed by reviewers, it did not add exciting novelty of the findings except for confirmation of previously published results in mouse models of obesity using human adipose tissues. Moreover, adding results of genome-wide profiling dilutes the focus of the manuscript. There are also some technical issues that require clarification or different way of presentation.

Below are some specific points:

1. In contrast to the statement, data presented in Fig.2D and 2E suggest that BMAL1 and RNA POLII binding to the promoters in OPAs remain rhythmic but is reduced in the amplitude. This will be consistent with luciferase imaging (Fig.1H) demonstrating rhythmicity in cultured OPAs isolated from obese patient
2. Data presented in Fig.3C states that pharmacological inhibition of NF-kB shortens the period of oscillations in cells isolated from obese patients, which is particularly effective after JSH-23 treatment (Fig. 3E). Which interval was used to calculate period? On the recording both look the same except for initial several hours that may result from drug treatment
3. All assays for measuring NF-kB activity was done using ELISA assay. It would be important to show Western blot data using phosphor-specific antibodies as this is the most representative way to support the statement. There is one Western blot data in Sippl. Figures; however, it is for total protein only; no change in total protein and changes in phosphor-protein must be shown side by side
4. All major conclusions regarding clock dysfunction in obesity are based on transcriptional data; however, to claim physiological significance, they have to be supported by protein analysis, which is absent
5. Model presenter in Fig.4H is confusing as there is no direct evidence that repression of PER2 promoter directly affects CCL2 secretion rhythm. Moreover, the clock in obese tissue is not broken as cells remain rhythmic with a different period
6. Fig. 5 and 6 show very modest and not statistically significant effect of obesity on BMAL1 binding to PER2 (Fig. 5C) and dramatic effect on promoters of other genes (Fig.6). The natural question arises: why the entire work is focused on PER2 and not on other transcriptional targets?
7. Data demonstrating no change in Ccl2 expression at the time of HFD-induced period lengthening must be shown
8. Does sodium salicylate have an effect on cycling in tissue culture?
9. Fig. 8J – no statistically significant differences, therefore the authors cannot state that the downregulation of Per2 expression was corrected.
10. In real-time PCR data, how the expression was normalized? It says "GAPDH (???)", 18S or Gapdh. Were Gapdh levels affected by obesity? It is important to add this info to all mRNA quantitation data (i.e. Per2/Gapdh relative abundance).

Overall, the manuscript will benefit if after editing it will be submitted to a more specialized journal

Reviewer #3 (Remarks to the Author):

Authors performed a great job to adequately answer reviewers' queries. The new experiments, in particular the genome wide analysis of BMAL1 binding in human obese adipose tissue, strongly strengthen the conclusions of the article. There are however a few things to correct before publication:

- Fig 3L: authors did not provide any information about the efficiency of the p50 siRNA

- Fig 8J: authors claimed that "the downregulation of Per2 expression was corrected". However, according to the figure, there is no statistical difference in Per2 expression after treatment with salicylate, and just a tendency. The sentence has therefore to be tuned down.

- In the rebuttal letter, authors claimed that "A link to the UCSC browser has been provided in the Reporting Summary". The link is indeed provided but did not allow the visualisation of the BMAL1 ChIP-Seq data. In addition, a table reporting the analysis these ChIP-Seq data (peak quantification in the different samples) has to be provided as a supplemental table.

Answers to the reviewers

We have carefully reviewed the comments and have revised the manuscript accordingly. The manuscript with tracked changes (red) has been uploaded. Our responses are given in a point-by-point manner below: each of the reviewers' comments will be in black and our responses will be highlighted in blue.

REVIEWER COMMENTS

Reviewer #1

The authors performed additional requested experiments and included new data.

The revised manuscript is now focused on the inflammatory cytokine CCL2. This focus is justified (authors answer) since no BMAL1 peaks were identified on other pro-inflammatory factors including TNF α , IL6, CCL4 and CCL5 (authors answer and new results).

1. a) *The regions with differential BMAL1 binding between obese and non-obese cells are not enriched in inflammatory-related pathways but in metabolism (new figure 7C), strongly suggesting that inflammation is not the main event impacted by clock dysfunction.*

The referee is right. Herein, we show that clock dysfunction is induced by inflammation and that inflammation is not the main event impacted by clock dysfunction. Instead, **metabolic inflammation and adipose tissue remodeling** may result from BMAL1 relocalization genome-wide.

Accordingly, we now provide a new MS title "*Circadian clock dysfunction in human omental fat links obesity to metabolic inflammation*" and have rewritten the abstract, as well as most of the discussion (pages 9-11, lines 356-425). We have also redrawn our proposed model in Fig. 4H.

b) The link with NF- κ B (in the title of the article) is strengthened with p65 ChIP and new data obtained from mice with conditional inactivation of Ikk β restricted to adipocytes but the revised version stresses also potential involvement of FOXO signalling pathway (Figure 7). Altogether, the data of the revised version of the manuscript are confused and the inter-relation between PER2 and inflammatory-related pathways in human omental fat suggested in the first version of the manuscript is no more so obvious. Altogether, the data of the revised version of the manuscript are confused and the inter-relation between PER2 and inflammatory-related pathways in human omental fat suggested in the first version of the manuscript is no more so obvious.

This criticism has also been raised by referee 2.

BMAL1 relocalization genome-wide results in changes in the expression of *PER2*, as changes in the expression of other genes linked to metabolic inflammation. In agreement with the referee suggestion, we now develop this latter point by adding **new experiments**.

- Thus, we selected several BMAL1 targets based on their relevance in the pathogenesis of obesity, and on a differential BMAL1 binding at their loci in omental fat from obese vs. non-obese subjects. For each target, we measured mRNA and protein levels in omental adipocyte precursors (OAPs). We found that many differential BMAL1 targets coding for factors involved in metabolic inflammation were actually dysregulated at the mRNA and protein levels in obesity and could therefore contribute to the Metabolic Syndrome (Results, page 7, lines 255-282, “Among the regions with differential BMAL1 binding between non-obese and obese OAPs ... levels in obese OAPs”; Fig. 6 has been edited and we added the new Fig. 7).
- We found the same phenomenon in mice. We thus showed that besides *Ccl2*, some targets coding metabolic factors (i.e., extracellular matrix remodeling components) were also dysregulated in HFD-induced obese mice in association with clock dysfunction (Results, page 8, lines 320-326, “when *Ccl2* expression was still unchanged ... suggesting that it might initiate these alterations”, new panel Fig 9.I). This metabolic inflammation was prevented in mice with conditional inactivation of *Ikkβ* restricted to adipocytes (Results, page 9, lines 350-352, “Expression of genes encoding extracellular matrix remodeling components was also significantly improved in AT from treated mice” and new panel in Fig.S6L-N).
- Accordingly, **the discussion** is less focused on CCL2, but **is aimed at integrating metabolic inflammation/adipose tissue remodeling with BMAL1 relocalization in obesity** (Discussion, pages 9-11, lines 356-425). Consequently, the model presented in Fig. 4H has also been redrawn (legend, pages 14-15, lines 534-545) and the abstract fully rewritten.

Accordingly, we now provide a **new title** “Circadian clock dysfunction in human omental fat links obesity to metabolic inflammation”.

2. *In addition, concerning the BMAL1 ChIPseq, it is mentioned that peaks were selected based on presence in at least 2 patients/ group with 3 patients included per groups. This strategy is adequate for preliminary approaches but will require additional patients for data representativeness.*
 - We performed the ChIP-Seq experiment, with n=3 patients per group (GSE149064, password: ulqrocgwlnatzkl). We now provide an excel table including a list of regions identified as differentially targeted by BMAL1 by DESeq2 in Data Source (n=3 patients per group). Detailed raw data including peak quantification (~ 7000 loci) are also provided, with n=3 patients per group.
 - It is important to emphasize that previous BMAL1 ChIP-Seq experiments published in high-impact factor journals have used a **similar or even smaller number of mouse samples or human cell line replicates** (e.g., Perelis *et al.*, *Science*, 2015, n=1; Wu, *et al.*, *Cell Metabolism*, 2017, n=1 and Levine *et al.*, *Mol. Cell*, 2020, n= 2). Moreover, in our paper, we provide data from human primary cells, which is more relevant than those from mouse samples/ human cell lines.
 - Eventually, ChIP-Seq may be viewed as **an exploratory method**. We selected some targets that may be involved into the pathogenesis of obesity (from Fig. 6, in the current version) and we now **confirm** the obesity-induced changes in BMAL1 binding **by using conventional ChIP** (n=4/group; new Fig .7). We have also measured target gene expression and protein levels in our whole cohort of patients (n=8/group) (Fig 7, and Results page 7, lines 274-282), thereby strengthening our initial ChIP-seq data.
3. *Concerning the limitation of the cohort (potential impact of high fasting glucose and not of obesity per se), the authors added two sentences in the discussion part but inclusion of “metabolic healthy obese individuals” will have strengthen the data.*

We acknowledge that it would have been interesting to study “metabolic healthy obese individuals”. Yet, one may speculate that BMA1 relocalization would have been less pronounced, as these patients have less omental fat and less inflammation than metabolic unhealthy obese ones (Bluher, *Endocrine reviews* 2020).

4. *As requested, the authors modified the method section to describe more precisely the protocol to isolate OPA or “omental preadipocytes”. It is now well accepted that stromal cells from fat depots are composed by vascular cells, immune cells, fibroblast, preadipocytes and progenitors. In vitro expansion of cells*

obtained after collagenase digestion of fat tissue as what used in the present study is described for adipose-derived stromal cells (ADSC) and not preadipocytes (will require additional selection steps). Single cell RNA seq approaches recently demonstrate the great heterogeneity of adipose tissue stromal cells, progenitors and preadipocytes.

a) The sentence "Preadipocytes can account for up to 50 % of the cells in fat tissue" must be removed. "adipocytes, either mature or progenitor cells": Progenitor cells are neither adipocytes (per definition) nor only preadipocytes, so please correct the sentence.

In agreement with the referee suggestion, we removed the inappropriate sentences (both removed from page 3). Moreover, according to our further characterization of these cell types and to recent studies in the field (reviewed in Marcelin *et al.*, *J Clin Invest*, 2019), we removed the term "preadipocytes" and replaced it by "adipocyte precursors" (OAPs, omental adipocyte precursors) throughout the MS.

b) The sentence "PDGFR α + CD31-CD45- OAPs isolated from obese patients" must be deleted since cells were neither immunoselected nor directly studied after isolation.

Human adipocyte precursors, which are prone to adipogenesis *in vitro*, are enriched in CD34⁺, PDGFR α ⁺, CD45⁻ (a leukocyte marker), and CD31⁻ (an endothelial cell-specific marker) cell populations derived from the stromal vascular fraction of adipose tissue (reviewed in Marcelin *et al.*, *J Clin Invest*, 2019). **We now fully characterize the adipocyte precursors by flow cytometry analysis** and measure the expression of an additional marker, CD34. This marker, together with the other ones (PDGFR α , CD31, CD45) are now studied **in our whole cohort of patients** (n=8 patients in each non-obese and obese group).

The Methods' section has been modified accordingly (page 18, lines 680-693) "We defined the OAPs as cells positive for Platelet-Derived Growth Factor Receptor α (PDGFR α ⁺ or Cluster of Differentiation 140a⁺, CD140a⁺) and CD34⁺ but negative for markers of leucocytes or endothelial cells (CD45⁻ or CD31⁻) using FACScalibur and FACSCantoll flow cytometers, with FlowJo Software (BD Biosciences). Briefly, flow cytometry analysis using Alexa Fluor[®] 647 mouse anti-human CD140a (BD Biosciences, #562798) allowed to determine that 97.9 \pm 0.4 % cells were PDGFR α ⁺ without significant difference between non-obese and obese cells (n=8 patients/ group). Intensity of CD9 expression in PDGFR α ⁺ cells was detected by simultaneous staining with Alexa Fluor[®] 647 mouse anti-human CD140a and CD9-PE antibodies (BD Biosciences, #555372). Simultaneous detection of CD34 using APC/Cyanine7 mouse anti-human CD34 (BioLegend, #343613) showed that 90 \pm 1 % cells were CD34⁺ without significant difference between non-obese and obese cells (n=8 patients/ group). The absence of CD31⁺ and CD45⁺ cells was attested using mouse anti-human CD45-FITC (Immunotools, #21810453X2) and mouse anti-human CD31-FITC antibodies (Miltenyi Biotec, #130-117-539) separately in PDGFR α ⁺ cells (n=8 patients/ group)."

We rephrased the sentence "PDGFR α + CD31-CD45- OAPs isolated from obese patients" accordingly "OAPs; PDGFR α ⁺ CD34⁺CD31⁻CD45⁻ cells obtained from omental stromal-vascular cells, see Methods" (page 3, lines 94-95). The word "isolated" has been removed from page 3.

5. Cell density (cell number/cm²) is never mentioned (neither for P0 nor Passaged cells from P0 cryopreserved cells) but to avoid changes due to distinct numbers of cell cycle between non obese and obese cells it must be controlled.

We now clearly indicate cell density in Methods (page 17 line 671; page 18 line 698, line 704, lines 705-796; page 19, line 722, line 723, lines 757-759 and line 771). Cell density was identical in non-obese and obese cells.

6. a) Finally, in most all figures, significance of "n" are unclear. "All data are represented as mean SEM [n = 5 (A-D) or 8 (E-I) patients per group, each culture obtained from a single patient" Why data from Figure 1A to D are obtained from 5 patients per group and not from the whole cohort (8 patients)?

We complied with the reviewer suggestion. Whenever possible, data have been obtained from the whole cohort of patients. This is now the case for Fig 1A to D, where measurements were obtained from 8 patients /group. The appropriate figure and legend have been modified accordingly. This is also the case for another figure, where the number of patients has been increased from 5 to 8 (see Fig3B and its legend, page 13, line 498).

b) *Did you perform “culture” of mature adipocytes?*

Thank you, this was a mistake. Human mature adipocytes are not cultured in this experiment. These cells are briefly incubated in Krebs-albumin buffer, as previously indicated in Methods.

c) (...) *What does mean “n = 8 cultures per condition, from 8 different patients”?*

Sorry, this sentence was misleading; it has now been rephrased: “n=8 patients/ group” (Legend Fig 1, line 473).

d) *Similarly, the term “independent culture” is unclear for OPA. Do you mean technical replicates (distinct P1 cells from P0 cryopreserved cells from one patients or distinct passages from cryopreserved P0 cells from one patients)? Do the authors perform statistical analyses on technical replicates, biological replicates (cryopreserved P0 cells from distinct patients) or both? Same questions for the means+/- SEM (from technical replicates, biological replicates or both). « n = 4 IPs per treatment from 2 obese patients, with 2 independent cultures per patient)” Does it mean that data originated from only two patients? Same question for figure 3L, 4G.*

We now **clarify how cell culture and biological donors’ replicates were used in Methods** page 18, line 673-679. *“In some experiments, we performed independent cultures (i.e., run at different times and for each time, from a new vial of cryopreserved OAPs) from patients chosen at random. We generated at most two independent cultures from a given biological donor. The donors were always chosen at random to avoid any bias of selection. The final number of independent cultures is always indicated in the legends. For the experiments with non-obese/ obese comparisons, the same number of cultures from non-obese and obese OAPs was always processed simultaneously.”*

We also indicated in the legend of Fig.2 *“n=6-8 independent cultures (i.e., run at different times and for each time, from a new vial of cryopreserved OAPs) from 3-4 subjects in each non-obese and obese group”.* (page 13, lines 485-487”).

We acknowledge that the sentence *“n = 4 IPs per treatment from 2 obese patients, with 2 independent cultures per patient”* may be ambiguous. It has been rephrased *“n= 4 independent cultures from 2 obese patients”* (Legend Fig 3F, G, page 14, line 505). This indeed implies that only 2 obese patients were used. However, in this case, each patient serves as its own control (paired t test) and very huge differences were observed between the treatment and the control conditions. We do not feel that the main message (potent effect of NF-κB inhibitors) would have changed if we had used a greater number of biological samples. Same comment for Fig 3L and 4G (Legends, page 4, lines 512-513).

Reviewer #2

General comments

The revised version of the manuscript was significantly improved in respect to its structure and clarity of presentation of the data. Impressive amount of information have been added (such as genome-wide identification of BMAL1 binding sites in adipose tissue).

1. *However it did not solve the main point of critique expressed by reviewers, it did not add exciting novelty of the findings except for confirmation of previously published results in mouse models of obesity using human*

adipose tissues.

Reviewer 2's main comment regarding "a lack of novelty" implies a comparison with the study from Hong *et al.*, *Genes Dev.*, 2018. However, in the present manuscript, the issue is different: we now address the function of omental adipose tissue and its role in the development of cardiometabolic disease, which cannot be compared to those of liver. Thus, the message in the present article is original.

The article from Hong *et al.*, 2018 focuses on BMAL1 cistrome in mouse liver. As BMAL1 predominantly binds DNA in a tissue-specific manner (Beytebiere *et al.*, *Genes Dev.*, 2019), we could not even infer that similar data would have been obtained in mouse fat tissue. Moreover, some data obtained in mouse may not be translated to humans (here, we studied human omental fat biopsies). In our paper, BMAL1 targets were actually not the same as the ones identified in mouse liver. Pathways identified in mouse liver included nutrient-responsive anabolic pathways such as lipogenesis and lipid metabolism, or cholesterol biosynthesis. In this new article, we highlight targets encoding factors involved in endocytosis, proteoglycans, FOXO signaling, RNA metabolism or response to hypoxia. Specifically, we focused on the involvement of proteins involved in adipocyte vesicular trafficking, extracellular matrix components, or key enzymes in metabolic processes, chromatin organization and oxidative stress. The altered production of these proteins may be involved in metabolic inflammation and unhealthy adipose remodeling leading to the metabolic syndrome.

2. *Moreover, adding results of genome-wide profiling dilutes the focus of the manuscript.*

The referee is right; this criticism has also been raised by referee 1 (see our answer 1b to referee 1). We now re-focus the manuscript on BMAL1 relocalization promoting metabolic inflammation in obesity rather than simply inflammation. We **add some new experiments in order to develop this point:**

- Thus, we selected several BMAL1 targets based on their relevance in the pathogenesis of obesity and on a differential BMAL1 binding at their loci in omental fat from obese vs. non-obese subjects. For each target, we measured mRNA and protein levels in omental adipocyte precursors (OAPs). We found that many differential BMAL1 targets coding for factors involved in metabolic inflammation were actually dysregulated at the mRNA and protein levels in obesity and could therefore contribute to the Metabolic Syndrome (Results, page 7, lines 255-282, "*Among the regions with differential BMAL1 binding between non-obese and obese OAPs ... levels in obese OAPs*"; Fig. 6 has been edited and we added the new Fig. 7).
- We found the same phenomenon in mice. We thus showed that besides *Ccl2*, some targets coding metabolic factors (*i.e.*, extracellular matrix remodeling components) were also dysregulated in HFD-induced obese mice in association with clock dysfunction (Results, page 8, lines 320-326, "*when Ccl2 expression was still unchanged ... suggesting that it might initiate these alterations*", new panel Fig 9.I). This metabolic inflammation was prevented in mice with conditional inactivation of *Ikkβ* restricted to adipocytes (Results, page 9, lines 350-352, "*Expression of genes encoding extracellular matrix remodeling components was also significantly improved in AT from treated mice*" and new panel in Fig.S6L-N).
- Accordingly, **the discussion** is less focused on CCL2, but **is aimed at integrating metabolic inflammation/adipose tissue remodeling with BMAL1 relocalization in obesity** (Discussion, pages 9-11, lines 356-425). Consequently, the model presented in Fig. 4H has also been redrawn (legend pages 14-15, lines 534-545) and the abstract fully rewritten.
We provided a new title "*Circadian clock dysfunction in human omental fat links obesity to metabolic inflammation*".

Specific points

There are also some technical issues that require clarification or different way of presentation. Below are some specific points:

1. *In contrast to the statement, data presented in Fig.2D and 2E suggest that BMAL1 and RNA POLII binding to the promoters in OPAs remain rhythmic but is reduced in the amplitude. This will be consistent with*

luciferase imaging (Fig.1H) demonstrating rhythmicity in cultured OPAs isolated from obese patient.

Fig. 2D and 2E. When using two-way ANOVA followed by *post hoc* Sidak's test, we failed to observe any significant differences in BMAL1 (or RNA POLII) binding to *PER2* in obese cells between the 2 collection times (27hr and 38hr post-synchronization), while a significant effect was observed in non-obese OPAs. Yet, we cannot exclude the possibility that obese cells maintain some rhythmicity but of reduced amplitude. We thus rephrased the relative sentences in Results and indicated that we did not disclose a *significant* rhythm of BMAL1 (and RNA POLII) binding to *PER2* in obese cells (Results, page 4, lines 151 and 154).

2. *Data presented in Fig.3C states that pharmacological inhibition of NF-κB shortens the period of oscillations in cells isolated from obese patients, which is particularly effective after JSH-23 treatment (Fig. 3E). Which interval was used to calculate period? On the recording both look the same except for initial several hours that may result from drug treatment.*

In the experiments with *in vitro* pharmacological inhibition of NF-κB, data recorded from the first 12h were excluded from the analysis. As already indicated in Methods, bioluminescence was recorded continuously for 4–7 days. For this specific experiment, the recording lasted 4 days (*i.e.*, the period was determined using the data from day 0.5 to day 4). Results indicate a clear effect of NF-κB Inhibition.

3. *All assays for measuring NF-κB activity was done using ELISA assay. It would be important to show Western blot data using phosphor-specific antibodies as this is the most representative way to support the statement. There is one Western blot data in Suppl. Figures; however, it is for total protein only; no change in total protein and changes in phosphor-protein must be shown side by side.*

We complied with the referee suggestion. We now show a Western Blot with phosphorylated p65 protein and total p65 levels in obese vs. non-obese subjects (page 20, lines 786-788, novel panel E in Fig. S8). As expected, there was a huge increase in phosphorylated p65 levels in obesity, while total p65 did not change. These data are consistent with those obtained by Phospho-NF-κBp65 ELISA.

4. *All major conclusions regarding clock dysfunction in obesity are based on transcriptional data; however, to claim physiological significance, they have to be supported by protein analysis, which is absent.*

The criticism of the referee is fully relevant. We **now focus on a panel of BMAL1 targets**, known to be involved in the **pathogenesis of obesity**, that were differentially bound by BMAL1 in obese vs. non-obese subjects. **For each target, we measured mRNA and protein levels in OAPs**. We found that many differential BMAL1 targets coding for factors involved in metabolic inflammation were actually dysregulated at the mRNA and protein levels in obesity and could therefore contribute to the Metabolic Syndrome (Results, page 7, lines 276-282, "*Next, we measured target gene expression and protein levels in our whole cohort of patients (...) transcript and protein levels in obese OAPs*").

5. *Model presenter in Fig.4H is confusing as there is no direct evidence that repression of PER2 promoter directly affects CCL2 secretion rhythm. Moreover, the clock in obese tissue is not broken as cells remain rhythmic with a different period—*

We did not actually provide any evidence that repression of *PER2* promoter directly affects *CCL2* rhythm. We modified the cartoon to avoid any confusion and also updated the model based on new data. The legend was adapted accordingly (pages 14-15, lines 534-545).

Indeed, the cartoon of the clock is aimed at indicating that clock function is altered in cells from obese patients (*e.g.*, the period is lengthened). We now indicate "circadian desynchrony" on top of the clock.

6. *Fig. 5 and 6 show very modest and not statistically significant effect of obesity on BMAL1 binding to PER2 (Fig. 5C) and dramatic effect on promoters of other genes (Fig.6). The natural question arises: why the entire work is focused on PER2 and not on other transcriptional targets?*

It is well established that *PER2* is one of the most important factors for setting the period of the circadian

clock (Chen *et al.*, *Mol Cell.*, 2009; Wilkins, *et al.*, *PLoS Comput Biol.*, 2007). In addition, tools available to monitor *PER2* transcription have been extensively described and herein, we monitored its period in OAPs as a readout for clock function. Although we acknowledge that ChIP-Seq experiments, performed in a small number of patients, showed a very modest and non-significant effect of obesity on BMAL1 binding to *PER2* promoter (Fig. 5C-D), conventional ChIP showed a striking decreased binding (~7-fold) in obese vs non-obese subjects (Fig 2B). Decrease in *PER2* mRNAs (Fig 1G) and period lengthening of *Per2* oscillations further support an effect of obesity on the clock repressor (Fig. 1H, I).

However, ChIP-Seq experiments enlightened a dramatic effect of obesity on changes in binding enrichment of BMAL1 at other loci (see fig. 6). We therefore **did not focus on *PER2* only, but rather expanded our focus to new BMAL1 targets, known to be involved in the pathogenesis of obesity and metabolic syndrome**. We found that many genes with differential BMAL1 binding between non-obese and obese OAPs code for factors involved in metabolic inflammation were actually dysregulated at the mRNA and protein levels in obesity and could therefore contribute to the Metabolic Syndrome (new Fig. 7). (Results, page 7, lines 255-282).

7. *Data demonstrating no change in Ccl2 expression at the time of HFD-induced period lengthening must be shown.*

Has been done. These data are now added as a new panel in Fig.9I.

8. Does sodium salicylate have an effect on cycling in tissue culture?

Salicylate treatment given *in vivo* for 1 month to mice simultaneously rendered obese by a HFD did not significantly change *PER2::LUC* period length in tissue explants cultured *ex vivo* (not shown). However, it should be emphasized that such salicylate treatment was of short duration and that period lengthening of *PER2* oscillations increased with HFD duration (higher after 3 months than 1 month of HFD; Compare Fig 9G vs. 9H. We therefore cannot exclude the possibility that the experimental conditions were not optimal to address this issue.

9. *Fig. 8J – no statistically significant differences, therefore the authors cannot state that the downregulation of Per2 expression was corrected.*

The referee is right. We now re-write more carefully this sentence “*the downregulation of Per2 expression was attenuated ..*” (now Fig.9K) (Results, page 9, lines 338-339).

10. *In real-time PCR data, how the expression was normalized? It says “GAPDH (???)”, 18S or Gapdh. Were Gapdh levels affected by obesity? It is important to add this info to all mRNA quantitation data (i.e. Per2/Gapdh relative abundance).*

As required, we added this information. Now, in the Method Section, we indicate that the expression of the studied gene is normalized to the reference gene: “*normalized to GAPDH for human adipose tissue, adipocytes and OAPs, 18s for mouse adipose tissue and adipocytes and Gapdh for mouse APs*” (page 21, lines 804-805). No more place to indicate this in the figure legends (limited to 350 words).

Whenever used, expression of the reference gene was **not impacted** by the conditions (**no significant differences** found between non-obese and obese patients, subcutaneous and omental fat, collection-time, and no significant effect of the treatments).

Overall, the manuscript will benefit if after editing it will be submitted to a more specialized journal. We feel that we have suitably addressed all the criticisms raised by the referees. Our data are novel and the main message of this paper is of interest for a broad readership.

Reviewer #3 (Remarks to the Author):

Authors performed a great job to adequately answer reviewers' queries. The new experiments, in particular the genome wide analysis of BMAL1 binding in human obese adipose tissue, strongly strengthen the conclusions of the article. There are however a few things to correct before publication:

1. Fig 3L: authors did not provide any information about the efficiency of the p50 siRNA

We have added this result, now shown as a new panel (Fig. S3B), which confirms the efficiency of the knock-down. A sentence has also been added to mention this result (Results, "with an additional 85% decrease in p50 mRNA levels", page 5, lines 184-185).

2. Fig 8J: authors claimed that "the downregulation of Per2 expression was corrected". However, according to the figure, there is no statistical difference in Per2 expression after treatment with salicylate, and just a tendency. The sentence has therefore to be tuned down.

The referee is right. We now re-write more carefully this sentence "*the downregulation of Per2 expression was attenuated ..*" (now Fig.9K) (Results, page 9, lines 338-339).

3. In the rebuttal letter, authors claimed that "A link to the UCSC browser has been provided in the Reporting Summary". The link is indeed provided but did not allow the visualisation of the BMAL1 ChIP-Seq data. In addition, a table reporting the analysis these ChIP-Seq data (peak quantification in the different samples) has to be provided as a supplemental table.

The data sets generated and analyzed during the current study are available in a GEO repository with the identifier GSE149064 (publicly available upon publication), and the Reviewers can use the following code: password: ulqrocgwlnatzkl.

<https://www.ncbi.nlm.nih.gov/geo/query/acc.cgi?acc=GSE149064>

We apologize for the Reviewer's issue with the USCC browser as it seemed to be the correct link.

https://genome.ucsc.edu/cgi-bin/hgTracks?db=hg38&lastVirtModeType=default&lastVirtModeExtraState=&virtModeType=default&virtMode=0&nonVirtPosition=&position=chrX%3A15367502%2D15795581&hgid=925855909_5WoMtcjvCOitiKTYPclArFePBxhr

Moreover, as required, peak quantification is now provided in the data source file.

REVIEWERS' COMMENTS

Reviewer #1 (Remarks to the Author):

The authors answered to the concerns and added new experiments. The new data are now in agreement with an impact of circadian clock dysfunction on metabolism and tissue remodelling rather than strictly inflammation. Title, abstract and discussion have been changed and clarity of the manuscript improved.

Reviewer #3 (Remarks to the Author):

This revised article includes the required new experiments, additional information and additional comments that provide further supports to the results. The described results are of great interest and the article is therefore a good candidate for publication.

Reviewer #4 (Remarks to the Author):

Please find below my comments to the authors in response to them addressing Reviewer 2's prior critique:

General comments

The revised version of the manuscript was significantly improved in respect to its structure and clarity of presentation of the data. Impressive amount of information have been added (such as genome-wide identification of BMAL1 binding sites in adipose tissue).

1. However it did not solve the main point of critique expressed by reviewers, it did not add exciting novelty of the findings except for confirmation of previously published results in mouse models of obesity using human adipose tissues.

Reviewer 2's main comment regarding "a lack of novelty" implies a comparison with the study from Hong et al., *Genes Dev.*, 2018. However, in the present manuscript, the issue is different: we now address the function of omental adipose tissue and its role in the development of cardiometabolic disease, which cannot be compared to those of liver. Thus, the message in the present article is original.

The article from Hong et al., 2018 focuses on BMAL1 cistrome in mouse liver. As BMAL1 predominantly binds DNA in a tissue-specific manner (Beytebiere et al, *Genes Dev.*, 2019), we could not even infer that similar data would have been obtained in mouse fat tissue. Moreover, some data obtained in mouse may not be translated to humans (here, we studied human omental fat biopsies). In our paper, BMAL1 targets were actually not the same as the ones identified in mouse liver. Pathways identified in mouse liver included nutrient-responsive anabolic pathways such as lipogenesis and lipid metabolism, or cholesterol biosynthesis. In this new article, we highlight targets encoding factors involved in endocytosis, proteoglycans, FOXO signaling, RNA metabolism or response to hypoxia. Specifically, we focused on the involvement of proteins involved in adipocyte vesicular trafficking, extracellular matrix components, or key enzymes in metabolic processes, chromatin organization and oxidative stress. The altered production of these proteins may be involved metabolic inflammation and unhealthy adipose remodeling leading to the metabolic syndrome.

I believe the latest version of the manuscript presents novel data which adds to the field. I cannot be sure that Reviewer 2 was comparing this work with Hong et al. but agree with the authors that this manuscript is significantly diverse from the former.

2. Moreover, adding results of genome-wide profiling dilutes the focus of the manuscript.

The referee is right; this criticism has also been raised by referee 1 (see our answer 1b to referee 1). We now re-focus the manuscript on BMAL1 relocalization promoting metabolic inflammation in obesity rather than simply inflammation. We add some new experiments in order to develop this

point:

- – Thus, we selected several BMAL1 targets based on their relevance in the pathogenesis of obesity and on a differential BMAL1 binding at their loci in omental fat from obese vs. non-obese subjects. For each target, we measured mRNA and protein levels in omental adipocyte precursors (OAPs). We found that many differential BMAL1 targets coding for factors involved in metabolic inflammation were actually dysregulated at the mRNA and protein levels in obesity and could therefore contribute to the Metabolic Syndrome (Results, page 7, lines 255-282, “Among the regions with differential BMAL1 binding between non-obese and obese OAPs ... levels in obese OAPs”; Fig. 6 has been edited and we added the new Fig. 7).
- – We found the same phenomenon in mice. We thus showed that besides *Ccl2*, some targets coding metabolic factors (i.e., extracellular matrix remodeling components) were also dysregulated in HFD- induced obese mice in association with clock dysfunction (Results, page 8, lines 320-326, “when *Ccl2* expression was still unchanged ... suggesting that it might initiate these alterations”, new panel Fig 9.I). This metabolic inflammation was prevented in mice with conditional inactivation of *Ikk β* restricted to adipocytes (Results, page 9, lines 350-352, “Expression of genes encoding extracellular matrix remodeling components was also significantly improved in AT from treated mice” and new panel in Fig.S6L-N).
- – Accordingly, the discussion is less focused on *CCL2*, but is aimed at integrating metabolic inflammation/adipose tissue remodeling with BMAL1 relocalization in obesity (Discussion, pages 9-11, lines 356-425). Consequently, the model presented in Fig. 4H has also been redrawn (legend pages 14-15, lines 534-545) and the abstract fully rewritten. We provided a new title “Circadian clock dysfunction in human omental fat links obesity to metabolic inflammation”.

I believe that the refocus of the manuscript is appropriate and warrants the inclusion of the new genome wide profiling data.

Specific points

There are also some technical issues that require clarification or different way of presentation. Below are some specific points:

1. In contrast to the statement, data presented in Fig.2D and 2E suggest that BMAL1 and RNA POLII binding to the promoters in OPAs remain rhythmic but is reduced in the amplitude. This will be consistent with luciferase imaging (Fig.1H) demonstrating rhythmicity in cultured OPAs isolated from obese patient.

Fig. 2D and 2E. When using two-way ANOVA followed by post hoc Sidak’s test, we failed to observe any significant differences in BMAL1 (or RNA POLII) binding to *PER2* in obese cells between the 2 collection times (27hr and 38hr post-synchronization), while a significant effect was observed in non-obese OPAs. Yet, we cannot exclude the possibility that obese cells maintain some rhythmicity but of reduced amplitude. We thus rephrased the relative sentences in Results and indicated that we did not disclose a significant rhythm of BMAL1 (and RNA POLII) binding to *PER2* in obese cells (Results, page 4, lines 151 and 154).

The authors have now made it clear that the binding of BMAL1 (and RNA POLII) to *Per2* is not statistically significant between the two time points measured (Figure 2D and E), however they have not addressed how this might relate to the lengthened period observed in obese OPAs (Figure 1H) – could the authors comment on this in the text?

2. Data presented in Fig.3C states that pharmacological inhibition of NF- κ B shortens the period of oscillations in cells isolated from obese patients, which is particularly effective after JSH-23 treatment (Fig. 3E). Which interval was used to calculate period? On the recording both look the same except for initial several hours that may result from drug treatment.

In the experiments with in vitro pharmacological inhibition of NF- κ B, data recorded from the first 12h were excluded from the analysis. As already indicated in Methods, bioluminescence was recorded continuously for 4–7 days. For this specific experiment, the recording lasted 4 days (i.e., the period was determined using the data from day 0.5 to day 4). Results indicate a clear effect of

NF- κ B Inhibition.

The period shortening is only 20 mins and 1h in these experiments, and these effects are difficult to see in bioluminescence traces (Figure 3C). The quantification in figure 3D does clarify this. To satisfy the reviewer's comments could the authors include in the methods section how many cycles were used to calculate the period in MetaCycle?

3. All assays for measuring NF- κ B activity was done using ELISA assay. It would be important to show Western blot data using phosphor-specific antibodies as this is the most representative way to support the statement. There is one Western blot data in Suppl. Figures; however, it is for total protein only; no change in total protein and changes in phosphor-protein must be shown side by side.

We complied with the referee suggestion. We now show a Western Blot with phosphorylated p65 protein and total p65 levels in obese vs. non-obese subjects (page 20, lines 786-788, novel panel E in Fig. S8). As expected, there was a huge increase in phosphorylated p65 levels in obesity, while total p65 did not change. These data are consistent with those obtained by Phospho-NF- κ Bp65 ELISA.

Supplementary Figure 8E is a great addition to the paper, is clearly presented and supports the narrative.

4. All major conclusions regarding clock dysfunction in obesity are based on transcriptional data; however, to claim physiological significance, they have to be supported by protein analysis, which is absent.

The criticism of the referee is fully relevant. We now focus on a panel of BMAL1 targets, known to be involved in the pathogenesis of obesity, that were differentially bound by BMAL1 in obese vs. non-obese subjects. For each target, we measured mRNA and protein levels in OAPs. We found that many differential BMAL1 targets coding for factors involved in metabolic inflammation were actually dysregulated at the mRNA and protein levels in obesity and could therefore contribute to the Metabolic Syndrome (Results, page 7, lines 276-282, "Next, we measured target gene expression and protein levels in our whole cohort of patients (...) transcript and protein levels in obese OAPs").

The authors have now used ELISAs to quantify levels of PAI1, MLL1 and GPX1 in human OAPs. The changes in protein levels in response to obesity support the findings from QPCR around transcript levels and I believe address the reviewer's comment.

5. Model presenter in Fig.4H is confusing as there is no direct evidence that repression of PER2 promoter directly affects CCL2 secretion rhythm. Moreover, the clock in obese tissue is not broken as cells remain rhythmic with a different period—

We did not actually provide any evidence that repression of PER2 promoter directly affects CCL2 rhythm. We modified the cartoon to avoid any confusion and also updated the model based on new data. The legend was adapted accordingly (pages 14-15, lines 534-545). Indeed, the cartoon of the clock is aimed at indicating that clock function is altered in cells from obese patients (e.g., the period is lengthened). We now indicate "circadian desynchrony" on top of the clock.

I like the schematic and consider it to be very clear. However, I question the use of the term "circadian desynchrony". To me this would suggest that this tissue is out of phase with surrounding clocks/time cues – which I don't believe you have shown. The data has shown period lengthening in adipose tissue from obese patients - why not state "period lengthening" in the figure?

6. Fig. 5 and 6 show very modest and not statistically significant effect of obesity on BMAL1 binding to PER2 (Fig. 5C) and dramatic effect on promoters of other genes (Fig.6). The natural question arises: why the entire work is focused on PER2 and not on other transcriptional targets?

It is well established that PER2 is one of the most important factors for setting the period of the circadian clock (Chen et al., *Mol Cell.*, 2009; Wilkins, et al., *PLoS Comput Biol.*, 2007). In addition, tools available to monitor PER2 transcription have been extensively described and herein, we monitored its period in OAPs as a readout for clock function. Although we acknowledge that ChIP-Seq experiments, performed in a small number of patients, showed a very modest and non-significant effect of obesity on BMAL1 binding to PER2 promoter (Fig. 5C-D), conventional ChIP showed a striking decreased binding (~7-fold) in obese vs non-obese subjects (Fig 2B). Decrease in PER2 mRNAs (Fig 1G) and period lengthening of Per2 oscillations further support an effect of obesity on the clock repressor (Fig. 1H, I).

However, ChIP-Seq experiments enlightened a dramatic effect of obesity on changes in binding enrichment of BMAL1 at other loci (see fig. 6). We therefore did not focus on PER2 only, but rather expanded our focus to new BMAL1 targets, known to be involved in the pathogenesis of obesity and metabolic syndrome. We found that many genes with differential BMAL1 binding between non-obese and obese OAPs code for factors involved in metabolic inflammation were actually dysregulated at the mRNA and protein levels in obesity and could therefore contribute to the Metabolic Syndrome (new Fig. 7). (Results, page 7, lines 255-282).

The authors have responded in full to this comment and I believe addressed this concern in full.

7. Data demonstrating no change in Ccl2 expression at the time of HFD-induced period lengthening must be shown.

Has been done. These data are now added as a new panel in Fig.9I.

This point has been addressed in full

8. Does sodium salicylate have an effect on cycling in tissue culture?

Salicylate treatment given in vivo for 1 month to mice simultaneously rendered obese by a HFD did not significantly change PER2::LUC period length in tissue explants cultured ex vivo (not shown). However, it should be emphasized that such salicylate treatment was of short duration and that period lengthening of PER2 oscillations increased with HFD duration (higher after 3 months than 1 month of HFD; Compare Fig 9G vs. 9H). We therefore cannot exclude the possibility that the experimental conditions were not optimal to address this issue.

In order to address the reviewer's point, could in vitro experiments be performed to ensure no acute effects of salicylate on period length? Alternatively, could the authors state in the text that they do not know the effects of salicylate on clock function in healthy tissue?

9. Fig. 8J – no statistically significant differences, therefore the authors cannot state that the downregulation of Per2 expression was corrected.

The referee is right. We now re-write more carefully this sentence "the downregulation of Per2 expression was attenuated .." (now Fig.9K) (Results, page 9, lines 338-339).

This fully addresses the reviewer's comment, thanks.

10. In real-time PCR data, how the expression was normalized? It says "GAPDH (???)", 18S or Gapdh. Were Gapdh levels affected by obesity? It is important to add this info to all mRNA quantitation data (i.e. Per2/Gapdh relative abundance).

As required, we added this information. Now, in the Method Section, we indicate that the expression of the studied gene is normalized to the reference gene: "normalized to GAPDH for human adipose tissue, adipocytes and OAPs, 18s for mouse adipose tissue and adipocytes and Gapdh for mouse APs" (page 21, lines 804-805). No more place to indicate this in the figure legends (limited to 350 words).

Whenever used, expression of the reference gene was not impacted by the conditions (no significant differences found between non-obese and obese patients, subcutaneous and omental

fat, collection-time, and no significant effect of the treatments).

The authors have addressed this comment in full.

Overall, the manuscript will benefit if after editing it will be submitted to a more specialized journal

We feel that we have suitably addressed all the criticisms raised by the referees. Our data are novel and the main message of this paper is of interest for a broad readership.

I believe the manuscript is of interest to a broad readership with interests including circadian biology, metabolism and inflammation.

REVIEWERS' COMMENTS

Reviewer #1 (Remarks to the Author):

The authors answered to the concerns and added new experiments. The new data are now in agreement with an impact of circadian clock dysfunction on metabolism and tissue remodelling rather than strictly inflammation. Title, abstract and discussion have been changed and clarity of the manuscript improved.

Reviewer #3 (Remarks to the Author):

This revised article includes the required new experiments, additional information and additional comments that provide further supports to the results. The described results are of great interest and the article is therefore a good candidate for publication.

Reviewer #4 (Remarks to the Author):

Please find below my comments to the authors in response to them addressing Reviewer 2's prior critique:

General comments

The revised version of the manuscript was significantly improved in respect to its structure and clarity of presentation of the data. Impressive amount of information have been added (such as genome-wide identification of BMAL1 binding sites in adipose tissue).

1. However it did not solve the main point of critique expressed by reviewers, it did not add exciting novelty of the findings except for confirmation of previously published results in mouse models of obesity using human adipose tissues.

Reviewer 2's main comment regarding "a lack of novelty" implies a comparison with the study from Hong et al., *Genes Dev.*, 2018. However, in the present manuscript, the issue is different: we now address the function of omental adipose tissue and its role in the development of cardiometabolic disease, which cannot be compared to those of liver. Thus, the message in the present article is original.

The article from Hong et al., 2018 focuses on BMAL1 cistrome in mouse liver. As BMAL1 predominantly binds DNA in a tissue-specific manner (Beytebiere et al, *Genes Dev.*, 2019), we could not even infer that similar data would have been obtained in mouse fat tissue. Moreover, some data obtained in mouse may not be translated to humans (here, we studied human omental fat biopsies). In our paper, BMAL1 targets were actually not the same as the ones identified in mouse liver. Pathways identified in mouse liver included nutrient-responsive anabolic pathways such as lipogenesis and lipid metabolism, or cholesterol biosynthesis. In this new article, we highlight targets encoding factors involved in endocytosis, proteoglycans, FOXO signaling, RNA metabolism or response to hypoxia. Specifically, we focused on the involvement of proteins involved in adipocyte vesicular trafficking, extracellular matrix components, or key enzymes in metabolic processes, chromatin organization and oxidative stress. The altered production of these proteins may be involved metabolic inflammation and unhealthy adipose remodeling leading to the metabolic syndrome.

I believe the latest version of the manuscript presents novel data which adds to the field. I cannot

be sure that Reviewer 2 was comparing this work with Hong et al. but agree with the authors that this manuscript is significantly diverse from the former.

2. Moreover, adding results of genome-wide profiling dilutes the focus of the manuscript.

The referee is right; this criticism has also been raised by referee 1 (see our answer 1b to referee 1). We now re-focus the manuscript on BMAL1 relocalization promoting metabolic inflammation in obesity rather than simply inflammation. We add some new experiments in order to develop this point:

- – Thus, we selected several BMAL1 targets based on their relevance in the pathogenesis of obesity and on a differential BMAL1 binding at their loci in omental fat from obese vs. non-obese subjects. For each target, we measured mRNA and protein levels in omental adipocyte precursors (OAPs). We found that many differential BMAL1 targets coding for factors involved in metabolic inflammation were actually dysregulated at the mRNA and protein levels in obesity and could therefore contribute to the Metabolic Syndrome (Results, page 7, lines 255-282, “Among the regions with differential BMAL1 binding between non-obese and obese OAPs ... levels in obese OAPs”; Fig. 6 has been edited and we added the new Fig. 7).

- – We found the same phenomenon in mice. We thus showed that besides *Ccl2*, some targets coding metabolic factors (i.e., extracellular matrix remodeling components) were also dysregulated in HFD- induced obese mice in association with clock dysfunction (Results, page 8, lines 320-326, “when *Ccl2* expression was still unchanged ... suggesting that it might initiate these alterations”, new panel Fig 9.I). This metabolic inflammation was prevented in mice with conditional inactivation of *Ikkβ* restricted to adipocytes (Results, page 9, lines 350-352, “Expression of genes encoding extracellular matrix remodeling components was also significantly improved in AT from treated mice” and new panel in Fig.S6L-N).

- – Accordingly, the discussion is less focused on *CCL2*, but is aimed at integrating metabolic inflammation/adipose tissue remodeling with BMAL1 relocalization in obesity (Discussion, pages 9-11, lines 356-425). Consequently, the model presented in Fig. 4H has also been redrawn (legend pages 14-15, lines 534-545) and the abstract fully rewritten.

We provided a new title “Circadian clock dysfunction in human omental fat links obesity to metabolic inflammation”.

I believe that the refocus of the manuscript is appropriate and warrants the inclusion of the new genome wide profiling data.

Specific points

There are also some technical issues that require clarification or different way of presentation. Below are some specific points:

1. In contrast to the statement, data presented in Fig.2D and 2E suggest that BMAL1 and RNA POLII binding to the promoters in OPAs remain rhythmic but is reduced in the amplitude. This will be consistent with luciferase imaging (Fig.1H) demonstrating rhythmicity in cultured OPAs isolated from obese patient.

Fig. 2D and 2E. When using two-way ANOVA followed by post hoc Sidak’s test, we failed to observe any significant differences in BMAL1 (or RNA POLII) binding to *PER2* in obese cells between the 2 collection times (27hr and 38hr post-synchronization), while a significant effect was observed in non-obese OPAs. Yet, we cannot exclude the possibility that obese cells maintain some rhythmicity but of reduced amplitude. We thus rephrased the relative sentences in Results and indicated that we did not disclose a significant rhythm of BMAL1 (and RNA POLII) binding to

PER2 in obese cells (Results, page 4, lines 151 and 154).

The authors have now made it clear that the binding of BMAL1 (and RNA POLII) to Per2 is not statistically significant between the two time points measured (Figure 2D and E), however they have not addressed how this might relate to the lengthened period observed in obese OAPs (Figure 1H) – could the authors comment on this in the text?

2. Data presented in Fig.3C states that pharmacological inhibition of NF-κB shortens the period of oscillations in cells isolated from obese patients, which is particularly effective after JSH-23 treatment (Fig. 3E). Which interval was used to calculate period? On the recording both look the same except for initial several hours that may result from drug treatment.

In the experiments with in vitro pharmacological inhibition of NF-κB, data recorded from the first 12h were excluded from the analysis. As already indicated in Methods, bioluminescence was recorded continuously for 4–7 days. For this specific experiment, the recording lasted 4 days (i.e., the period was determined using the data from day 0.5 to day 4). Results indicate a clear effect of NF-κB Inhibition.

The period shortening is only 20 mins and 1h in these experiments, and these effects are difficult to see in bioluminescence traces (Figure 3C). The quantification in figure 3D does clarify this. To satisfy the reviewer's comments could the authors include in the methods section how many cycles were used to calculate the period in MetaCycle?

3. All assays for measuring NF-κB activity was done using ELISA assay. It would be important to show Western blot data using phosphor-specific antibodies as this is the most representative way to support the statement. There is one Western blot data in Suppl. Figures; however, it is for total protein only; no change in total protein and changes in phosphor-protein must be shown side by side.

We complied with the referee suggestion. We now show a Western Blot with phosphorylated p65 protein and total p65 levels in obese vs. non-obese subjects (page 20, lines 786-788, novel panel E in Fig. S8). As expected, there was a huge increase in phosphorylated p65 levels in obesity, while total p65 did not change. These data are consistent with those obtained by Phospho-NF-κBp65 ELISA.

Supplementary Figure 8E is a great addition to the paper, is clearly presented and supports the narrative.

4. All major conclusions regarding clock dysfunction in obesity are based on transcriptional data; however, to claim physiological significance, they have to be supported by protein analysis, which is absent.

The criticism of the referee is fully relevant. We now focus on a panel of BMAL1 targets, known to be involved in the pathogenesis of obesity, that were differentially bound by BMAL1 in obese vs. non-obese subjects. For each target, we measured mRNA and protein levels in OAPs. We found that many differential BMAL1 targets coding for factors involved in metabolic inflammation were actually dysregulated at the mRNA and protein levels in obesity and could therefore contribute to the Metabolic Syndrome (Results, page 7, lines 276-282, "Next, we measured target gene expression and protein levels in our whole cohort of patients (...) transcript and protein levels in obese OAPs").

The authors have now used ELISAs to quantify levels of PAI1, MLL1 and GPX1 in human OAPs. The changes in protein levels in response to obesity support the findings from QPCR around transcript levels and I believe address the reviewer's comment.

5. Model presenter in Fig.4H is confusing as there is no direct evidence that repression of PER2 promoter directly affects CCL2 secretion rhythm. Moreover, the clock in obese tissue is not broken as cells remain rhythmic with a different period—

We did not actually provide any evidence that repression of PER2 promoter directly affects CCL2 rhythm. We modified the cartoon to avoid any confusion and also updated the model based on new data. The legend was adapted accordingly (pages 14-15, lines 534-545).

Indeed, the cartoon of the clock is aimed at indicating that clock function is altered in cells from obese patients (e.g., the period is lengthened). We now indicate "circadian desynchrony" on top of the clock.

I like the schematic and consider it to be very clear. However, I question the use of the term "circadian desynchrony". To me this would suggest that this tissue is out of phase with surrounding clocks/time cues – which I don't believe you have shown. The data has shown period lengthening in adipose tissue from obese patients - why not state "period lengthening" in the figure?

6. Fig. 5 and 6 show very modest and not statistically significant effect of obesity on BMAL1 binding to PER2 (Fig. 5C) and dramatic effect on promoters of other genes (Fig.6). The natural question arises: why the entire work is focused on PER2 and not on other transcriptional targets?

It is well established that PER2 is one of the most important factors for setting the period of the circadian clock (Chen et al., Mol Cell., 2009; Wilkins, et al., PLoS Comput Biol., 2007). In addition, tools available to monitor PER2 transcription have been extensively described and herein, we monitored its period in OAPs as a readout for clock function. Although we acknowledge that ChIP-Seq experiments, performed in a small number of patients, showed a very modest and non-significant effect of obesity on BMAL1 binding to PER2 promoter (Fig. 5C-D), conventional ChIP showed a striking decreased binding (~7-fold) in obese vs non-obese subjects (Fig 2B). Decrease in PER2 mRNAs (Fig 1G) and period lengthening of Per2 oscillations further support an effect of obesity on the clock repressor (Fig. 1H, I).

However, ChIP-Seq experiments enlightened a dramatic effect of obesity on changes in binding enrichment of BMAL1 at other loci (see fig. 6). We therefore did not focus on PER2 only, but rather expanded our focus to new BMAL1 targets, known to be involved in the pathogenesis of obesity and metabolic syndrome. We found that many genes with differential BMAL1 binding between non-obese and obese OAPs code for factors involved in metabolic inflammation were actually dysregulated at the mRNA and protein levels in obesity and could therefore contribute to the Metabolic Syndrome (new Fig. 7). (Results, page 7, lines 255-282).

The authors have responded in full to this comment and I believe addressed this concern in full.

7. Data demonstrating no change in Ccl2 expression at the time of HFD-induced period lengthening must be shown.

Has been done. These data are now added as a new panel in Fig.9I.

This point has been addressed in full

8. Does sodium salicylate have an effect on cycling in tissue culture?

Salicylate treatment given in vivo for 1 month to mice simultaneously rendered obese by a HFD did not significantly change PER2::LUC period length in tissue explants cultured ex vivo (not shown). However, it should be emphasized that such salicylate treatment was of short duration and that period lengthening of PER2 oscillations increased with HFD duration (higher after 3 months than 1 month of HFD; Compare Fig 9G vs. 9H. We therefore cannot exclude the possibility that the experimental conditions were not optimal to address this issue.

In order to address the reviewer's point, could in vitro experiments be performed to ensure no acute effects of salicylate on period length? Alternatively, could the authors state in the text that they do not know the effects of salicylate on clock function in healthy tissue?

9. Fig. 8J – no statistically significant differences, therefore the authors cannot state that the downregulation of Per2 expression was corrected.

The referee is right. We now re-write more carefully this sentence “the downregulation of Per2 expression was attenuated ..” (now Fig.9K) (Results, page 9, lines 338-339).

This fully addresses the reviewer's comment, thanks.

10. In real-time PCR data, how the expression was normalized? It says “GAPDH (???)”, 18S or Gapdh. Were Gapdh levels affected by obesity? It is important to add this info to all mRNA quantitation data (i.e. Per2/Gapdh relative abundance).

As required, we added this information. Now, in the Method Section, we indicate that the expression of the studied gene is normalized to the reference gene: “normalized to GAPDH for human adipose tissue, adipocytes and OAPs, 18s for mouse adipose tissue and adipocytes and Gapdh for mouse APs” (page 21, lines 804-805). No more place to indicate this in the figure legends (limited to 350 words).

Whenever used, expression of the reference gene was not impacted by the conditions (no significant differences found between non-obese and obese patients, subcutaneous and omental fat, collection-time, and no significant effect of the treatments).

The authors have addressed this comment in full.

Overall, the manuscript will benefit if after editing it will be submitted to a more specialized journal

We feel that we have suitably addressed all the criticisms raised by the referees. Our data are novel and the main message of this paper is of interest for a broad readership.

I believe the manuscript is of interest to a broad readership with interests including circadian biology, metabolism and inflammation.

Manuscript NCOMMS-19-09284C

Answers to the reviewers

We have carefully reviewed the minor comments and have revised the manuscript accordingly. The manuscript with tracked changes (red) has been uploaded. Our responses are given in a point-by-point manner below.

REVIEWERS' COMMENTS

Reviewer #1 (Remarks to the Author):

The authors answered to the concerns and added new experiments. The new data are now in agreement with an impact of circadian clock dysfunction on metabolism and tissue remodelling rather than strictly inflammation. Title, abstract and discussion have been changed and clarity of the manuscript improved.

No comments.

Reviewer #3 (Remarks to the Author):

This revised article includes the required new experiments, additional information and additional comments that provide further supports to the results. The described results are of great interest and the article is therefore a good candidate for publication.

No comments.

Reviewer #4 (Remarks to the Author):

Please find below my comments to the authors in response to them addressing Reviewer 2's prior critique:

General comments

The revised version of the manuscript was significantly improved in respect to its structure and clarity of presentation of the data. Impressive amount of information have been added (such as genome-wide identification of BMAL1 binding sites in adipose tissue).

1. However it did not solve the main point of critique expressed by reviewers, it did not add exciting novelty of the findings except for confirmation of previously published results in mouse models of obesity using human adipose tissues.

Reviewer 2's main comment regarding "a lack of novelty" implies a comparison with the study from Hong et al., Genes Dev., 2018. However, in the present manuscript, the issue is different: we now address the function of omental adipose tissue and its role in the development of

cardiometabolic disease, which cannot be compared to those of liver. Thus, the message in the present article is original.

The article from Hong et al., 2018 focuses on BMAL1 cistrome in mouse liver. As BMAL1 predominantly binds DNA in a tissue-specific manner (Beytebiere et al, Genes Dev., 2019), we could not even infer that similar data would have been obtained in mouse fat tissue. Moreover, some data obtained in mouse may not be translated to humans (here, we studied human omental fat biopsies). In our paper, BMAL1 targets were actually not the same as the ones identified in mouse liver. Pathways identified in mouse liver included nutrient-responsive anabolic pathways such as lipogenesis and lipid metabolism, or cholesterol biosynthesis. In this new article, we highlight targets encoding factors involved in endocytosis, proteoglycans, FOXO signaling, RNA metabolism or response to hypoxia. Specifically, we focused on the involvement of proteins involved in adipocyte vesicular trafficking, extracellular matrix components, or key enzymes in metabolic processes, chromatin organization and oxidative stress. The altered production of these proteins may be involved metabolic inflammation and unhealthy adipose remodeling leading to the metabolic syndrome.

I believe the latest version of the manuscript presents novel data which adds to the field. I cannot be sure that Reviewer 2 was comparing this work with Hong et al. but agree with the authors that this manuscript is significantly diverse from the former.

No comments.

2. Moreover, adding results of genome-wide profiling dilutes the focus of the manuscript.

The referee is right; this criticism has also been raised by referee 1 (see our answer 1b to referee 1). We now re-focus the manuscript on BMAL1 relocalization promoting metabolic inflammation in obesity rather than simply inflammation. We add some new experiments in order to develop this point:

- – Thus, we selected several BMAL1 targets based on their relevance in the pathogenesis of obesity and on a differential BMAL1 binding at their loci in omental fat from obese vs. non-obese subjects. For each target, we measured mRNA and protein levels in omental adipocyte precursors (OAPs). We found that many differential BMAL1 targets coding for factors involved in metabolic inflammation were actually dysregulated at the mRNA and protein levels in obesity and could therefore contribute to the Metabolic Syndrome (Results, page 7, lines 255-282, “Among the regions with differential BMAL1 binding between non-obese and obese OAPs ... levels in obese OAPs”; Fig. 6 has been edited and we added the new Fig. 7).

- – We found the same phenomenon in mice. We thus showed that besides Ccl2, some targets coding metabolic factors (i.e., extracellular matrix remodeling components) were also dysregulated in HFD- induced obese mice in association with clock dysfunction (Results, page 8, lines 320-326, “when Ccl2 expression was still unchanged ... suggesting that it might initiate these alterations”, new panel Fig 9.I). This metabolic inflammation was prevented in mice with conditional inactivation of Ikk β restricted to adipocytes (Results, page 9, lines 350-352, “Expression of genes encoding extracellular matrix remodeling components was also significantly improved in AT from treated mice” and new panel in Fig.S6L-N).

- – Accordingly, the discussion is less focused on CCL2, but is aimed at integrating metabolic inflammation/adipose tissue remodeling with BMAL1 relocalization in obesity (Discussion, pages 9-11, lines 356-425). Consequently, the model presented in Fig. 4H has also been redrawn (legend pages 14-15, lines 534-545) and the abstract fully rewritten.

We provided a new title “Circadian clock dysfunction in human omental fat links obesity to metabolic inflammation”.

I believe that the refocus of the manuscript is appropriate and warrants the inclusion of the new genome wide profiling data.

No comments.

Specific points

There are also some technical issues that require clarification or different way of presentation. Below are some specific points:

1. In contrast to the statement, data presented in Fig.2D and 2E suggest that BMAL1 and RNA POLII binding to the promoters in OPAs remain rhythmic but is reduced in the amplitude. This will be consistent with luciferase imaging (Fig.1H) demonstrating rhythmicity in cultured OPAs isolated from obese patient.

Fig. 2D and 2E. When using two-way ANOVA followed by post hoc Sidak's test, we failed to observe any significant differences in BMAL1 (or RNA POLII) binding to PER2 in obese cells between the 2 collection times (27hr and 38hr post-synchronization), while a significant effect was observed in non-obese OPAs. Yet, we cannot exclude the possibility that obese cells maintain some rhythmicity but of reduced amplitude. We thus rephrased the relative sentences in Results and indicated that we did not disclose a significant rhythm of BMAL1 (and RNA POLII) binding to PER2 in obese cells (Results, page 4, lines 151 and 154).

The authors have now made it clear that the binding of BMAL1 (and RNA POLII) to Per2 is not statistically significant between the two time points measured (Figure 2D and E), however they have not addressed how this might relate to the lengthened period observed in obese OAPs (Figure 1H) – could the authors comment on this in the text?

Done. To clarify, we added the following sentence (Results, page 4, lines 154-157): “The significantly different RNA POLII initiation in obese vs. non-obese OAPs, together with a different rhythm, suggests that the period of *PER2* transcription, its amplitude or both might be different between the two conditions. In agreement, this period was significantly lengthened in obesity (Fig. 1H-I).

2. Data presented in Fig.3C states that pharmacological inhibition of NF-κB shortens the period of oscillations in cells isolated from obese patients, which is particularly effective after JSH-23 treatment (Fig. 3E). Which interval was used to calculate period? On the recording both look the same except for initial several hours that may result from drug treatment.

In the experiments with in vitro pharmacological inhibition of NF-κB, data recorded from the first 12h were excluded from the analysis. As already indicated in Methods, bioluminescence was recorded continuously for 4–7 days. For this specific experiment, the recording lasted 4 days (i.e., the period was determined using the data from day 0.5 to day 4). Results indicate a clear effect of NF-κB Inhibition.

The period shortening is only 20 mins and 1h in these experiments, and these effects are difficult to see in bioluminescence traces (Figure 3C). The quantification in figure 3D does clarify this. To

satisfy the reviewer's comments could the authors include in the methods section how many cycles were used to calculate the period in MetaCycle?

Done. To clarify, we added the following sentence (Methods, page 21, lines 794-795): "The analyses were performed on 4 cycles or more, except for Fig 3D-E and 9F top where 3.6 cycles were used."

3. All assays for measuring NF-kB activity was done using ELISA assay. It would be important to show Western blot data using phosphor-specific antibodies as this is the most representative way to support the statement. There is one Western blot data in Suppl. Figures; however, it is for total protein only; no change in total protein and changes in phosphor-protein must be shown side by side.

We complied with the referee suggestion. We now show a Western Blot with phosphorylated p65 protein and total p65 levels in obese vs. non-obese subjects (page 20, lines 786-788, novel panel E in Fig. S8). As expected, there was a huge increase in phosphorylated p65 levels in obesity, while total p65 did not change. These data are consistent with those obtained by Phospho-NF-kBp65 ELISA.

Supplementary Figure 8E is a great addition to the paper, is clearly presented and supports the narrative.

No comments.

4. All major conclusions regarding clock dysfunction in obesity are based on transcriptional data; however, to claim physiological significance, they have to be supported by protein analysis, which is absent.

The criticism of the referee is fully relevant. We now focus on a panel of BMAL1 targets, known to be involved in the pathogenesis of obesity, that were differentially bound by BMAL1 in obese vs. non-obese subjects. For each target, we measured mRNA and protein levels in OAPs. We found that many differential BMAL1 targets coding for factors involved in metabolic inflammation were actually dysregulated at the mRNA and protein levels in obesity and could therefore contribute to the Metabolic Syndrome (Results, page 7, lines 276-282, "Next, we measured target gene expression and protein levels in our whole cohort of patients (...) transcript and protein levels in obese OAPs").

The authors have now used ELISAs to quantify levels of PAI1, MLL1 and GPX1 in human OAPs. The changes in protein levels in response to obesity support the findings from QPCR around transcript levels and I believe address the reviewer's comment.

No comments.

5. Model presenter in Fig.4H is confusing as there is no direct evidence that repression of PER2 promoter directly affects CCL2 secretion rhythm. Moreover, the clock in obese tissue is not broken as cells remain rhythmic with a different period—

We did not actually provide any evidence that repression of PER2 promoter directly affects CCL2 rhythm. We modified the cartoon to avoid any confusion and also updated the model based on

new data. The legend was adapted accordingly (pages 14-15, lines 534-545).

Indeed, the cartoon of the clock is aimed at indicating that clock function is altered in cells from obese patients (e.g., the period is lengthened). We now indicate “circadian desynchrony” on top of the clock.

I like the schematic and consider it to be very clear. However, I question the use of the term “circadian desynchrony”. To me this would suggest that this tissue is out of phase with surrounding clocks/time cues – which I don’t believe you have shown. The data has shown period lengthening in adipose tissue from obese patients - why not state “period lengthening” in the figure?

Done. “Clock desynchrony” has been replaced by “circadian period lengthening” in Fig.4H. Also replaced in the legend (now “lengthening of the circadian period”, page 14, line 543).

6. Fig. 5 and 6 show very modest and not statistically significant effect of obesity on BMAL1 binding to PER2 (Fig. 5C) and dramatic effect on promoters of other genes (Fig.6). The natural question arises: why the entire work is focused on PER2 and not on other transcriptional targets?

It is well established that PER2 is one of the most important factors for setting the period of the circadian clock (Chen et al., Mol Cell., 2009; Wilkins, et al., PLoS Comput Biol., 2007). In addition, tools available to monitor PER2 transcription have been extensively described and herein, we monitored its period in OAPs as a readout for clock function. Although we acknowledge that ChIP-Seq experiments, performed in a small number of patients, showed a very modest and non-significant effect of obesity on BMAL1 binding to PER2 promoter (Fig. 5C-D), conventional ChIP showed a striking decreased binding (~7-fold) in obese vs non-obese subjects (Fig 2B). Decrease in PER2 mRNAs (Fig 1G) and period lengthening of Per2 oscillations further support an effect of obesity on the clock repressor (Fig. 1H, I).

However, ChIP-Seq experiments enlightened a dramatic effect of obesity on changes in binding enrichment of BMAL1 at other loci (see fig. 6). We therefore did not focus on PER2 only, but rather expanded our focus to new BMAL1 targets, known to be involved in the pathogenesis of obesity and metabolic syndrome. We found that many genes with differential BMAL1 binding between non-obese and obese OAPs code for factors involved in metabolic inflammation were actually dysregulated at the mRNA and protein levels in obesity and could therefore contribute to the Metabolic Syndrome (new Fig. 7). (Results, page 7, lines 255-282).

The authors have responded in full to this comment and I believe addressed this concern in full.

No comments.

7. Data demonstrating no change in Ccl2 expression at the time of HFD-induced period lengthening must be shown.

Has been done. These data are now added as a new panel in Fig.9I.

This point has been addressed in full

No comments.

8. Does sodium salicylate have an effect on cycling in tissue culture?

Salicylate treatment given in vivo for 1 month to mice simultaneously rendered obese by a HFD

did not significantly change PER2::LUC period length in tissue explants cultured ex vivo (not shown). However, it should be emphasized that such salicylate treatment was of short duration and that period lengthening of PER2 oscillations increased with HFD duration (higher after 3 months than 1 month of HFD; Compare Fig 9G vs. 9H. We therefore cannot exclude the possibility that the experimental conditions were not optimal to address this issue.

In order to address the reviewer's point, could in vitro experiments be performed to ensure no acute effects of salicylate on period length? Alternatively, could the authors state in the text that they do not know the effects of salicylate on clock function in healthy tissue?

Done. As required, we added the following sentence (Results, page 9 lines 340-341): "Yet, salicylate was not tested in fat from control mice."

9. Fig. 8J – no statistically significant differences, therefore the authors cannot state that the downregulation of Per2 expression was corrected.

The referee is right. We now re-write more carefully this sentence "the downregulation of Per2 expression was attenuated .." (now Fig.9K) (Results, page 9, lines 338-339).

This fully addresses the reviewer's comment, thanks.

No comments.

10. In real-time PCR data, how the expression was normalized? It says "GAPDH (???)", 18S or Gapdh. Were Gapdh levels affected by obesity? It is important to add this info to all mRNA quantitation data (i.e. Per2/Gapdh relative abundance).

As required, we added this information. Now, in the Method Section, we indicate that the expression of the studied gene is normalized to the reference gene: "normalized to GAPDH for human adipose tissue, adipocytes and OAPs, 18s for mouse adipose tissue and adipocytes and Gapdh for mouse APs" (page 21, lines 804-805). No more place to indicate this in the figure legends (limited to 350 words).

Whenever used, expression of the reference gene was not impacted by the conditions (no significant differences found between non-obese and obese patients, subcutaneous and omental fat, collection-time, and no significant effect of the treatments).

The authors have addressed this comment in full.

No comments.

Overall, the manuscript will benefit if after editing it will be submitted to a more specialized journal

We feel that we have suitably addressed all the criticisms raised by the referees. Our data are novel and the main message of this paper is of interest for a broad readership.

I believe the manuscript is of interest to a broad readership with interests including circadian biology, metabolism and inflammation.

No comments.

REVIEWERS' COMMENTS

Reviewer #1 (Remarks to the Author):

The authors answered to the concerns and added new experiments. The new data are now in agreement with an impact of circadian clock dysfunction on metabolism and tissue remodelling rather than strictly inflammation. Title, abstract and discussion have been changed and clarity of the manuscript improved.

Reviewer #3 (Remarks to the Author):

This revised article includes the required new experiments, additional information and additional comments that provide further supports to the results. The described results are of great interest and the article is therefore a good candidate for publication.

Reviewer #4 (Remarks to the Author):

Please find below my comments to the authors in response to them addressing Reviewer 2's prior critique:

General comments

The revised version of the manuscript was significantly improved in respect to its structure and clarity of presentation of the data. Impressive amount of information have been added (such as genome-wide identification of BMAL1 binding sites in adipose tissue).

1. However it did not solve the main point of critique expressed by reviewers, it did not add exciting novelty of the findings except for confirmation of previously published results in mouse models of obesity using human adipose tissues.

Reviewer 2's main comment regarding "a lack of novelty" implies a comparison with the study from Hong et al., *Genes Dev.*, 2018. However, in the present manuscript, the issue is different: we now address the function of omental adipose tissue and its role in the development of cardiometabolic disease, which cannot be compared to those of liver. Thus, the message in the present article is original.

The article from Hong et al., 2018 focuses on BMAL1 cistrome in mouse liver. As BMAL1 predominantly binds DNA in a tissue-specific manner (Beytebiere et al, *Genes Dev.*, 2019), we could not even infer that similar data would have been obtained in mouse fat tissue. Moreover, some data obtained in mouse may not be translated to humans (here, we studied human omental fat biopsies). In our paper, BMAL1 targets were actually not the same as the ones identified in mouse liver. Pathways identified in mouse liver included nutrient-responsive anabolic pathways such as lipogenesis and lipid metabolism, or cholesterol biosynthesis. In this new article, we highlight targets encoding factors involved in endocytosis, proteoglycans, FOXO signaling, RNA metabolism or response to hypoxia. Specifically, we focused on the involvement of proteins involved in adipocyte vesicular trafficking, extracellular matrix components, or key enzymes in metabolic processes, chromatin organization and oxidative stress. The altered production of these proteins may be involved metabolic inflammation and unhealthy adipose remodeling leading to the metabolic syndrome.

I believe the latest version of the manuscript presents novel data which adds to the field. I cannot

be sure that Reviewer 2 was comparing this work with Hong et al. but agree with the authors that this manuscript is significantly diverse from the former.

2. Moreover, adding results of genome-wide profiling dilutes the focus of the manuscript.

The referee is right; this criticism has also been raised by referee 1 (see our answer 1b to referee 1). We now re-focus the manuscript on BMAL1 relocalization promoting metabolic inflammation in obesity rather than simply inflammation. We add some new experiments in order to develop this point:

- – Thus, we selected several BMAL1 targets based on their relevance in the pathogenesis of obesity and on a differential BMAL1 binding at their loci in omental fat from obese vs. non-obese subjects. For each target, we measured mRNA and protein levels in omental adipocyte precursors (OAPs). We found that many differential BMAL1 targets coding for factors involved in metabolic inflammation were actually dysregulated at the mRNA and protein levels in obesity and could therefore contribute to the Metabolic Syndrome (Results, page 7, lines 255-282, “Among the regions with differential BMAL1 binding between non-obese and obese OAPs ... levels in obese OAPs”; Fig. 6 has been edited and we added the new Fig. 7).

- – We found the same phenomenon in mice. We thus showed that besides *Ccl2*, some targets coding metabolic factors (i.e., extracellular matrix remodeling components) were also dysregulated in HFD- induced obese mice in association with clock dysfunction (Results, page 8, lines 320-326, “when *Ccl2* expression was still unchanged ... suggesting that it might initiate these alterations”, new panel Fig 9.I). This metabolic inflammation was prevented in mice with conditional inactivation of *Ikkβ* restricted to adipocytes (Results, page 9, lines 350-352, “Expression of genes encoding extracellular matrix remodeling components was also significantly improved in AT from treated mice” and new panel in Fig.S6L-N).

- – Accordingly, the discussion is less focused on *CCL2*, but is aimed at integrating metabolic inflammation/adipose tissue remodeling with BMAL1 relocalization in obesity (Discussion, pages 9-11, lines 356-425). Consequently, the model presented in Fig. 4H has also been redrawn (legend pages 14-15, lines 534-545) and the abstract fully rewritten.

We provided a new title “Circadian clock dysfunction in human omental fat links obesity to metabolic inflammation”.

I believe that the refocus of the manuscript is appropriate and warrants the inclusion of the new genome wide profiling data.

Specific points

There are also some technical issues that require clarification or different way of presentation. Below are some specific points:

1. In contrast to the statement, data presented in Fig.2D and 2E suggest that BMAL1 and RNA POLII binding to the promoters in OPAs remain rhythmic but is reduced in the amplitude. This will be consistent with luciferase imaging (Fig.1H) demonstrating rhythmicity in cultured OPAs isolated from obese patient.

Fig. 2D and 2E. When using two-way ANOVA followed by post hoc Sidak’s test, we failed to observe any significant differences in BMAL1 (or RNA POLII) binding to *PER2* in obese cells between the 2 collection times (27hr and 38hr post-synchronization), while a significant effect was observed in non-obese OPAs. Yet, we cannot exclude the possibility that obese cells maintain some rhythmicity but of reduced amplitude. We thus rephrased the relative sentences in Results and indicated that we did not disclose a significant rhythm of BMAL1 (and RNA POLII) binding to

PER2 in obese cells (Results, page 4, lines 151 and 154).

The authors have now made it clear that the binding of BMAL1 (and RNA POLII) to Per2 is not statistically significant between the two time points measured (Figure 2D and E), however they have not addressed how this might relate to the lengthened period observed in obese OAPs (Figure 1H) – could the authors comment on this in the text?

2. Data presented in Fig.3C states that pharmacological inhibition of NF-κB shortens the period of oscillations in cells isolated from obese patients, which is particularly effective after JSH-23 treatment (Fig. 3E). Which interval was used to calculate period? On the recording both look the same except for initial several hours that may result from drug treatment.

In the experiments with in vitro pharmacological inhibition of NF-κB, data recorded from the first 12h were excluded from the analysis. As already indicated in Methods, bioluminescence was recorded continuously for 4–7 days. For this specific experiment, the recording lasted 4 days (i.e., the period was determined using the data from day 0.5 to day 4). Results indicate a clear effect of NF-κB Inhibition.

The period shortening is only 20 mins and 1h in these experiments, and these effects are difficult to see in bioluminescence traces (Figure 3C). The quantification in figure 3D does clarify this. To satisfy the reviewer's comments could the authors include in the methods section how many cycles were used to calculate the period in MetaCycle?

3. All assays for measuring NF-κB activity was done using ELISA assay. It would be important to show Western blot data using phosphor-specific antibodies as this is the most representative way to support the statement. There is one Western blot data in Suppl. Figures; however, it is for total protein only; no change in total protein and changes in phosphor-protein must be shown side by side.

We complied with the referee suggestion. We now show a Western Blot with phosphorylated p65 protein and total p65 levels in obese vs. non-obese subjects (page 20, lines 786-788, novel panel E in Fig. S8). As expected, there was a huge increase in phosphorylated p65 levels in obesity, while total p65 did not change. These data are consistent with those obtained by Phospho-NF-κBp65 ELISA.

Supplementary Figure 8E is a great addition to the paper, is clearly presented and supports the narrative.

4. All major conclusions regarding clock dysfunction in obesity are based on transcriptional data; however, to claim physiological significance, they have to be supported by protein analysis, which is absent.

The criticism of the referee is fully relevant. We now focus on a panel of BMAL1 targets, known to be involved in the pathogenesis of obesity, that were differentially bound by BMAL1 in obese vs. non-obese subjects. For each target, we measured mRNA and protein levels in OAPs. We found that many differential BMAL1 targets coding for factors involved in metabolic inflammation were actually dysregulated at the mRNA and protein levels in obesity and could therefore contribute to the Metabolic Syndrome (Results, page 7, lines 276-282, "Next, we measured target gene expression and protein levels in our whole cohort of patients (...) transcript and protein levels in obese OAPs").

The authors have now used ELISAs to quantify levels of PAI1, MLL1 and GPX1 in human OAPs. The changes in protein levels in response to obesity support the findings from QPCR around transcript levels and I believe address the reviewer's comment.

5. Model presenter in Fig.4H is confusing as there is no direct evidence that repression of PER2 promoter directly affects CCL2 secretion rhythm. Moreover, the clock in obese tissue is not broken as cells remain rhythmic with a different period—

We did not actually provide any evidence that repression of PER2 promoter directly affects CCL2 rhythm. We modified the cartoon to avoid any confusion and also updated the model based on new data. The legend was adapted accordingly (pages 14-15, lines 534-545).

Indeed, the cartoon of the clock is aimed at indicating that clock function is altered in cells from obese patients (e.g., the period is lengthened). We now indicate "circadian desynchrony" on top of the clock.

I like the schematic and consider it to be very clear. However, I question the use of the term "circadian desynchrony". To me this would suggest that this tissue is out of phase with surrounding clocks/time cues – which I don't believe you have shown. The data has shown period lengthening in adipose tissue from obese patients - why not state "period lengthening" in the figure?

6. Fig. 5 and 6 show very modest and not statistically significant effect of obesity on BMAL1 binding to PER2 (Fig. 5C) and dramatic effect on promoters of other genes (Fig.6). The natural question arises: why the entire work is focused on PER2 and not on other transcriptional targets?

It is well established that PER2 is one of the most important factors for setting the period of the circadian clock (Chen et al., Mol Cell., 2009; Wilkins, et al., PLoS Comput Biol., 2007). In addition, tools available to monitor PER2 transcription have been extensively described and herein, we monitored its period in OAPs as a readout for clock function. Although we acknowledge that ChIP-Seq experiments, performed in a small number of patients, showed a very modest and non-significant effect of obesity on BMAL1 binding to PER2 promoter (Fig. 5C-D), conventional ChIP showed a striking decreased binding (~7-fold) in obese vs non-obese subjects (Fig 2B). Decrease in PER2 mRNAs (Fig 1G) and period lengthening of Per2 oscillations further support an effect of obesity on the clock repressor (Fig. 1H, I).

However, ChIP-Seq experiments enlightened a dramatic effect of obesity on changes in binding enrichment of BMAL1 at other loci (see fig. 6). We therefore did not focus on PER2 only, but rather expanded our focus to new BMAL1 targets, known to be involved in the pathogenesis of obesity and metabolic syndrome. We found that many genes with differential BMAL1 binding between non-obese and obese OAPs code for factors involved in metabolic inflammation were actually dysregulated at the mRNA and protein levels in obesity and could therefore contribute to the Metabolic Syndrome (new Fig. 7). (Results, page 7, lines 255-282).

The authors have responded in full to this comment and I believe addressed this concern in full.

7. Data demonstrating no change in Ccl2 expression at the time of HFD-induced period lengthening must be shown.

Has been done. These data are now added as a new panel in Fig.9I.

This point has been addressed in full

8. Does sodium salicylate have an effect on cycling in tissue culture?

Salicylate treatment given in vivo for 1 month to mice simultaneously rendered obese by a HFD did not significantly change PER2::LUC period length in tissue explants cultured ex vivo (not shown). However, it should be emphasized that such salicylate treatment was of short duration and that period lengthening of PER2 oscillations increased with HFD duration (higher after 3 months than 1 month of HFD; Compare Fig 9G vs. 9H. We therefore cannot exclude the possibility that the experimental conditions were not optimal to address this issue.

In order to address the reviewer's point, could in vitro experiments be performed to ensure no acute effects of salicylate on period length? Alternatively, could the authors state in the text that they do not know the effects of salicylate on clock function in healthy tissue?

9. Fig. 8J – no statistically significant differences, therefore the authors cannot state that the downregulation of Per2 expression was corrected.

The referee is right. We now re-write more carefully this sentence “the downregulation of Per2 expression was attenuated ..” (now Fig.9K) (Results, page 9, lines 338-339).

This fully addresses the reviewer's comment, thanks.

10. In real-time PCR data, how the expression was normalized? It says “GAPDH (???)”, 18S or Gapdh. Were Gapdh levels affected by obesity? It is important to add this info to all mRNA quantitation data (i.e. Per2/Gapdh relative abundance).

As required, we added this information. Now, in the Method Section, we indicate that the expression of the studied gene is normalized to the reference gene: “normalized to GAPDH for human adipose tissue, adipocytes and OAPs, 18s for mouse adipose tissue and adipocytes and Gapdh for mouse APs” (page 21, lines 804-805). No more place to indicate this in the figure legends (limited to 350 words).

Whenever used, expression of the reference gene was not impacted by the conditions (no significant differences found between non-obese and obese patients, subcutaneous and omental fat, collection-time, and no significant effect of the treatments).

The authors have addressed this comment in full.

Overall, the manuscript will benefit if after editing it will be submitted to a more specialized journal

We feel that we have suitably addressed all the criticisms raised by the referees. Our data are novel and the main message of this paper is of interest for a broad readership.

I believe the manuscript is of interest to a broad readership with interests including circadian biology, metabolism and inflammation.

Manuscript NCOMMS-19-09284C

Answers to the reviewers

We have carefully reviewed the minor comments and have revised the manuscript accordingly. The manuscript with tracked changes (red) has been uploaded. Our responses are given in a point-by-point manner below.

REVIEWERS' COMMENTS

Reviewer #1 (Remarks to the Author):

The authors answered to the concerns and added new experiments. The new data are now in agreement with an impact of circadian clock dysfunction on metabolism and tissue remodelling rather than strictly inflammation. Title, abstract and discussion have been changed and clarity of the manuscript improved.

No comments.

Reviewer #3 (Remarks to the Author):

This revised article includes the required new experiments, additional information and additional comments that provide further supports to the results. The described results are of great interest and the article is therefore a good candidate for publication.

No comments.

Reviewer #4 (Remarks to the Author):

Please find below my comments to the authors in response to them addressing Reviewer 2's prior critique:

General comments

The revised version of the manuscript was significantly improved in respect to its structure and clarity of presentation of the data. Impressive amount of information have been added (such as genome-wide identification of BMAL1 binding sites in adipose tissue).

1. However it did not solve the main point of critique expressed by reviewers, it did not add exciting novelty of the findings except for confirmation of previously published results in mouse models of obesity using human adipose tissues.

Reviewer 2's main comment regarding "a lack of novelty" implies a comparison with the study from Hong et al., Genes Dev., 2018. However, in the present manuscript, the issue is different: we now address the function of omental adipose tissue and its role in the development of

cardiometabolic disease, which cannot be compared to those of liver. Thus, the message in the present article is original.

The article from Hong et al., 2018 focuses on BMAL1 cistrome in mouse liver. As BMAL1 predominantly binds DNA in a tissue-specific manner (Beytebiere et al, Genes Dev., 2019), we could not even infer that similar data would have been obtained in mouse fat tissue. Moreover, some data obtained in mouse may not be translated to humans (here, we studied human omental fat biopsies). In our paper, BMAL1 targets were actually not the same as the ones identified in mouse liver. Pathways identified in mouse liver included nutrient-responsive anabolic pathways such as lipogenesis and lipid metabolism, or cholesterol biosynthesis. In this new article, we highlight targets encoding factors involved in endocytosis, proteoglycans, FOXO signaling, RNA metabolism or response to hypoxia. Specifically, we focused on the involvement of proteins involved in adipocyte vesicular trafficking, extracellular matrix components, or key enzymes in metabolic processes, chromatin organization and oxidative stress. The altered production of these proteins may be involved metabolic inflammation and unhealthy adipose remodeling leading to the metabolic syndrome.

I believe the latest version of the manuscript presents novel data which adds to the field. I cannot be sure that Reviewer 2 was comparing this work with Hong et al. but agree with the authors that this manuscript is significantly diverse from the former.

No comments.

2. Moreover, adding results of genome-wide profiling dilutes the focus of the manuscript.

The referee is right; this criticism has also been raised by referee 1 (see our answer 1b to referee 1). We now re-focus the manuscript on BMAL1 relocalization promoting metabolic inflammation in obesity rather than simply inflammation. We add some new experiments in order to develop this point:

- – Thus, we selected several BMAL1 targets based on their relevance in the pathogenesis of obesity and on a differential BMAL1 binding at their loci in omental fat from obese vs. non-obese subjects. For each target, we measured mRNA and protein levels in omental adipocyte precursors (OAPs). We found that many differential BMAL1 targets coding for factors involved in metabolic inflammation were actually dysregulated at the mRNA and protein levels in obesity and could therefore contribute to the Metabolic Syndrome (Results, page 7, lines 255-282, “Among the regions with differential BMAL1 binding between non-obese and obese OAPs ... levels in obese OAPs”; Fig. 6 has been edited and we added the new Fig. 7).

- – We found the same phenomenon in mice. We thus showed that besides Ccl2, some targets coding metabolic factors (i.e., extracellular matrix remodeling components) were also dysregulated in HFD- induced obese mice in association with clock dysfunction (Results, page 8, lines 320-326, “when Ccl2 expression was still unchanged ... suggesting that it might initiate these alterations”, new panel Fig 9.I). This metabolic inflammation was prevented in mice with conditional inactivation of Ikk β restricted to adipocytes (Results, page 9, lines 350-352, “Expression of genes encoding extracellular matrix remodeling components was also significantly improved in AT from treated mice” and new panel in Fig.S6L-N).

- – Accordingly, the discussion is less focused on CCL2, but is aimed at integrating metabolic inflammation/adipose tissue remodeling with BMAL1 relocalization in obesity (Discussion, pages 9-11, lines 356-425). Consequently, the model presented in Fig. 4H has also been redrawn (legend pages 14-15, lines 534-545) and the abstract fully rewritten.

We provided a new title “Circadian clock dysfunction in human omental fat links obesity to metabolic inflammation”.

I believe that the refocus of the manuscript is appropriate and warrants the inclusion of the new genome wide profiling data.

No comments.

Specific points

There are also some technical issues that require clarification or different way of presentation. Below are some specific points:

1. In contrast to the statement, data presented in Fig.2D and 2E suggest that BMAL1 and RNA POLII binding to the promoters in OPAs remain rhythmic but is reduced in the amplitude. This will be consistent with luciferase imaging (Fig.1H) demonstrating rhythmicity in cultured OPAs isolated from obese patient.

Fig. 2D and 2E. When using two-way ANOVA followed by post hoc Sidak's test, we failed to observe any significant differences in BMAL1 (or RNA POLII) binding to PER2 in obese cells between the 2 collection times (27hr and 38hr post-synchronization), while a significant effect was observed in non-obese OPAs. Yet, we cannot exclude the possibility that obese cells maintain some rhythmicity but of reduced amplitude. We thus rephrased the relative sentences in Results and indicated that we did not disclose a significant rhythm of BMAL1 (and RNA POLII) binding to PER2 in obese cells (Results, page 4, lines 151 and 154).

The authors have now made it clear that the binding of BMAL1 (and RNA POLII) to Per2 is not statistically significant between the two time points measured (Figure 2D and E), however they have not addressed how this might relate to the lengthened period observed in obese OAPs (Figure 1H) – could the authors comment on this in the text?

Done. To clarify, we added the following sentence (Results, page 4, lines 154-157): “The significantly different RNA POLII initiation in obese vs. non-obese OAPs, together with a different rhythm, suggests that the period of *PER2* transcription, its amplitude or both might be different between the two conditions. In agreement, this period was significantly lengthened in obesity (Fig. 1H-I).

2. Data presented in Fig.3C states that pharmacological inhibition of NF-κB shortens the period of oscillations in cells isolated from obese patients, which is particularly effective after JSH-23 treatment (Fig. 3E). Which interval was used to calculate period? On the recording both look the same except for initial several hours that may result from drug treatment.

In the experiments with in vitro pharmacological inhibition of NF-κB, data recorded from the first 12h were excluded from the analysis. As already indicated in Methods, bioluminescence was recorded continuously for 4–7 days. For this specific experiment, the recording lasted 4 days (i.e., the period was determined using the data from day 0.5 to day 4). Results indicate a clear effect of NF-κB Inhibition.

The period shortening is only 20 mins and 1h in these experiments, and these effects are difficult to see in bioluminescence traces (Figure 3C). The quantification in figure 3D does clarify this. To

satisfy the reviewer's comments could the authors include in the methods section how many cycles were used to calculate the period in MetaCycle?

Done. To clarify, we added the following sentence (Methods, page 21, lines 794-795): "The analyses were performed on 4 cycles or more, except for Fig 3D-E and 9F top where 3.6 cycles were used."

3. All assays for measuring NF-kB activity was done using ELISA assay. It would be important to show Western blot data using phosphor-specific antibodies as this is the most representative way to support the statement. There is one Western blot data in Suppl. Figures; however, it is for total protein only; no change in total protein and changes in phosphor-protein must be shown side by side.

We complied with the referee suggestion. We now show a Western Blot with phosphorylated p65 protein and total p65 levels in obese vs. non-obese subjects (page 20, lines 786-788, novel panel E in Fig. S8). As expected, there was a huge increase in phosphorylated p65 levels in obesity, while total p65 did not change. These data are consistent with those obtained by Phospho-NF-kBp65 ELISA.

Supplementary Figure 8E is a great addition to the paper, is clearly presented and supports the narrative.

No comments.

4. All major conclusions regarding clock dysfunction in obesity are based on transcriptional data; however, to claim physiological significance, they have to be supported by protein analysis, which is absent.

The criticism of the referee is fully relevant. We now focus on a panel of BMAL1 targets, known to be involved in the pathogenesis of obesity, that were differentially bound by BMAL1 in obese vs. non-obese subjects. For each target, we measured mRNA and protein levels in OAPs. We found that many differential BMAL1 targets coding for factors involved in metabolic inflammation were actually dysregulated at the mRNA and protein levels in obesity and could therefore contribute to the Metabolic Syndrome (Results, page 7, lines 276-282, "Next, we measured target gene expression and protein levels in our whole cohort of patients (...) transcript and protein levels in obese OAPs").

The authors have now used ELISAs to quantify levels of PAI1, MLL1 and GPX1 in human OAPs. The changes in protein levels in response to obesity support the findings from QPCR around transcript levels and I believe address the reviewer's comment.

No comments.

5. Model presenter in Fig.4H is confusing as there is no direct evidence that repression of PER2 promoter directly affects CCL2 secretion rhythm. Moreover, the clock in obese tissue is not broken as cells remain rhythmic with a different period—

We did not actually provide any evidence that repression of PER2 promoter directly affects CCL2 rhythm. We modified the cartoon to avoid any confusion and also updated the model based on

new data. The legend was adapted accordingly (pages 14-15, lines 534-545).

Indeed, the cartoon of the clock is aimed at indicating that clock function is altered in cells from obese patients (e.g., the period is lengthened). We now indicate “circadian desynchrony” on top of the clock.

I like the schematic and consider it to be very clear. However, I question the use of the term “circadian desynchrony”. To me this would suggest that this tissue is out of phase with surrounding clocks/time cues – which I don’t believe you have shown. The data has shown period lengthening in adipose tissue from obese patients - why not state “period lengthening” in the figure?

Done. “Clock desynchrony” has been replaced by “circadian period lengthening” in Fig.4H. Also replaced in the legend (now “lengthening of the circadian period”, page 14, line 543).

6. Fig. 5 and 6 show very modest and not statistically significant effect of obesity on BMAL1 binding to PER2 (Fig. 5C) and dramatic effect on promoters of other genes (Fig.6). The natural question arises: why the entire work is focused on PER2 and not on other transcriptional targets?

It is well established that PER2 is one of the most important factors for setting the period of the circadian clock (Chen et al., Mol Cell., 2009; Wilkins, et al., PLoS Comput Biol., 2007). In addition, tools available to monitor PER2 transcription have been extensively described and herein, we monitored its period in OAPs as a readout for clock function. Although we acknowledge that ChIP-Seq experiments, performed in a small number of patients, showed a very modest and non-significant effect of obesity on BMAL1 binding to PER2 promoter (Fig. 5C-D), conventional ChIP showed a striking decreased binding (~7-fold) in obese vs non-obese subjects (Fig 2B). Decrease in PER2 mRNAs (Fig 1G) and period lengthening of Per2 oscillations further support an effect of obesity on the clock repressor (Fig. 1H, I).

However, ChIP-Seq experiments enlightened a dramatic effect of obesity on changes in binding enrichment of BMAL1 at other loci (see fig. 6). We therefore did not focus on PER2 only, but rather expanded our focus to new BMAL1 targets, known to be involved in the pathogenesis of obesity and metabolic syndrome. We found that many genes with differential BMAL1 binding between non-obese and obese OAPs code for factors involved in metabolic inflammation were actually dysregulated at the mRNA and protein levels in obesity and could therefore contribute to the Metabolic Syndrome (new Fig. 7). (Results, page 7, lines 255-282).

The authors have responded in full to this comment and I believe addressed this concern in full.

No comments.

7. Data demonstrating no change in Ccl2 expression at the time of HFD-induced period lengthening must be shown.

Has been done. These data are now added as a new panel in Fig.9I.

This point has been addressed in full

No comments.

8. Does sodium salicylate have an effect on cycling in tissue culture?

Salicylate treatment given in vivo for 1 month to mice simultaneously rendered obese by a HFD

did not significantly change PER2::LUC period length in tissue explants cultured ex vivo (not shown). However, it should be emphasized that such salicylate treatment was of short duration and that period lengthening of PER2 oscillations increased with HFD duration (higher after 3 months than 1 month of HFD; Compare Fig 9G vs. 9H. We therefore cannot exclude the possibility that the experimental conditions were not optimal to address this issue.

In order to address the reviewer's point, could in vitro experiments be performed to ensure no acute effects of salicylate on period length? Alternatively, could the authors state in the text that they do not know the effects of salicylate on clock function in healthy tissue?

Done. As required, we added the following sentence (Results, page 9 lines 340-341): "Yet, salicylate was not tested in fat from control mice."

9. Fig. 8J – no statistically significant differences, therefore the authors cannot state that the downregulation of Per2 expression was corrected.

The referee is right. We now re-write more carefully this sentence "the downregulation of Per2 expression was attenuated .." (now Fig.9K) (Results, page 9, lines 338-339).

This fully addresses the reviewer's comment, thanks.

No comments.

10. In real-time PCR data, how the expression was normalized? It says "GAPDH (???)", 18S or Gapdh. Were Gapdh levels affected by obesity? It is important to add this info to all mRNA quantitation data (i.e. Per2/Gapdh relative abundance).

As required, we added this information. Now, in the Method Section, we indicate that the expression of the studied gene is normalized to the reference gene: "normalized to GAPDH for human adipose tissue, adipocytes and OAPs, 18s for mouse adipose tissue and adipocytes and Gapdh for mouse APs" (page 21, lines 804-805). No more place to indicate this in the figure legends (limited to 350 words).

Whenever used, expression of the reference gene was not impacted by the conditions (no significant differences found between non-obese and obese patients, subcutaneous and omental fat, collection-time, and no significant effect of the treatments).

The authors have addressed this comment in full.

No comments.

Overall, the manuscript will benefit if after editing it will be submitted to a more specialized journal

We feel that we have suitably addressed all the criticisms raised by the referees. Our data are novel and the main message of this paper is of interest for a broad readership.

I believe the manuscript is of interest to a broad readership with interests including circadian biology, metabolism and inflammation.

No comments.